# Reason-RFT: Reinforcement Fine-Tuning for Visual Reasoning of Vision Language Models

**Huajie Tan**[1,2,*], **Yuheng Ji**[2,3,4,*], **Xiaoshuai Hao**[2,*], **Xiansheng Chen**[2,*],
**Pengwei Wang**[2,†], **Zhongyuan Wang**[2], **Shanghang Zhang**[1,2,✉]

[1] State Key Laboratory of Multimedia Information Processing, School of Computer Science, Peking University
[2] Beijing Academy of Artificial Intelligence [3] Institute of Automation, Chinese Academy of Sciences
[4] School of Artificial Intelligence, University of Chinese Academy of Sciences

## Abstract

Visual reasoning abilities play a crucial role in understanding complex multimodal data, advancing both domain-specific applications and artificial general intelligence (AGI). Existing methods improve Vision-Language Models (VLMs) reasoning via Chain-of-Thought (CoT) supervised fine-tuning, using meticulously annotated training data to enhance visual reasoning capabilities. However, this training paradigm may lead to overfitting and cognitive rigidity, restricting the model's generalization ability to transfer visual reasoning skills under domain shift and limiting its real-world applicability. To address these limitations, we propose **Reason-RFT**, the first two-stage reinforcement fine-tuning framework for visual reasoning: (1) Supervised Fine-Tuning (SFT) with curated CoT data activates the reasoning potential of VLMs, followed by (2) Group Relative Policy Optimization (GRPO)-based reinforcement learning that generates multiple reasoning-response pairs, significantly enhancing the capability to address ubiquitous domain shift in visual reasoning tasks. To evaluate the visual reasoning capabilities of *Reason-RFT*, we reconstructed a comprehensive dataset encompassing visual counting, structural perception, and spatial transformation, serving as a benchmark for systematic assessment across three core dimensions. Experimental results demonstrate three key advantages: (1) *Performance Enhancement*: achieving state-of-the-art results across multiple tasks, outperforming mainstream open-source and proprietary models; (2) *Generalization Superiority*: consistently maintaining robust performance in addressing domain shift in typical visual reasoning tasks, outperforming alternative paradigms; (3) *Data Efficiency*: excelling in few-shot learning scenarios while surpassing full-dataset SFT baselines. *Reason-RFT* introduces a robust training paradigm in visual reasoning, and please refer to project website: Reason-RFT.

## 1 Introduction

Visual reasoning is pivotal for understanding complex multimodal data and advancing artificial general intelligence (AGI) [1, 2], making it a central focus in intelligent systems research. Recent advancements in image recognition [3–9], interactive security [10–12] and scene understanding [13, 14] have enabled transformative applications in healthcare [15, 16], robotics [17–24], and autonomous driving [25–31]. Consequently, enhancing visual reasoning capabilities has garnered significant attention from both industry and academia for its potential to drive transformative advancements.

Researchers have explored two primary categories of methods to enhance visual reasoning capabilities: (1) neural-symbolic methods [32–36], which integrate symbolic reasoning with neural networks to improve interpretability and modularity, and (2) Supervised Fine-Tuning (SFT) based on vision-language models (VLMs) [37, 38], which utilize end-to-end training to strengthen reasoning abilities.

---

* Equal contribution. † Project leader. ✉ Corresponding author: shanghang@pku.edu.cn.

39th Conference on Neural Information Processing Systems (NeurIPS 2025).

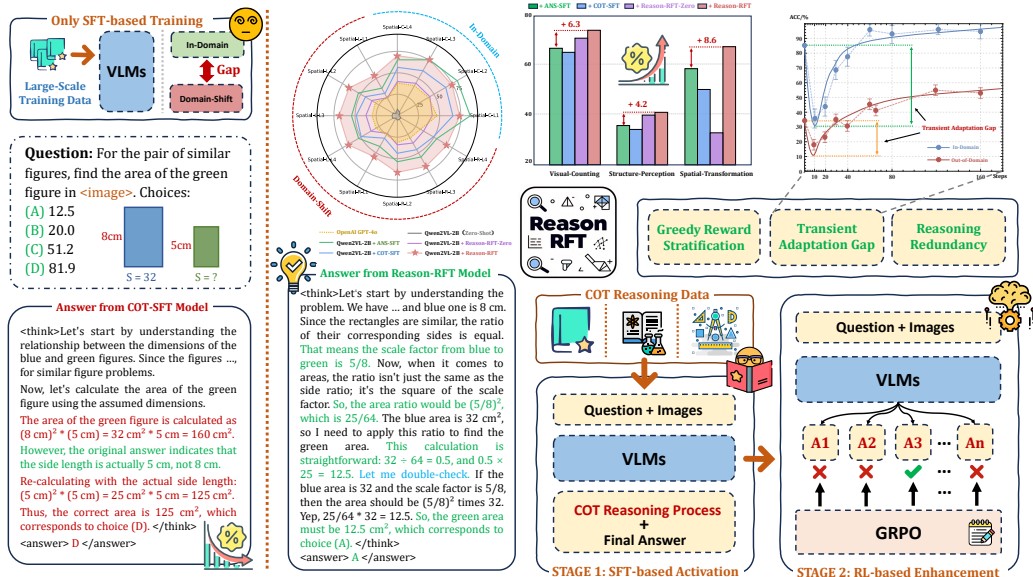

Figure 1: **Overview of Reason-RFT.** Compared to traditional SFT-based methods, our proposed Reason-RFT framework demonstrates superior generalization in visual reasoning tasks, excelling in reasoning improvement, domain shift adaptability, and data efficiency.

However, both approaches face significant limitations. Neural-symbolic methods are hindered by high complexity and a strong reliance on program generation, while SFT is constrained by its dependence on high-quality Chain-of-Thought (CoT) annotated data and meticulously designed data mixing strategies, leading to issues such as overfitting, cognitive rigidity, and limited adaptability to domain shift. These challenges reduce their effectiveness in real-world applications.

Recent advances such as GPT-o1 [2], DeepSeek-R1 [39], and Kimi-1.5 [40] show that reinforcement learning (RL) during post-training enhances reasoning in coding and mathematics. RL offers a dynamic alternative to SFT by enabling exploration and feedback-driven optimization, which can improve performance with limited labeled data. However, pure RL methods often lack robustness to domain shifts—such as changes in visual appearance or configuration, limiting their generalization capacity in real-world visual reasoning scenarios.

To address this, we propose *Reason-RFT*, the first two-stage reinforcement fine-tuning framework designed to enhance generalization in visual reasoning tasks. First, we employ SFT with CoT reasoning to activate the model's potential reasoning capabilities, using a high-quality domain-specific visual reasoning dataset tailored to stimulate related reasoning abilities. Subsequently, we further enhance reasoning potential through Group Relative Policy Optimization (GRPO), demonstrating that *Reason-RFT* improves robustness under distribution shifts by enhancing the model's reasoning capabilities. To evaluate its effectiveness, we constructed a high-quality dataset covering visual counting, structure perception, and spatial transformation, serving as a benchmark for evaluating three core capabilities of visual reasoning. Extensive experiments highlight three key advantages of *Reason-RFT*: (1) **Performance Improvement**: It achieves strong results on visual reasoning tasks, including visual counting, structure perception, and spatial transformation, outperforming mainstream VLMs; (2) **Enhanced Generalization**: It consistently exceeds both SFT-only and RL-only baselines under domain shift conditions, as demonstrated through comprehensive evaluations; (3) **Data Efficiency**: It reaches over 90% of the SFT-only performance while using less than 5% of the data. These results underscore the effectiveness and efficiency of *Reason-RFT*, establishing it as a robust framework for advancing visual reasoning. Our main contributions are summarized as follows.

- We introduce *Reason-RFT*, a two-stage reinforcement fine-tuning framework that significantly enhances the visual reasoning capabilities of VLMs by effectively combining the complementary strengths of SFT-based and RL-based methods.

- We provide a systematic analysis of SFT-based and RL-based paradigms on visual reasoning tasks, identifying the limitations of SFT and the advantages of RL in improving reasoning ability, handling domain shifts, and achieving data-efficient learning.

- We reconstruct a comprehensive dataset spanning three core domains: visual counting, structure perception, and spatial transformation, serving as a benchmark for evaluating visual cognition, geometric understanding, and spatial generalization.

- Extensive experiments validate the proposed framework, demonstrating its practicality and effectiveness, and providing a new perspective for reinforcement-driven multi-modal training.

## 2  Related Work

**Visual Reasoning** Visual reasoning is a core challenge in advancing AGI, requiring models to perform complex cognitive tasks grounded in visual perception [1, 41–50]. It underpins a wide range of applications, including visual counting [1, 51], geometric problem-solving [41, 52–55], visual transformation reasoning [56, 57], scientific analysis [58, 59], and robotic task planning [17, 60, 61]. Traditional approaches rely on program generation [36, 62, 63] or neural-symbolic reasoning [32–35], while recent advances in VLMs leverage large language models (LLMs) to enhance reasoning capabilities. For instance, LLaVA-CoT [37] employs multi-stage SFT with CoT prompting [64], and Insight-V [65] integrates SFT with RL. DeepSeek-R1-Zero [66] further introduces a rule-guided RL framework that substantially improves reasoning performance. Building upon the DeepSeek-R1 [66], our work provides a comparative analysis of SFT-based and RL-based paradigms, demonstrating the advantages of R1-style methods in enhancing visual reasoning.

**Post-Training** Post-Training is a crucial phase for enhancing the performance of LLMs and VLMs, bridging pre-trained models and their real-world applications [67–70]. It primarily involves two methodologies: *SFT* [71, 72] and *RL* [73–77]. SFT adapts pre-trained models to specific tasks using task-oriented datasets, often formatted as instructions. Research like FLAN [78] highlights the importance of diverse instruction-tuning datasets for improving zero-shot performance, while iterative processes, such as Llama 3.1's six-round strategy [79], integrate rejection sampling, synthetic data, and human annotations. RL aligns models with human preferences or task-specific goals through feedback mechanisms. Reinforcement Learning from Human Feedback (RLHF) [74] refines models using human preference data, as seen in Llama 3.1 [79] and Nemotron-4 [80], which use reward modeling techniques like DPO [81] and RPO [80]. For example, TÜLU3 [82] employs length-normalized DPO, while DeepSeek-V3 [83] combines rule-based and model-based reward systems. Recently, DeepSeek-R1 [66] achieved significant text reasoning improvements through pure RL [84]. Our work first adapts R1 methodologies to VLMs, enhancing visual reasoning, and systematically compares SFT-based and RL-based paradigms in visual reasoning tasks.

## 3  Methodology

In this section, we introduce ***Reason-RFT***, a novel two-stage training strategy to enhance the reasoning capabilities of VLMs in complex visual reasoning tasks. As shown in Fig. 2, the framework comprises two stages: (1) *SFT-based Visual Reasoning Activation*, which uses SFT with high-quality CoT reasoning data to activate the model's domain-specific reasoning capabilities, and (2) *RL-based Reasoning Enhancement*, which employs the GRPO algorithm with rule-based rewards to further push the upper limits of the model's reasoning potential.

### 3.1  STAGE 1: SFT-based Reasoning Activation

In the initial stage, we employ SFT on a structured visual reasoning dataset containing step-by-step reasoning processes. This stage trains the model to decompose complex tasks into logical steps. Each sample is represented as $(x, q, r, a)$, where $x$ denotes the input images, $q$ is the question, $r$ is the reasoning steps, and $a$ is the final answer. The training objective maximizes the likelihood of generating both $r$ and $a$ given $(x, q)$:

$$\mathcal{L}_{\text{SFT}} = -\mathbb{E}_{(x,q,r,a) \sim \mathcal{D}} \sum_{t=1}^{T} \log \pi_\theta(y_t \mid x, q, y_{<t}), \tag{1}$$

where $\mathcal{D}$ denotes the dataset, $y$ represents the concatenated sequence of $r$ and $a$, and $\pi_\theta$ denotes the model's token distribution. The resulting model $\pi_{\text{CoT}}$ is used to initialize the subsequent stage, providing a stable foundation for RL-based reasoning enhancement.

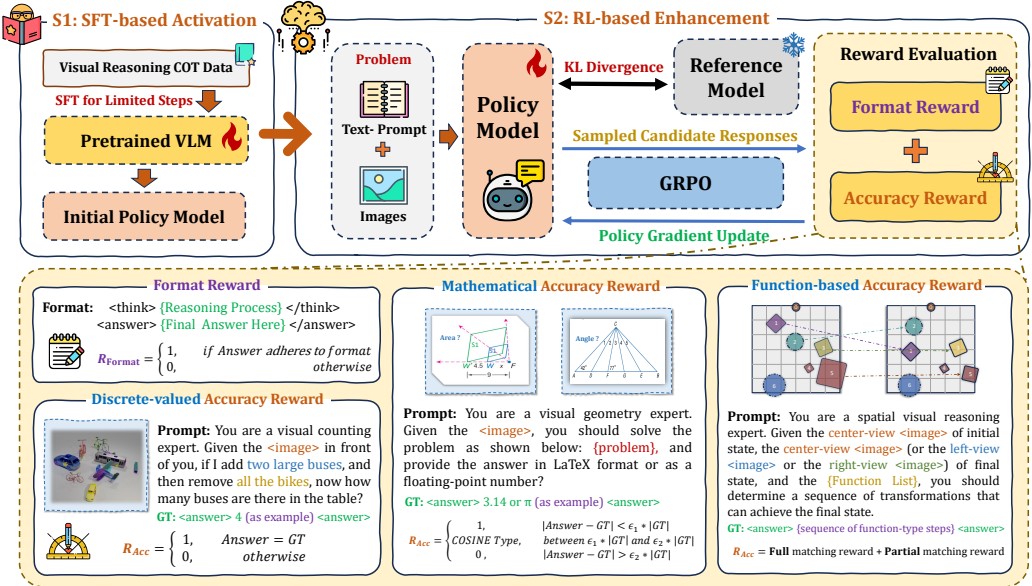

Figure 2: **Framework of Reason-RFT**. Reason-RFT adopts a two-stage training paradigm for visual reasoning. The first stage applies SFT with CoT reasoning to establish strong task-specific priors. In the second stage, GRPO is employed to further enhance reasoning capability and generalization.

## 3.2 STAGE 2: RL-based Reasoning Enhancement

In the second stage, we refine $\pi_{\text{CoT}}$ using GRPO, leveraging RL for its efficiency and scalability. Unlike Proximal Policy Optimization (PPO), which requires a computationally expensive value network, GRPO calculates relative advantages by comparing rewards within a group of sampled actions, reducing computational overhead and simplifying optimization. This makes GRPO particularly suitable for visual reasoning tasks.

**Sampling Action Groups** For each input state $s = (x, q)$, GRPO samples a group of actions $\{a_1, a_2, \ldots, a_G\}$ from the current policy $\pi_\theta$, initialized from $\pi_{\text{CoT}}$. The sampling process is:

$$a_i \sim \pi_\theta(a \mid x, q), \quad \text{for } i = 1, 2, \ldots, G. \tag{2}$$

This strategy ensures diverse responses, promoting exploration and preventing premature convergence.

**Reward Evaluation.** Each sampled action $a_i$ receives a reward $R(a_i)$ based on verifiable criteria, yielding a reward set $\{r_1, r_2, \ldots, r_G\}$. For visual reasoning tasks, the reward $R(a_i)$ is composed of a format reward $R_{\text{format}}(a_i)$, which enforces structured outputs, and an accuracy reward $R_{\text{acc}}(a_i)$, which measures correctness. This formulation balances structural alignment and factual precision in reasoning. The reward function is defined as:

$$R(a_i) = R_{\text{format}}(a_i) + R_{\text{acc}}(a_i). \tag{3}$$

**Policy Update with Relative Advantage** Rewards are normalized within the sampled group to compute relative advantages $\{A_1, A_2, \ldots, A_G\}$, defined as:

$$A_i = \frac{r_i - \text{mean}\{r_1, r_2, \ldots, r_G\}}{\text{std}\{r_1, r_2, \ldots, r_G\}}. \tag{4}$$

Based on these advantages, the policy is updated to reinforce actions with positive advantages and reduce the probability of less effective ones. To maintain training stability, the update is constrained by minimizing the KL divergence between the updated and reference policies.

## 3.3 Reward Design for Visual Reasoning Tasks

For the diverse requirements of visual reasoning tasks, including visual counting, structure perception, and spatial transformation, our reward design integrates two essential components: *format reward* and *accuracy reward*. The format reward is uniformly applied across all tasks, ensuring that the

model strictly adheres to a structured response format for consistency. For the accuracy reward, we carefully tailor the design to the specific characteristics of each task, as shown in Fig. 2, creating task-specific reward mechanisms to evaluate the correctness of the model's responses.

**Format Reward** This component ensures structured and interpretable responses by requiring the model to adhere to a predefined template: reasoning within `<think>` and `</think>` and the final answer within `<answer>` and `</answer>`. A reward of 1 is only given for strict adherence.

**Accuracy Reward** This component evaluates the correctness of the model's responses, ensuring alignment with ground truth across diverse visual reasoning tasks. To address task diversity, we design tailored reward mechanisms for discrete-valued, mathematical, and function-based problems. Each mechanism is crafted to handle the unique characteristics of its problem category, enabling precise and fair evaluation. Below, we introduce the three reward forms.

- **Discrete-valued Type** This reward type applies to visual counting and structure perception tasks, where answers are discrete values (*e.g.*, multiple-choice or integer-based responses). The accuracy reward $R_{\text{acc}}(a_i)$ is defined as:

$$R_{\text{acc}}(a_i) = \begin{cases} 1, & \text{if } a_{\text{pred}} = a_{\text{gt}} \\ 0, & \text{otherwise,} \end{cases} \tag{5}$$

  where $a_{\text{pred}}$ is the predicted answer and $a_{\text{gt}}$ is the ground truth. This discrete reward penalizes deviations from ground truth, ensuring precision in tasks requiring unambiguous answers.

- **Mathematical Type** This reward type is designed for structure perception tasks involving numerical answers, such as floating-point values or LaTeX-formatted expressions. It uses a tolerance-based evaluation to account for minor numerical deviations, which is defined as:

$$R_{\text{acc}}(a_i) = \frac{1}{2} \left[ \cos \left( \pi \times \frac{|a_{\text{pred}} - a_{\text{gt}}| - \epsilon_1 \times |a_{\text{gt}}|}{(\epsilon_2 - \epsilon_1) \times |a_{\text{gt}}|} \right) + 1 \right], \tag{6}$$

  where $a_{\text{pred}}$ is the predicted answer, $a_{\text{gt}}$ is the ground truth, $\epsilon_1$ is the tolerance threshold for an exact match (*e.g.*, 0.05), and $\epsilon_2$ is the upper bound for partial rewards (*e.g.*, 0.20). If $|a_{\text{pred}} - a_{\text{gt}}| < \epsilon_1 \times |a_{\text{gt}}|$, the reward is 1 (exact match); if $|a_{\text{pred}} - a_{\text{gt}}| > \epsilon_2 \times |a_{\text{gt}}|$, the reward is 0 (incorrect). This formulation ensures smooth transitions between full and partial rewards, enabling fair evaluation of numerical accuracy.

- **Function-based Type** This reward type is designed for spatial transformation tasks requiring a sequence of transformation functions. The accuracy reward $R_{\text{acc}}(a_i)$ evaluates the alignment between the predicted sequence $T_{\text{pred}}$ and the ground truth $T_{\text{gt}}$, computed as:

$$R_{\text{acc}}(a_i) = \frac{\text{len}(T_{\text{pred}}^{f+o+v}) + \alpha \cdot \text{len}(T_{\text{pred}}^{f+o/v}) + \beta \cdot \text{len}(T_{\text{pred}}^{f})}{\max(\text{len}(T_{\text{pred}}), \text{len}(T_{\text{gt}}))}, \tag{7}$$

  where $T_{\text{pred}}^{f+o+v}$ is the subset of transformation steps with complete matches (w/ function, object, and value), $T_{\text{pred}}^{f+o/v}$ are the subsets with partial and only-function matches (w/ function and object, or w/ function and value), $T_{\text{pred}}^{f}$ is the subset with only-function matches. $\alpha$ and $\beta$ are the weighting coefficients for partial matches. This formulation ensures nuanced evaluation for flexible adjustment of partial match contributions.

# 4 Experiments

We design experiments to investigate the following key research questions:

- **RQ1**: How effective is *Reason-RFT* in reasoning, generalization, and data efficiency?
- **RQ2**: Why is the STAGE 1 of SFT with CoT reasoning necessary?
- **RQ3**: Why is the STAGE 2 of reinforcement fine-tuning necessary?
- **RQ4**: How does reward design affect *Reason-RFT*'s performance?
- **RQ5**: What training dynamics emerge during reinforcement fine-tuning, and how do they shape the reasoning behavior of *Reason-RFT*?

### 4.1 Experimental Details

**Datasets** In this paper, we comprehensively evaluate the visual reasoning capabilities of our method by leveraging six existing datasets, enhanced through subtask categorization, error-prone data filtering, and dataset restructuring. Detailed protocols for data filtering and restructuring are provided in Sec. A. Specifically, we define three task categories as follows.

- **Visual Counting** is a multimodal reasoning task evaluating the integration of linguistic, visual, and mathematical skills by solving arithmetic problems in 3D block-based scenes. Specifically, we filtered and corrected 35K samples from CLEVR-Math [1] for training and 1K test samples for in-domain (ID) evaluation. To assess generalization under domain-shift (DS), we constructed 1K new samples using 3D assets from Super-CLEVR [51], including two subsets: direct arithmetic (DS-D) and mixed arithmetic (DS-M). Refer to the Appendix Sec. A.1 for details.

- **Structure Perception** is a structural reasoning task requiring models to analyze relationships in geometries, imaging structures, chart layouts, and architectural designs. We filtered 4.5K training samples and 820 ID test samples from Geo170K [52] and Math360K [55], along with 800 samples from Geometry3K [85] to evaluate DS adaptability. See the Appendix Sec. A.2.

- **Spatial Transformation** is a spatial-visual reasoning task requiring models to infer single- or multi-step transformations by analyzing initial and final states of 3D scenes from multiple perspectives (*e.g.*, center, left, right). We generated 100K samples using TRANCE [56], covering four difficulty levels, and selected 60K for training and 6K for testing through a specific filtering process. For DS evaluation, identical scenes are rendered from left/right viewpoints (DS-L/R) to test perspective-change robustness. Details can be found in the Appendix Sec. A.3.

**Evaluation Metrics** We use accuracy-rate (Acc) as the primary metric [86]. For numerical answers, correctness is verified by mathematical equivalence to the ground truth. For multiple-choice questions, we perform a string match. For function-type sequences, we use stepwise multi-level evaluation.

**Implementation Details** We utilize Qwen2-VL-2B and Qwen2-VL-7B [87] as the backbone models for our experiments. Our implementation is built on the open-source frameworks Open-R1 [88] and vLLM [89], ensuring reproducibility and scalability. All experiments were conducted on a cluster of servers, each equipped with 8×A800 GPUs. For further details, see the Appendix Sec. B.

**Training Paradigms and Baselines** To assess the performance and generalization of different training strategies, we compare: (1) SFT-based methods—ANS-SFT, which fine-tunes on answer generation, and CoT-SFT, which uses supervised learning with CoT reasoning; and (2) RL-based methods—Reason-RFT-Zero, which applies RL without reasoning activation stage, and Reason-RFT, which uses limited CoT data for reasoning activation before RL training. For comprehensive experiments, we use Qwen2-VL-Instruct [87] as the base model (both 2B and 7B variants). In addition, we also select the most advanced open-source models [90–94] and the proprietary models [95, 96] as baselines to evaluate the performance of different paradigms.

### 4.2 Overall Evaluation of Reason-RFT Framework (RQ1)

To evaluate Reason-RFT, we evaluate Reason-RFT using 2B- and 7B-parameter models on three visual reasoning tasks. The results are summarized as follows.

**Strong reasoning performance across ID tasks.** As shown in Tab. 1, Reason-RFT achieves performance comparable to or better than both SFT- and RL-based methods across all tasks. In *visual counting*, Reason-RFT-Zero achieves the best performance among all models in the 7B setting. In *structure perception*, Reason-RFT outperforms most open-source and proprietary baselines in the 7B setting and remains competitive with top models such as InternVL-2.5-8B [92]. In *spatial transformation*, Reason-RFT matches or exceeds SFT-based methods while consistently outperforming all baselines. These results demonstrate that Reason-RFT effectively integrates the strengths of both SFT and RL in structured reasoning tasks.

**Superior generalization under DS.** Under DS settings, Reason-RFT shows substantial gains over both traditional baselines and alternative training paradigms. In *visual counting*, it outperforms ANS-SFT by 10.95% (2B) and 13.93% (7B). In *structure perception*, Reason-RFT achieves the highest performance in the 2B model, with an 6.93% gain over CoT-SFT, and remains highly competitive in

Table 1: **Results on three visual reasoning tasks.** The best results among different training paradigms are highlighted in **bold**, while the second-best results are underlined. "ID" denotes in-domain test data, and "DS" denotes domain-shift test data.

| Method | Visual Counting | | | | Structure Perception | | | Spatial Transformation | | | |
|---|---|---|---|---|---|---|---|---|---|---|---|
| | ID | DS-D | DS-M | AVG | ID | DS | AVG | ID | DS-L | DS-R | AVG |
| **Proprietary Models** | | | | | | | | | | | |
| GPT-4o-2024-08-06 [95] | 68.10 | 42.54 | 9.60 | 40.08 | 50.18 | 43.49 | 46.83 | 42.55 | 28.67 | 29.76 | 35.88 |
| Gemini-1.5-Pro [96] | 61.80 | 41.20 | 26.40 | 43.13 | 50.12 | 48.38 | 49.45 | 26.22 | 18.76 | 19.88 | 22.77 |
| **Open-Source Models** | | | | | | | | | | | |
| Qwen2.5-VL-3B-Inst. [90] | 75.90 | 50.93 | 4.40 | 43.74 | 36.75 | 37.44 | 37.09 | 8.57 | 8.26 | 8.31 | 8.42 |
| Phi-3.5-Vision-4B-Inst. [91] | 21.40 | 18.27 | 6.00 | 15.22 | 36.83 | 50.25 | 43.54 | 7.42 | 2.45 | 4.02 | 5.33 |
| Qwen2.5-VL-7B-Inst. [90] | 74.60 | 46.00 | 2.80 | 41.13 | 44.00 | 45.61 | 44.80 | 19.63 | 13.12 | 13.42 | 16.45 |
| InternVL-2.5-8B [92] | 93.50 | 46.13 | 2.80 | 47.48 | 63.00 | 47.32 | 51.60 | 7.19 | 6.62 | 6.63 | 6.91 |
| Llama-3.2-11B-Vision [93] | 10.30 | 9.60 | 9.20 | 9.70 | 13.75 | 20.85 | 17.30 | 8.22 | 8.40 | 9.03 | 8.47 |
| Pixtral-12B [94] | 42.60 | 25.33 | 15.60 | 27.84 | 30.38 | 36.09 | 33.23 | 7.35 | 5.03 | 5.22 | 6.42 |
| **Qwen2VL-2B-Instruct** | | | | | | | | | | | |
| Zero-Shot | 82.40 | 42.67 | 0.00 | 41.69 | 25.86 | 20.63 | 23.25 | 3.78 | 4.60 | 4.67 | 4.35 |
| + ANS-SFT | 96.20 | 51.07 | 5.20 | 50.82 | **51.34** | 22.50 | 36.92 | **77.39** | 49.24 | 50.33 | 58.99 |
| + CoT-SFT | 85.50 | 49.73 | **36.80** | 57.34 | 43.05 | 25.25 | 34.15 | 64.37 | 43.19 | 42.86 | 50.14 |
| + Reason-RFT-Zero | **98.40** | 58.00 | 5.20 | 53.87 | 47.68 | 32.50 | 40.09 | 42.13 | 34.07 | 33.41 | 33.74 |
| + Reason-RFT | 96.80 | **60.00** | 28.40 | **61.77** | 49.03 | 33.13 | **41.08** | 74.61 | **64.05** | **64.08** | **67.58** |
| **Qwen2VL-7B-Instruct** | | | | | | | | | | | |
| Zero-Shot | 98.60 | 54.53 | 4.80 | 52.64 | 43.30 | 43.88 | 43.59 | 13.53 | 12.72 | 12.78 | 13.01 |
| + ANS-SFT | 95.00 | 42.53 | 8.00 | 48.51 | 51.34 | 25.38 | 38.36 | **82.19** | 54.29 | 54.83 | 63.77 |
| + CoT-SFT | 87.30 | 45.33 | 33.60 | 55.41 | 50.49 | 33.00 | 41.75 | 81.31 | 47.90 | 47.80 | 59.00 |
| + Reason-RFT-Zero | **99.40** | **63.60** | 21.20 | 61.40 | 55.00 | **54.75** | **54.88** | 67.67 | 57.20 | 56.15 | 60.34 |
| + Reason-RFT | 95.60 | 56.13 | **35.60** | **62.44** | **59.27** | 49.25 | 54.26 | 79.97 | **59.36** | **58.61** | **65.98** |

the 7B model. Most notably, in *the spatial transformation* task, the 2B Reason-RFT model surpasses GPT-4o [95] by 31.7%, showcasing remarkable generalization under DS.

**High training efficiency.** To evaluate data efficiency during training, we train all methods on the *spatial transformation* task and monitor intermediate ID and DS performance (Fig. 3). Additional results for *visual counting* and *structure perception* are provided in Appendix Sec. C. In the 2B model, Reason-RFT achieves 70% of the final performance of Reason-RFT-Zero using only 3% of the training data (1,600 samples), and reaches 82.5% with 9% of the data. In the 7B model, it achieves over 92% of Reason-RFT-Zero's performance using just 3% of the data. These results confirm that Reason-RFT exhibits strong training efficiency in both ID and DS scenarios. The data-efficient nature of Reason-RFT renders it particularly effective for few-shot learning, offering significant potential for applications where labeled data is limited.

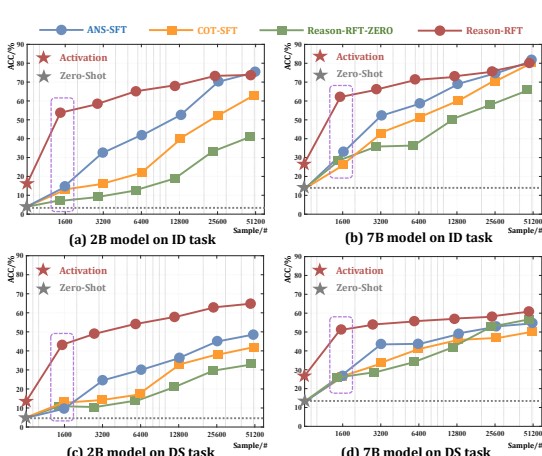

Figure 3: Results of different methods and model sizes on *spatial transformation* task across training.

### 4.3 Effect of STAGE 1 on Initialization (RQ2)

To investigate the role of CoT-SFT in initialization, we compare four baselines across three tasks. The results in Tab. 1 reveal the following: *(1) Consistent performance gains from CoT-SFT.* Across all three tasks and both 2B and 7B model scales, Reason-RFT consistently outperforms Reason-RFT-Zero following stage 1 reasoning activation. This improvement is particularly notable when the model is small and the task involves complex output structures. For example, in the *spatial transformation* task—which requires function-like serialized outputs—the 2B Reason-RFT model surpasses Reason-RFT-Zero by 33.84%. *(2) Smaller models benefit more from CoT-SFT priors.* In the *visual counting* task under the DS-M setting, the 2B model with CoT-SFT outperforms Reason-RFT-Zero by 31.6%.

Although the gap narrows in the 7B model, CoT-SFT still yields substantial gains. This indicates that pure RL-based methods struggle to adapt from direct arithmetic to mixed arithmetic reasoning under DS, whereas CoT-SFT provides effective inductive priors for such adaptation. Moreover, under the same amount of CoT-SFT data, the 2B model still underperforms its 7B counterpart, highlighting the increased reliance of smaller models on CoT-SFT for acquiring reasoning capabilities.

### 4.4 Effect of STAGE 2 on Generalization (RQ3)

To evaluate the impact of RL in stage 2, we compare the generalization performance of CoT-SFT and Reason-RFT across three visual reasoning tasks under DS. As shown in Tab. 1 and Fig. 4, the results reveal the following: ***Reinforcement fine-tuning improves generalization beyond CoT-SFT.*** Across all domain-shift settings, Reason-RFT consistently outperforms CoT-SFT, demonstrating that reinforcement learning significantly enhances model robustness. For instance, in the *visual counting* task, the 7B Reason-RFT achieves a combined DS-D and DS-M score 12.8% higher than CoT-SFT. The improvement is even more pronounced in structure-sensitive tasks such as *spatial transformation*, where the 2B Reason-RFT exceeds CoT-SFT by 21.04% on average across DS-L and DS-R. These results indicate that CoT-SFT alone yields limited generalization, while reinforcement fine-tuning enables better adaptation to compositional and layout-dependent variations.

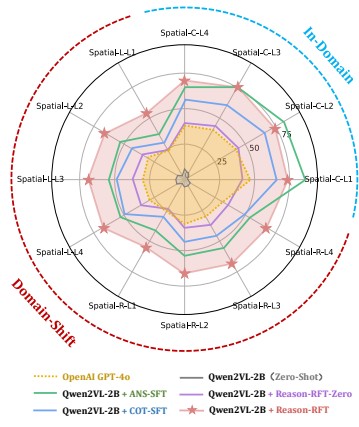

Figure 4: Results of DS *v.s.* ID on *spatial transformation* task.

### 4.5 Exploration on Reward Design (RQ4)

**Format Reward.** In DeepSeek-R1 [39], the format reward enforces the use of `<think>` and `<answer>` tokens to structure reasoning in textual tasks. To better support visual reasoning, we extend this with `<summary>` and `<caption>` tokens to incorporate visual observations via caption-style prompts. As shown in Tab. 2, this improves Reason-RFT-Zero but has limited effect on Reason-RFT. We attribute this to Reason-RFT's prior CoT supervision, which likely helps it internalize caption-like structures in stage 1, reducing the benefit of explicit tags. In contrast, Reason-RFT-Zero benefits more from such structural cues, indicating greater sensitivity to format-level guidance.

Table 2: Results of different format reward strategies on the *spatial transformation* task.

| Setting | ID | DS-L | DS-R | AVG |
|---|---|---|---|---|
| **Qwen2VL-2B-Instruct** | | | | |
| Reason-RFT-Zero | **42.13** | 34.07 | 33.41 | 33.74 |
| + visual tokens | 42.01 | **36.05** | **35.97** | **38.01** |
| Reason-RFT | **74.61** | **64.05** | **64.08** | **69.33** |
| + visual tokens | 71.99 | 60.13 | 59.87 | 65.99 |
| **Qwen2VL-7B-Instruct** | | | | |
| Reason-RFT-Zero | 67.67 | 57.2 | 56.15 | 62.17 |
| + visual tokens | **70.28** | **59.52** | **57.01** | **64.27** |
| Reason-RFT | **79.97** | **59.36** | **58.61** | **69.48** |
| + visual tokens | 79.85 | 58.71 | 57.98 | 69.09 |

**Accuracy Reward.** We explore accuracy reward design in the *spatial transformation* task, which requires predicting transformation sequences in a structured format. The formulation in Eq. 7 introduces coefficients $\alpha$ and $\beta$ to control tolerance for partial matches. We test three settings: (1) $\alpha = 0, \beta = 0$ (exact match only), (2) $\alpha = 0.50, \beta = 0.25$ (partial credit), and (3) $\alpha = -0.25, \beta = -0.50$ (penalized partial matches). Results on 2B and 7B models (Tab. 3) show that: (1) partial credit improves ID performance but harms generalization, suggesting "soft rewards" reduce ro-

Table 3: Results of different accuracy reward strategies on the *spatial transformation* task.

| Setting | $\alpha$ | $\beta$ | ID | DS-L | DS-R | AVG |
|---|---|---|---|---|---|---|
| **Qwen2VL-2B-Instruct** | | | | | | |
| Baseline | 0 | 0 | 74.61 | 64.05 | 64.08 | **69.33** |
| (a) | 0.50 | 0.25 | **79.18** | 56.36 | 55.45 | 67.54 |
| (b) | -0.25 | -0.50 | 73.69 | **64.41** | **64.72** | 69.13 |
| **Qwen2VL-7B-Instruct** | | | | | | |
| Baseline | 0 | 0 | 79.97 | 59.36 | 58.61 | 69.48 |
| (a) | 0.50 | 0.25 | **80.89** | 53.20 | 52.61 | 66.90 |
| (b) | -0.25 | -0.50 | 75.03 | **64.83** | **63.18** | **69.52** |

bustness; (2) penalizing partial matches improves generalization under domain shift, indicating "hard rewards" better support serialized reasoning.

### 4.6 Training Dynamics and Reasoning Behavior Analysis (RQ5)

**Greedy Reward Stratification.** This phenomenon captures the model's tendency, particularly in Reason-RFT-Zero, to prioritize easier-to-optimize rewards (e.g., format reward) before addressing more challenging objectives (e.g., accuracy reward). As shown in Fig. 5, the reasoning token length initially drops, then gradually increases and stabilizes. This dynamic correlates with the format

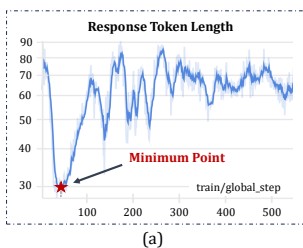 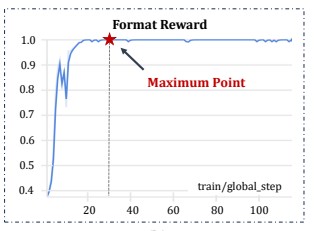 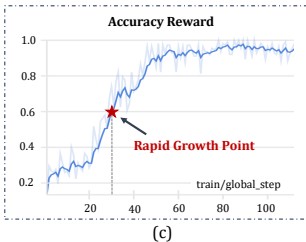

(a)       (b)       (c)

Figure 5: **Analysis of Greedy Reward Stratification.** The model's reasoning token length first decreases, then gradually rises and stabilizes during Reason-RFT-Zero training. The peak of the format reward coincides with the accuracy reward's rapid growth phase.

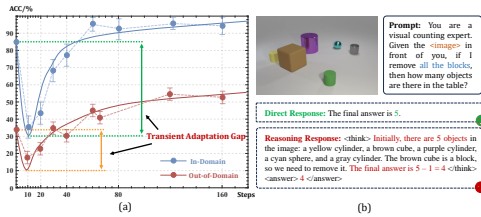 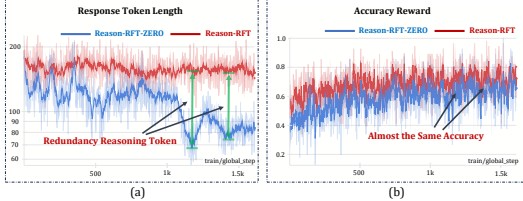

Figure 6: **Illustration of the Transient Adaptation Gap.** (a) shows a sharp drop and recovery in both ID and DS test performances within the early training steps. (b) shows a case study of the prediction result on early step.

Figure 7: **Analysis of Reasoning Redundancy.** (a) shows the reasoning token length curves for Reason-RFT-Zero and Reason-RFT during training. (b) displays their accuracy reward, with both paradigms converging to similar accuracy rate.

reward quickly reaching a plateau, followed by a sharp rise in the accuracy reward. We infer that the model initially simplifies its outputs to adapt rapidly to structured format expectations, and only later allocates learning capacity to improve reasoning correctness.

**Transient Adaptation Gap.** This refers to a temporary performance degradation observed in the early training phase of Reason-RFT-Zero. When the model shifts from directly predicting answers to producing structured reasoning traces, it experiences a brief adaptation bottleneck—marked by a sharp decline and gradual recovery in accuracy. Fig. 6 (a) illustrates this drop within the first 100 steps on the *visual counting* task under both ID and DS settings. A case analysis in Fig. 6 (b) further reveals that forcing structured reasoning prematurely may lead to incorrect outputs, highlighting the sensitivity of early-stage training to reasoning format constraints.

**Reasoning Redundancy.** This phenomenon concerns the discrepancy in reasoning token length between models trained with and without CoT activation. In the *structure perception* task, Reason-RFT and Reason-RFT-Zero attain similar accuracy, yet the former generates significantly longer reasoning traces (Fig. 7). This likely stems from Reason-RFT's use of CoT data distilled from advanced models (e.g., GPT-4o), which encourages verbose reasoning during stage 1. In the absence of penalties or length control in reinforcement fine-tuning, such verbosity persists. By contrast, Reason-RFT-Zero converges to more concise reasoning through reward-driven exploration. We hypothesize that these longer chains in Reason-RFT may introduce unnecessary computational overhead or reflect overthinking relative to task complexity.

## 5 Conclusion

In this paper, we propose ***Reason-RFT***, a novel reinforcement fine-tuning framework that enhances the generalization capabilities of visual reasoning models. By integrating SFT with CoT reasoning activation data and GRPO-based reinforcement learning, *Reason-RFT* effectively mitigates key challenges such as overfitting and cognitive rigidity, thereby improving cross-domain transferability and real-world applicability. To support systematic evaluation, we reconstruct a comprehensive dataset covering visual counting, structure perception, and spatial transformation tasks, establishing a robust benchmark for assessing model performance across diverse scenarios. Extensive experiments demonstrate the effectiveness of *Reason-RFT*, providing valuable insights for advancing visual reasoning research and introducing a new paradigm in multimodal learning.

## Acknowledgments

This work was supported by the National Natural Science Foundation of China (62476011), and the National Science and Technology Major Project (No. 2022ZD0117800).

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

# Appendix

This supplementary material provides additional details on the proposed method and experimental results that could not be included in the main manuscript due to page limitations. Specifically, this appendix is organized as follows.

- Sec. A provides more details on the evaluation of reasoning tasks and discusses how we collected, filtered, and reconstructed a high-quality dataset.
- Sec. B outlines the models and training processes, providing more detailed experimental specifics.
- Sec. C presents comprehensive experimental results.
- Sec. D details the pipeline of CoT date generation.
- Sec. E presents detailed composition of diffenernt mixed CoT datasets.
- Sec. F shows the comparison of CoT quality before and after RL.
- Sec. G includes more visualization cases.
- Sec. H introduces the limitations of our Reason-RFT and its societal impact.

## A   Details of Evaluation Reasoning Tasks

### A.1   Visual Counting

**Task Definition** Visual Counting is a multimodal reasoning task that evaluates the integration of linguistic, visual, and mathematical capabilities by requiring models to solve arithmetic problems in dynamic visual scenes composed of 3D blocks with diverse attributes, including color, size, material, and shape. The task consists of four distinct reasoning types: **1) Subtraction**, which involves counting objects after removing a specified subset based on given attributes; **2) Addition**, where models must compute totals after inserting new objects with defined quantities and properties; **3) Adversarial**, a challenging variant designed as trick questions in which operations are performed on one set of objects while the query targets an unrelated or unaffected subset, testing the model's robustness against deceptive scenarios; and **4) Multi-Hop**, which requires sequential reasoning through multiple addition or subtraction steps to arrive at the final count. This task challenges models to perform attribute-based reasoning in dynamic visual contexts, emphasizing cross-modal understanding and reasoning capabilities. Some examples are shown in Fig. 8.

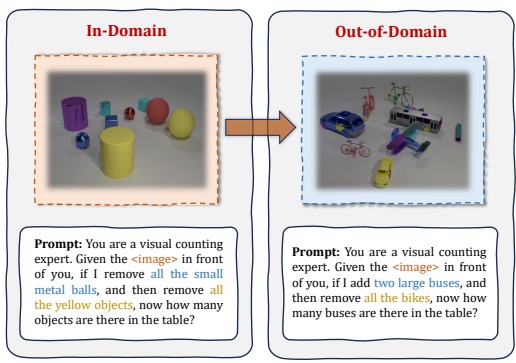

Figure 8: The sample of Visual Counting.

**Dataset Preparation** For In-Domain (ID) dataset, we refined the original dataset from CLEVR-Math [1] by filtering out low-quality or incorrect samples using GPT-4o, resulting in a clean dataset comprising 35K training samples and 1K test samples. These samples are categorized into four specific types: subtraction, addition, adversarial, and multihop-subtraction. To evaluate Domain-Shift (DS) generalization, we extended CLEVR-Math by enhancing the diversity of objects through the incorporation of 3D assets from Super-CLEVR [51], which leads to the creation of Super-CLEVR-Math, an advanced benchmark with 1K test samples designed to assess model generalization under increased complexity. These test samples are also divided into four task types: addition, subtraction, subtraction-multihop and addition-subtraction. The test samples are further categorized into four task types: addition, subtraction, subtraction-multihop, and addition-subtraction. The first three constitute the DS-D subset, while addition-subtraction forms the DS-M subset. Notably, the mixed addition-subtraction type introduces a novel category consisting multi-steps of both addition and subtraction, which is not present in CLEVR-Math, further elevating the benchmark's challenge.

**Reward Design** Following the reward methodology of DeepSeek-R1 [39], we define two distinct reward functions: Format Reward and Accuracy Reward. The Format Reward is assigned

a value of 1 if the response adheres to the predefined template structure, specifically in the form of `<think>...</think><answer>...</answer>`; otherwise, it is assigned a value of 0. The Accuracy Reward is assigned a value of 1 if the numerical counting result in the response is correct; otherwise, it is assigned a value of 0. This dual-reward mechanism ensures both structural compliance and numerical accuracy in model responses.

## A.2 Structure Perception

**Task Definition** Structure Perception represents a complex class of visual mathematical reasoning tasks, which focuses on assessing the model's capacity to determine geometric structure relationships and perform calculations involving angles, lengths, areas, and other geometric properties. The task includes problems such as identifying congruent or similar shapes, calculating perimeters and areas, determining angles between lines or shapes, and solving problems related to geometric transformations (e.g., rotations, translations, and reflections). By combining mathematical rigor with visual reasoning, this task challenges models to demonstrate a deep understanding of geometric principles in both abstract and real-world scenarios. Some examples are shown in Fig. 9.

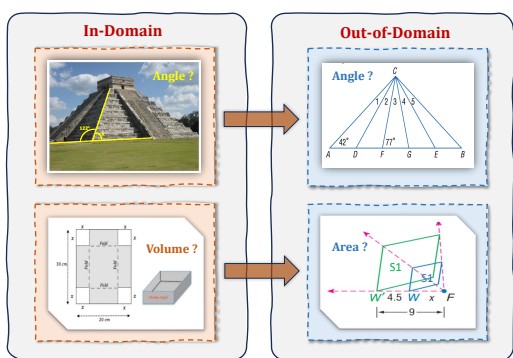

Figure 9: The sample of Structure Perception.

**Dataset Preparation** For the ID dataset, we provide GeoMath-4K5, a dataset specifically designed for geometric problem solving, which is constructed based on Math360K [55] and Geo170K [52]. To ensure data quality, we employed GPT-4o to filter out incorrect samples and removed those with answers that were neither numerical nor included in the provided options, thereby streamlining the validation process during training and testing. This refinement process resulted in a curated dataset consisting of 4.5K training samples and 820 test samples. For DS evaluation, we selected 800 samples from Geometry3K [85] (including 400 multiple-choice and 400 open-ended questions) to comprehensively assess the model's generalization capabilities on geometry reasoning.

**Reward Design** We maintain the same Format Reward as used in the Visual Counting task above. The Accuracy Reward is extended to support the evaluation of both multiple-choice questions and mathematical expressions, ensuring comprehensive assessment across various problem types. Specifically, mathematical reward type is designed for Structure Perception tasks involving numerical answers, such as floating-point values or LaTeX-formatted expressions. It uses a tolerance-based evaluation to account for minor numerical deviations. The accuracy reward $R_{\text{acc}}(a_i)$ is defined as:

$$R_{\text{acc}}(a_i) = \frac{1}{2}\left[\cos\left(\pi \times \frac{|a_{\text{pred}} - a_{\text{gt}}| - \epsilon_1 \times |a_{\text{gt}}|}{(\epsilon_2 - \epsilon_1) \times |a_{\text{gt}}|}\right) + 1\right], \tag{8}$$

where $a_{\text{pred}}$ is the predicted answer, $a_{\text{gt}}$ is the ground truth, $\epsilon_1$ is the tolerance threshold for an exact match (*e.g.*, 0.05), and $\epsilon_2$ is the upper bound for partial rewards (*e.g.*, 0.20). If $|a_{\text{pred}} - a_{\text{gt}}| < \epsilon_1 \times |a_{\text{gt}}|$, the reward is 1 (exact match); if $|a_{\text{pred}} - a_{\text{gt}}| > \epsilon_2 \times |a_{\text{gt}}|$, the reward is 0 (incorrect). This formulation ensures smooth transitions between full and partial rewards, enabling fair numerical evaluation.

## A.3 Spatial Transformation

**Task Definition** Spatial Transformation is a spatial-visual reasoning task designed to infer single-step or multi-step transformation actions by analyzing the initial and final visual states from multiple perspectives (*e.g.*, center, left, right). The task utilizes transformation functions, including `change_size`, `change_color`, `change_material`, `change_shape`, and `change_position`, to modify object properties such as size, color, material, shape, and position using predefined values. This task evaluates the model's ability to reason about spatial relationships and object transformations across diverse viewpoints in dynamic visual scenarios. Some examples are shown in Fig. 10.

**Dataset Preparation** We generated 100K samples using the environment and configuration from Trance [56], with each sample comprising initial object attributes, front-view image of initial state,

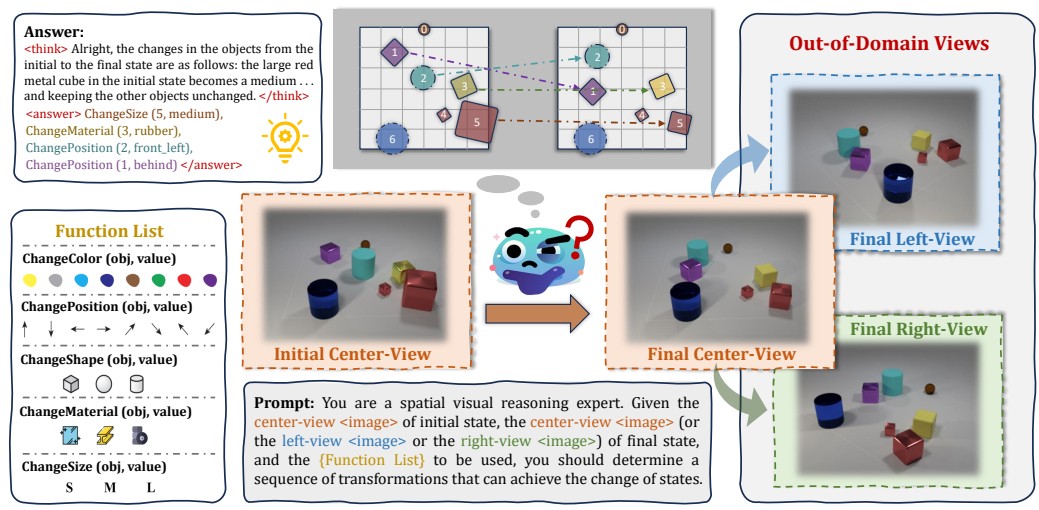

Figure 10: The sample of Spatial Transformation.

and images of final state captured from front, left, and right perspectives. To ensure high data quality, we implemented a rigorous filtering process: (1) removing samples containing occluded or invisible objects in either the initial or final states, (2) eliminating redundant actions within the transformation sequences, and (3) consolidating multi-step displacement actions, which collectively ensure the uniqueness and correctness of the solutions. The refined dataset consists of 60K training samples and 6K test samples. For the training set, we constructed the Trans-Center-60K dataset using the Center-Center configuration, which pairs front-view initial and final state images. For ID evaluation, we derived the Trans-Center-6K dataset from the 6K test samples under the same Center-Center configuration. To evaluate DS generalization, we constructed two additional datasets: Trans-Left-6K (DS-L) and Trans-Right-6K (DS-R), leveraging the Center-Left and Center-Right configurations to assess the model's generalization capabilities in spatial reasoning under viewpoint conditions.

**Reward Design** For the Format Reward, we adopted the same formulation as used in the Visual Counting task. As for the Accuracy Reward, a specialized design was developed to evaluate the sequence of transformation functions. Function-based type is designed for Spatial Transformation tasks requiring a sequence of transformation functions. The accuracy reward $R_{\text{acc}}(a_i)$ evaluates the alignment between the predicted sequence $T_{\text{pred}}$ and the ground truth $T_{\text{gt}}$, computed as:

$$R_{\text{acc}}(a_i) = \frac{\text{len}(T_{\text{pred}}^{f+o+v}) + \alpha \cdot \text{len}(T_{\text{pred}}^{f+o/v}) + \beta \cdot \text{len}(T_{\text{pred}}^{f})}{\max(\text{len}(T_{\text{pred}}), \text{len}(T_{\text{gt}}))}, \tag{9}$$

where $T_{\text{pred}}^{f+o+v}$ is the subset of transformation steps with complete matches (w/ function, object, and value), $T_{\text{pred}}^{f+o/v}$ are the subsets with partial and only-function matches (w/ function and object, or w/ function and value), $T_{\text{pred}}^{f}$ is the subset with only-function matches. $\alpha$ and $\beta$ are the weighting coefficients for partial matches. This formulation ensures nuanced evaluation, rewarding both exact and partially correct responses while allowing flexible adjustment of partial match contributions.

**System Prompts** For the Spatial Transformation task, we designed two versions of the system prompt. The first version specifies the answer output format using the `<think>` and `<answer>` tags, while the second version includes additional outputs `<summary>` and `<caption>` for experiments on exploration of format reward design in the main paper. These two versions are illustrated in Fig. 11 and Fig. 12, respectively.

## B    Details of Models and Training

We utilize Qwen2-VL-2B and Qwen2-VL-7B [87] as the backbone models for our experiments. Our implementation is built on the open-source frameworks Open-R1 [88] and vLLM [89], ensuring reproducibility and scalability. All experiments were conducted on a cluster of servers, each equipped with 8×A800 GPUs. For the Visual Counting task and Spatial Transformation task, we trained

Table 4: **Detailed configuration for each training stage of Reason-RFT.** The table presents the training parameters for the 2B model and 7B model across three visual reasoning tasks. The parameters marked with * correspond to Visual Counting / Structure Perception / Spatial Transformation.

| | | Qwen2-VL-2B | | Qwen2-VL-7B | |
| | | Stage-1 | Stage-2 | Stage-1 | Stage-2 |
|---|---|---|---|---|---|
| Data | **Dataset** | CoT dataset | Non-CoT dataset | CoT dataset | Non-CoT dataset |
| | **#Samples** | 1.6K | 35K / 4.5K / 60K * | 1.6K | 35K / 4.5K / 60K * |
| Model | **Trainable Part** | Full Model | Full Model | Full Model | Full Model |
| | **#Tunable Parameters** | 2.21B | 2.21B | 8.29B | 8.29B |
| Training | **Per-device Batch Size** | 8 | 1 | 8 | 1 |
| | **Gradient Accumulation** | 2 | 2 | 2 | 2 |
| | **LR:** $\{\psi_v^{\mathbf{ViT}}, \phi_v^{\mathbf{LLM}}\}$ | $1\times10^{-5}$ | $1\times10^{-6}$ | $1\times10^{-5}$ | $1\times10^{-6}$ |
| | **Epoch** | 1 | 1 / 5 / 1 * | 1 | 1 / 5 / 1 * |
| | **Optimizer** | AdamW | AdamW | AdamW | AdamW |
| | **Deepspeed** | Zero3 | Zero3 | Zero3 | Zero3 |
| | **Weight Decay** | 0.1 | 0.0 | 0.1 | 0.0 |
| | **Warmup Ratio** | 0.03 | 0.00 | 0.03 | 0.00 |
| | **LR Schedule** | Cosine | Cosine | Cosine | Cosine |
| | **Max Seq. Length** | 32768 | 4096 | 32768 | 4096 |
| | **Max Compl. Length** | – | 512 | – | 512 |
| | **Num. of Compl.** | – | 8 | – | 4 |
| | **GPU Nums** | $1 \times 8$ | $1 \times 8$ | $1 \times 8$ | $1 \times 8$ |

the models for 1 epoch each on their respective training datasets, ensuring sufficient exposure to task-specific patterns while avoiding overfitting. For the Structure Perception task, due to its GeoMath training dataset consisting of a relatively small number of training samples (a total of 4,500), we extended the training duration to 5 epochs to allow the models to fully capture the underlying structural and geometric relationships. In the Reason-RFT training pipeline, all models underwent an initial CoT activation stage with 1,600 samples before proceeding to the RL phase. More details on training process of each models are shown in Tab. 4

## C  More Experiment Results

**Exploration on COT Activation Data** To investigate the impact of differently composed CoT activation data on Reason-RFT, we construct two distinct datasets: a mixed domain-specific dataset, which integrates relevant yet distinct data from in-domain tasks, and a mixed general-domain dataset, encompassing a broader range of visual reasoning tasks (*e.g.*, graph topology, visual puzzles). The

Table 5: Results of various mixed CoT activation datasets on the Structure Perception task.

| Setting | CoT Activation Data | ID | DS | AVG |
|---|---|---|---|---|
| Baseline | GeoMath-only data | 59.27 | 49.25 | 54.26 |
| (a) | Mixed Specific-Domain data | 50.61 | 45.35 | 48.02 |
| (b) | Mixed General-Domain data | 42.51 | 40.25 | 41.38 |

detailed dataset composition is provided in Appendix Sec. E. Using these datasets, we perform Reason-RFT training on Structure Perception task, with the results detailed in Tab. 5. From this, two key points emerge: (1) As the proportion of in-domain training data decreases, the model's performance on specific tasks declines; (2) Models trained on more diverse visual reasoning domain data may also exhibit a reduction in domain-specific performance.

**Results on Different Backbones** We further validate the effectiveness of *Reason-RFT* on stronger or alternative vision–language backbones. We report results on three visual reasoning tasks: Visual Counting (T1), Structure Perception (T2), and Spatial Transformation (T3) in the combined Tab. 6. *Reason-RFT* achieves the strongest averages across backbones and tasks, with especially large margins on domain-shifted splits while keeping in-domain (ID) performance near ceiling. On **Qwen2.5-VL-3B**, for **T1** (Visual Counting) ID is already saturated (99.0 with Reason-RFT-Zero vs. 98.8 with Reason-RFT), yet *Reason-RFT* markedly improves robustness on DS: +9.2 on DS-D (68.7 vs. 59.5 vs. CoT-SFT) and +5.6 on DS-M (54.8 vs. 49.2 vs. CoT-SFT), with an especially large +44.0 over Reason-RFT-Zero on DS-M (54.8 vs. 10.8), yielding the best T1 AVG (74.1); for **T2** (Structure Perception) it is best on both ID/DS (59.0/56.6), beating CoT-SFT by +2.9 (ID) and +7.2 (DS) and Reason-RFT-Zero by +4.2 (ID) and +2.1 (DS), indicating that RL enhances stepwise structural reasoning rather than overfitting; for **T3** (Spatial Transformation) ANS-SFT

Table 6: **Results on different backbones across three tasks.** Best is **bold**; second-best is underlined. "ID" denotes in-domain; "DS-*" denotes domain-shifted splits. Missing results are shown as "–".

| Backbone | Method | Visual Counting (T1) | | | | Structure Perception (T2) | | | Spatial Transformation (T3) | | | |
|---|---|---|---|---|---|---|---|---|---|---|---|---|
| | | ID | DS-D | DS-M | AVG | ID | DS | AVG | ID | DS-L | DS-R | AVG |
| Qwen2.5-VL-3B | Zero-Shot | 75.9 | 50.9 | 4.4 | 43.7 | 36.8 | 37.4 | 37.1 | 8.6 | 8.3 | 8.3 | 8.4 |
| | + ANS-SFT | 97.4 | 51.5 | 6.0 | 51.6 | 53.0 | 31.8 | 42.4 | **91.1** | 47.0 | 46.8 | 61.6 |
| | + CoT-SFT | 89.2 | 59.5 | 49.2 | 66.0 | 56.1 | 49.4 | 52.7 | 81.6 | 46.1 | 44.2 | 57.3 |
| | + Reason-RFT-Zero | 99.0 | 58.9 | 10.8 | 56.2 | 54.8 | 54.5 | 54.6 | 68.5 | 49.5 | 48.0 | 55.3 |
| | + Reason-RFT | 98.8 | **68.7** | **54.8** | **74.1** | **59.0** | **56.6** | **57.8** | 86.7 | **55.2** | **54.4** | **65.4** |
| InternVL3-2B | Zero-Shot | 79.30 | 51.20 | 5.10 | 45.20 | – | – | – | – | – | – | – |
| | + ANS-SFT | 96.80 | 52.00 | 6.50 | 51.77 | – | – | – | – | – | – | – |
| | + CoT-SFT | 88.90 | 60.10 | 50.20 | 66.40 | – | – | – | – | – | – | – |
| | + Reason-RFT-Zero | 98.90 | 59.40 | 12.30 | 56.87 | – | – | – | – | – | – | – |
| | + Reason-RFT | **99.10** | **69.80** | **55.90** | **74.93** | – | – | – | – | – | – | – |

attains the top ID (91.1) but is less robust, whereas *Reason-RFT* trades a modest $-4.4$ on ID (86.7) for substantial DS gains of $+8.2$ on DS-L (55.2 vs. 47.0) and $+7.6$ on DS-R (54.4 vs. 46.8), delivering the best AVG (65.4) and a superior ID/DS Pareto. On **InternVL3-2B** for **T1**, *Reason-RFT* is best on ID/DS-D/DS-M (99.10/69.80/55.90) and AVG (74.93), improving over ANS-SFT, CoT-SFT, and Reason-RFT-Zero by $+23.16$, $+8.53$, and $+18.06$, respectively, with the largest domain-shift margin on DS-M of $+43.60$ over Reason-RFT-Zero (55.90 vs. 12.30). Taken together, these trends across architecturally distinct backbones indicate that the benefits of *Reason-RFT* are backbone-agnostic, improving both coherence-driven reasoning and out-of-distribution reliability.

**Evaluation on General Benchmarks** Although *Reason-RFT* is primarily designed to enhance domain-specific visual reasoning abilities, we conduct a thorough evaluation on general benchmarks to verify whether our approach compromises the model's general reasoning capabilities. Tab. 7 presents the results on four widely adopted datasets: MMMU [97], RealWorldQA [98], MathVision [99], and AI2D [100]. Across all tasks and model scales, *Reason-RFT* consistently maintains or even improves general performance. For instance, on the 2B model, *Reason-RFT* achieves the highest scores on MMMU

Table 7: Evaluation results on general benchmarks.

| Method | General | | | |
|---|---|---|---|---|
| | MMMU | RealWorldQA | MathVision | AI2D |
| **Qwen2VL-2B-Instruct** | | | | |
| Zero-Shot | 39.89 | **61.31** | 12.50 | 72.50 |
| + ANS-SFT | 40.56 | 48.76 | **15.79** | 68.20 |
| + CoT-SFT | 34.00 | 37.78 | 12.99 | 65.36 |
| + Reason-RFT-Zero | 39.30 | 42.81 | 13.00 | 74.61 |
| + Reason-RFT | **41.14** | 53.06 | 14.82 | **75.24** |
| **Qwen2VL-7B-Instruct** | | | | |
| Zero-Shot | **54.10** | **67.19** | 16.30 | **83.00** |
| + ANS-SFT | 42.66 | 48.10 | 9.12 | 78.30 |
| + CoT-SFT | 44.67 | 36.46 | 15.30 | 73.25 |
| + Reason-RFT-Zero | 46.44 | 45.10 | 10.86 | 75.28 |
| + Reason-RFT | 50.04 | 61.31 | **17.60** | 81.70 |

(41.14) and AI2D (75.24), outperforming both zero-shot baselines and other supervised fine-tuning approaches such as ANS-SFT and CoT-SFT. Notably, it also improves performance on the challenging MathVision task (14.82), demonstrating its robustness in spatial reasoning. For the larger 7B model, *Reason-RFT* again surpasses ANS-SFT and CoT-SFT by large margins, particularly on RealWorldQA (61.31) and MMMU (50.04), while maintaining strong results on AI2D (81.70). These results suggest that *Reason-RFT* not only scales effectively with model size but also introduces no observable performance degradation on general benchmarks. In summary, the empirical evidence supports that *Reason-RFT* enhances domain-specific reasoning while preserving—if not enhancing—general visual-language reasoning capabilities. This confirms the robustness and transferability of our method, making it a strong alternative to conventional fine-tuning paradigms.

**Performance at Different Training Steps** Fig. 13 and Fig. 14 illustrate the ID and DS performance of all the training methods across three visual reasoning tasks, evaluated at various training sample sizes. This analysis helps us understand how each method scales with training data. More detail evaluation results for each subset of three tasks are in Tab. 10 - Tab. 17. We systematically varied the number of training samples, from minimal to substantial, allowing us to identify performance thresholds and data efficiency for each method in both ID and DS contexts. Key findings from this analysis include: Data Efficiency of Reason-RFT: *Reason-RFT* demonstrates exceptional data efficiency, achieving approximately 70% of the performance of *Reason-RFT-Zero* with only 3% of the training data (1,600 samples), and 82.5% with just 9%. Robust Generalization to DS scenarios: In the 7B model, *Reason-RFT* achieves over 92% of *Reason-RFT-Zero*'s performance using just 3% of the training data, showcasing its strong generalization capabilities. Comparison Across Methods: *Reason-RFT* consistently outperforms other methods, particularly in data-constrained scenarios,

indicating its suitability for applications with limited data availability. Performance Saturation: As training sample size increases, some methods experience performance plateaus, suggesting that beyond a certain point, additional data yields diminishing returns.

In conclusion, the evaluation of performance across different training samples not only highlights the strengths of *Reason-RFT* in terms of data efficiency and generalization but also provides critical insights into the performance dynamics of various methods. These findings are essential for practitioners aiming to maximize performance while effectively managing training resources.

# D More Details on CoT Data Construction

This section expands the pipeline of CoT generation by detailing both the automated and manual components used to construct our *CoT-SFT* corpus.

**(1) Automated Generation.** We instantiate CoT drafts using reasoning-guided prompt templates such as *"Let's break down the problem step by step. . . "* and *"To answer this, we need to consider. . . "*. Templates are combined with model prompting (GPT-4o [95] and Gemini-Pro [96]) under temperature-controlled sampling ($T=0.7$, top-$k=50$, top-$p=0.9$). To increase coverage and depth, we insert hand-crafted, subtask-specific few-shot exemplars that bias toward explicit intermediate justifications and error-checking behavior.

**(2) Automated Filtering.** Each generated CoT is screened by two criteria:

*Length range.* For each subtask $s$, we compute a target trajectory length $\bar{L}_s$ from a balanced mixture of 50% human-written and 50% model-generated samples:

$$\bar{L}_s \;=\; \tfrac{1}{2}\big(\bar{L}_s^{\text{human}} + \bar{L}_s^{\text{model}}\big),$$

where $\bar{L}_s^{\text{human}}$ and $\bar{L}_s^{\text{model}}$ are computed as sample means over their respective sets after basic de-duplication. A candidate with length $L_i$ is retained iff

$$0.6\,\bar{L}_s \;\leq\; L_i \;\leq\; 1.4\,\bar{L}_s,$$

where $L_i$ is measured in tokens by our training-time tokenizer; the factors $0.6$ and $1.4$ were selected via a small pilot study to trim outliers while preserving diversity.

*Inconsistency.* We discard the responses that contradict the known ground truth, including self-inconsistent counts, incompatible algebraic steps, or reasoning that invalidates later conclusions.

For reference, the empirical trajectory-length statistics (mean $\mu$, stdev $\sigma$) across tasks are:

| Task | $\mu$ (tokens) | $\sigma$ (tokens) |
|---|---|---|
| Visual Counting | 70 | 30 |
| Structural Perception | 180 | 80 |
| Spatial Transformation | 400 | 120 |

where $\mu$ and $\sigma$ are computed over the curated pool *after* automated filtering and *before* manual review.

**(3) Human Verification.** We randomly sample $10\%$ of CoT drafts from eachtask for manual review, focusing on (i) step-to-step coherence, (ii) logical validity, and (iii) alignment between the reasoning chain and the final answer. Typical failure modes include: (i) a correct final answer supported by an incorrect chain (e.g., deriving triangle area via the Pythagorean theorem); (ii) internal contradictions, such as stating "There are 3 red blocks on the left and 2 on the right" and later concluding the total is 6. A follow-up quality audit found that, prior to human verification, approximately $3.8\%$ of samples contained critical logical flaws; after verification, the residual error rate fell below $1\%$, indicating high post-cleanup reliability.

**Discussion.** Automated generation with calibrated sampling and subtask-specific few-shots provides diverse yet structured CoTs; the length- and consistency-based filters remove overly terse/verbose or self-contradictory drafts; targeted human verification further suppresses high-severity errors. Together, these stages yield a CoT–SFT dataset with improved coherence and faithfulness, while maintaining scalability and reproducibility.

# E   Detail on Mixed CoT Datasets

As shown in Tab. 8, we presents a comprehensive overview of the datasets utilized for all of our visual reasoning experiments, categorized into three experimental groups. All of them are CoT-annotated by GPT-4o [95] The Main Experiment section includes three large-scale datasets: Visual-Counting (35,000 samples) for quantitative analysis, Structure-Perception (4,500 samples) for structural understanding, and Spatial-Transformation (60,000 samples) assessing spatial reasoning capabilities. For Ablation Studies, two mixed-domain subsets were constructed: (1) The Mixed General-Domain set comprises 11 CoT-annotated datasets spanning scientific reasoning (AI2D [59], ScienceQA [101]), topological graph problems (GVLQA series [102]), and pattern recognition (PuzzleVQA [103], IconQA [104], Raven [105]). (2) The Mixed Specific-Domain set focuses exclusively on geometric reasoning, featuring GeoQA [106], GeomVerse [53], and Geometry3K [85] with progressively complex problem structures. All datasets were standardized to ensure training compatibility.

Table 8: Datasets Overview for Visual Reasoning Tasks

| Dataset Name | Samples | Reasoning Type | Description |
|---|---|---|---|
| **Main Experiment** | | | |
| Visual-Counting | 35,000 | Visual Counting | Full dataset for visual counting task |
| Structure-Perception | 4,500 | Structure Perception | Full dataset for structural perception tasks |
| Spatial-Transformation | 60,000 | Spatial Transformation | Full dataset for spatial transformation tasks |
| **Ablation Experiment (Mixed General-Domain)** | | | |
| AI2D [59] | 1,467 | Scientific Reasoning | Scientific diagram interpretation |
| ScienceQA [101] | 2,112 | Scientific Reasoning | Science question answering |
| GVLQA-connectivity [102] | 1,199 | Topological Reasoning | Graph connectivity problems |
| GVLQA-cycle [102] | 1,194 | Topological Reasoning | Cycle detection in graphs |
| GVLQA-hamilton [102] | 1,158 | Topological Reasoning | Hamiltonian path problems |
| GVLQA-topology [102] | 1,070 | Topological Reasoning | General topology questions |
| GVLQA-matching [102] | 1,193 | Topological Reasoning | Graph matching tasks |
| PuzzleVQA [103] | 1,618 | Pattern/Puzzle | Visual puzzle solving |
| IconQA [104] | 5,270 | Pattern/Puzzle | Icon-based question answering |
| Raven [105] | 982 | Pattern/Puzzle | Raven's Progressive Matrices |
| **Ablation Experiment (Mixed Specific-Domain)** | | | |
| GeoQA [106] | 1,500 | Geometric Reasoning | Geometric problem solving |
| GeomVerse [53] | 2,841 | Geometric Reasoning | Advanced geometry challenges |
| Geometry3K [85] | 3,794 | Geometric Reasoning | Comprehensive geometry problems |

# F   Comparison of CoT Quality Before and After RL

**Setting.** We compare the **Qwen2VL-3B** model trained with *Reason-RFT* (Stage 2, post-RL) against the same backbone trained with only *CoT-SFT* (Stage 1, pre-RL) on the Structure Perception task. Unless otherwise noted, statistics are computed over a random sample of $n = 100$ problem instances.

**Qualitative findings.** Despite the high textual similarity between the two variants, the post-RL model exhibits stronger logical coherence across intermediate steps, with fewer broken or skipped chains of inference. For example, in Fig. 16 (case 2), the pre-RL model correctly infers a formula but omits the subsequent multiplication by 2, an error that is notably less frequent after RL. In addition, the post-RL model more often displays reflective behaviors (*e.g., "let me double check"*) that are rarely observed in pre-RL outputs as shown in the math example of Fig. 1.

Table 9: Summary of comparative metrics on *Structure Perception* ($n = 100$). Positive values indicate post-RL improvements.

| Metric | Change (Post–Pre) |
|---|---|
| Reasoning Step Count | +2.7 steps |
| Prompting Words | +14% |
| Logical Connectives | +23% |
| Answer Accuracy | +20.56% |

**Quantitative protocol.** We assess three dimensions of chain-of-thought (CoT) quality: (i) ***Reasoning Step Count***—the number of explicitly delimited reasoning steps per sample, obtained via automatic counting with GPT-4o [95]; (ii) ***Lexical Usage*** of two categories of expressions: *Prompting words*

(*e.g., "oh I see"*, *"let me think step by step"*, *"let me double check"*) and *Logical connectives* (*e.g., "so"*, *"therefore"*, *"first"*, *"but"*, *"moreover"*); and (iii) ***Answer Accuracy*** as reported for the accuracy rate of Structure Perception task.

**Results and interpretation.** Post-RL training increases the average CoT *granularity* (as reflected by the larger step count), the *organizational scaffolding* of reasoning (higher usage of prompting phrases and discourse connectives), and the *task effectiveness* (higher final-answer accuracy). Taken together, these observations indicate that reinforcement learning with Reason-RFT enhances both the coherence and utility of CoT: it reduces fragile or truncated chains, encourages reflective self-checks, and translates these behaviors into substantial accuracy gains.

## G Visualization

In this section, we present additional visualization results on general visual reasoning and three specific task reasoning, see Fig. 15 - Fig. 23. Reason-RFT demonstrates superior performance over CoT-SFT in terms of logical consistency, reasoning quality, and correctness. CoT-SFT's flaws stem from incorrect assumptions and misinterpretations, highlighting the importance of accurate problem interpretation and reasoning in visual reasoning tasks.

## H Limitations and Societal Impact

**Limitations** While Reason-RFT has demonstrated strong performance in visual reasoning tasks, there are still areas to address. Future work will explore its application across a range of computer vision models, scaling to larger architectures (e.g., 32B/72B), and integrating large-scale Mixture of Experts (MoE) models to evaluate generalization. We will also extend the framework to complex downstream scenarios, such as embodied AI and autonomous driving, testing its effectiveness in real-world applications that require sophisticated visual reasoning and real-time decision-making.

**Societal Impact** The advancements of Reason-RFT in visual reasoning have important societal implications. By enhancing generalization and cross-domain transferability, this framework can improve AI applications in areas like medical imaging, autonomous driving, and assistive technologies for the visually impaired. It also reduces overfitting and cognitive rigidity, leading to more reliable and interpretable AI systems that foster trust in human-AI collaboration. The reconstructed benchmark dataset allows for fair evaluation, promoting research in robust AI. However, ethical considerations, such as biases in training data and responsible deployment, must be addressed to prevent misuse. Overall, Reason-RFT paves the way for adaptable and trustworthy AI, benefiting industries, researchers, and society.

## System Prompt for Spatial Transformation Task

'''Your need to complete the spatial visual reasoning task according to the following rules.

### Task Description:

Given the image of the initial state, the image of the final state, and the attributes of the initial objects, you should determine a transformation that can achieve the change of states.

The **attributes of the initial objects** are provided as a list of tuples in the following format:
**('object_id', 'shape', 'size', 'color', 'material')**
Each tuple represents an object and its properties in the initial state.

The transformation should be a sequence of functions with a length ranging from 1 to 4, where each function is represented as **'func(object_id, value)'**.

### Available functions and values:

1. **'change_size(object_id, value)'** - Changes the object to a new size relative to its initial size.
   - Possible values: `['small', 'medium', 'large']`

2. **'change_color(object_id, value)'** - Changes the object to a new color relative to its initial color.
   - Possible values: `['yellow', 'gray', 'cyan', 'blue', 'brown', 'green', 'red', 'purple']`

3. **'change_material(object_id, value)'** - Changes the object to a new material relative to its initial material.
   - Possible values: `['glass', 'metal', 'rubber']`

4. **'change_shape(object_id, value)'** - - Changes the object to a new shape relative to its initial shape.
   - Possible values: `['cube', 'sphere', 'cylinder']`

5. **'change_position(object_id, value)'** - Moves the object to a new position relative to its initial location.
   - Possible values: `['front', 'behind', 'left', 'right', 'front_left', 'front_right', 'behind_left', 'behind_right']`
   - 'front' means moving forward along the object's initial direction.
   - 'behind' means moving backward along the object's initial direction.
   - 'left' means moving to the left of the object's initial orientation.
   - 'right' means moving to the right of the object's initial orientation.
   - 'front_left' means moving diagonally toward the front and left of the initial location.
   - 'front_right' means moving diagonally toward the front and right of the initial location.
   - 'behind_left' means moving diagonally toward the behind and left of the initial location.
   - 'behind_right' means moving diagonally toward the behind and right of the initial location.

### Output Format

You should first thinks about the reasoning process internally and then provides the user with the answer. The **reasoning process** and **answer** are enclosed within specific tags:

- **Reasoning process**: Provide a chain-of-thought, logical explanation of the problem. This should outline step-by-step reasoning, enclosed within `<think>...</think>`

- **Final answer (sequence of functions only)**: Enclosed within `<answer>...</answer>`

Now, it's your turn!

{Question} Output the thinking process in <think> </think> and final answer in <answer> </answer> tags.
'''

Figure 11: The system prompt used in Spatial Transformation task.

**System Prompt for Spatial Transformation Task (Add <summary> <caption> in FORMAT)**

"'Your need to complete the spatial visual reasoning task according to the following rules.

### Task Description:

Given the image of the initial state, the image of the final state, and the attributes of the initial objects, you should determine a transformation that can achieve the change of states.

The **attributes of the initial objects** are provided as a list of tuples in the following format:
**('object_id', 'shape', 'size', 'color', 'material')**
Each tuple represents an object and its properties in the initial state.

The transformation should be a sequence of functions with a length ranging from 1 to 4, where each function is represented as **'func(object_id, value)'**.

### Available functions and values:

1. **'change_size(object_id, value)'** - Changes the object to a new size relative to its initial size.
   - Possible values: `['small', 'medium', 'large']`

2. **'change_color(object_id, value)'** - Changes the object to a new color relative to its initial color.
   - Possible values: `['yellow', 'gray', 'cyan', 'blue', 'brown', 'green', 'red', 'purple']`

3. **'change_material(object_id, value)'** - Changes the object to a new material relative to its initial material.
   - Possible values: `['glass', 'metal', 'rubber']`

4. **'change_shape(object_id, value)'** - - Changes the object to a new shape relative to its initial shape.
   - Possible values: `['cube', 'sphere', 'cylinder']`

5. **'change_position(object_id, value)'** - Moves the object to a new position relative to its initial location.
   - Possible values: `['front', 'behind', 'left', 'right', 'front_left', 'front_right', 'behind_left', 'behind_right']`
   - 'front' means moving forward along the object's initial direction.
   - 'behind' means moving backward along the object's initial direction.
   - 'left' means moving to the left of the object's initial orientation.
   - 'right' means moving to the right of the object's initial orientation.
   - 'front_left' means moving diagonally toward the front and left of the initial location.
   - 'front_right' means moving diagonally toward the front and right of the initial location.
   - 'behind_left' means moving diagonally toward the behind and left of the initial location.
   - 'behind_right' means moving diagonally toward the behind and right of the initial location.

### Output Format

You should first thinks about the reasoning process internally and then provides the user with the answer. The **reasoning process** and **answer** are enclosed within specific tags:

- **Summary process**: Summary how you will approach the problem and explain the steps you will take to reach the answer, enclosed within `<summary>...</summary>`

- **Caption process**: Provide a detailed description of the image, particularly emphasizing the aspects related to the question, enclosed within `<caption>...</caption>`

- **Reasoning process**: Provide a chain-of-thought, logical explanation of the problem. This should outline step-by-step reasoning, enclosed within `<think>...</think>`

- **Final answer (sequence of functions only)**: Enclosed within `<answer>...</answer>`

Now, it's your turn!

{Question} Output the summary process in <summary> </summary>, caption process in <caption>...</caption>, thinking process in <think> </think> and final answer in <answer> </answer> tags.
"'

Figure 12: The system prompt used in Spatial Transformation task w/ `<summary>` and `<caption>` tags in format.

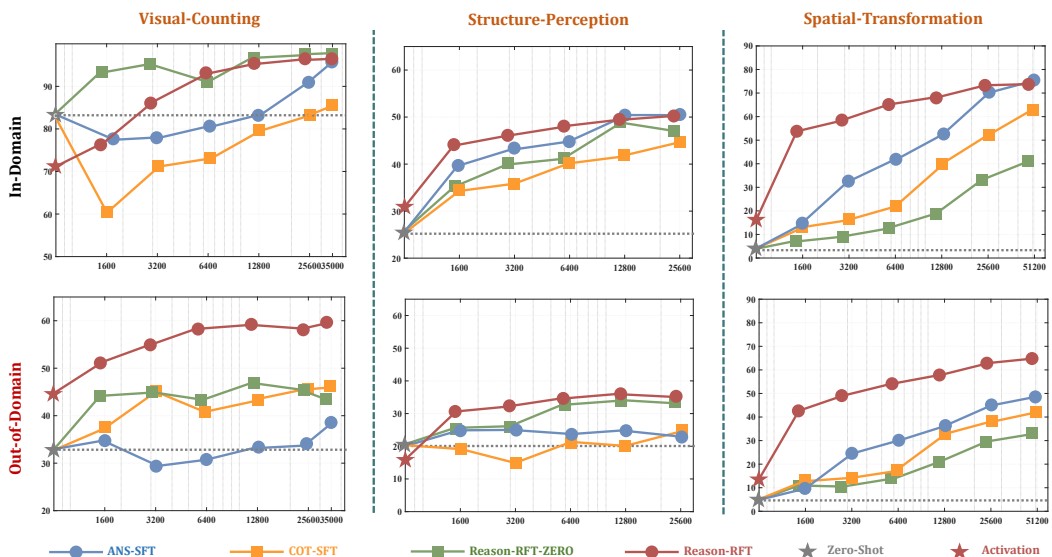

Figure 13: Results of all methods on Qwen2VL-2B-Instruct, ID and DS performance at different training checkpoints.

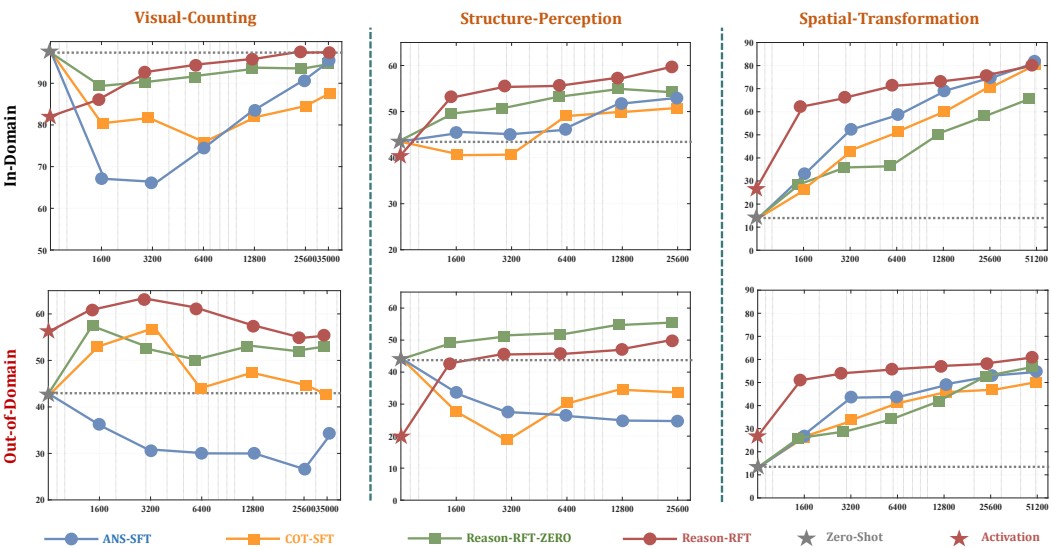

Figure 14: Results of all methods on Qwen2VL-7B-Instruct, ID and DS performance at different training checkpoints.

| Methods | Steps | Visual Counting | | | | |
|---|---|---|---|---|---|---|
| | | Clevr-Math (ID) | | | | |
| | | adversarial | sub-multi | addition | subtraction | AVG |
| **Zero-Shot** | - | 93.60 | 84.00 | 55.60 | 96.40 | 82.40 |
| ANS-SFT | 100 | 83.60 | 56.40 | 91.20 | 81.60 | 78.20 |
| | 200 | 69.20 | 67.60 | 91.60 | 82.00 | 77.60 |
| | 400 | 81.60 | 65.60 | 90.80 | 84.80 | 80.70 |
| | 600 | 72.40 | 73.20 | 92.40 | 89.20 | 81.80 |
| | 800 | 78.40 | 77.20 | 82.80 | 90.40 | 82.20 |
| | 1200 | 85.60 | 78.00 | 91.60 | 95.60 | 87.70 |
| | 1600 | 92.80 | 82.40 | 94.80 | 96.80 | 91.70 |
| | 2187 | 95.20 | 92.80 | 97.60 | 99.20 | 96.20 |
| CoT-SFT | 100 | 49.20 | 40.00 | 82.00 | 69.20 | 60.10 |
| | 200 | 65.20 | 55.60 | 88.00 | 76.40 | 71.30 |
| | 400 | 66.00 | 57.20 | 90.00 | 79.60 | 73.20 |
| | 600 | 67.20 | 59.20 | 87.20 | 82.80 | 74.10 |
| | 800 | 77.60 | 61.60 | 92.40 | 85.20 | 79.20 |
| | 1200 | 76.80 | 70.00 | 91.20 | 93.60 | 82.90 |
| | 1600 | 80.80 | 66.80 | 91.60 | 92.00 | 82.80 |
| | 2187 | 83.20 | 71.20 | 93.20 | 94.40 | 85.50 |
| Reason-RFT-Zero | 100 | 92.80 | 88.80 | 94.40 | 96.00 | 93.00 |
| | 200 | 95.60 | 91.60 | 95.60 | 97.60 | 95.10 |
| | 400 | 92.00 | 87.60 | 84.00 | 96.40 | 90.00 |
| | 600 | 94.40 | 92.80 | 93.60 | 96.00 | 94.20 |
| | 800 | 96.40 | 96.40 | 96.00 | 98.80 | 96.90 |
| | 1200 | 98.40 | 95.60 | 100.00 | 99.60 | 98.40 |
| | 1600 | 96.40 | 94.80 | 98.80 | 99.60 | 97.40 |
| | 2500 | 98.40 | 95.60 | 99.60 | 100.00 | 98.40 |
| Reason-RFT | 100 | 89.60 | 73.20 | 93.60 | 95.60 | 88.00 |
| | 200 | 89.20 | 78.00 | 95.20 | 96.40 | 89.70 |
| | 400 | 92.80 | 82.40 | 95.20 | 97.60 | 92.00 |
| | 600 | 94.80 | 86.00 | 96.80 | 97.20 | 93.70 |
| | 800 | 96.80 | 88.40 | 96.80 | 98.80 | 95.20 |
| | 1200 | 94.80 | 86.00 | 96.40 | 98.80 | 94.00 |
| | 1600 | 94.40 | 91.60 | 97.20 | 99.60 | 95.70 |
| | 2500 | 98.40 | 92.80 | 96.80 | 99.20 | 96.80 |

Table 10: Complete experimental results of Qwen2VL-2B-Instruct on the Clevr-Math test set after training on Clevr-Math. "sub-multi" donates the subtraction-multihop task.

| Methods | Steps | Visual Counting | | | | | | |
|---|---|---|---|---|---|---|---|---|
| | | Super-Clevr-Math (DS) | | | | | | |
| | | addition | subtraction | add-sub | sub-multi | AVG | DS-D | DS-M |
| **Zero-Shot** | - | 10.40 | 54.40 | 0.00 | 63.20 | 32.00 | 42.67 | 0.00 |
| ANS-SFT | 100 | 51.20 | 37.60 | 11.60 | 39.20 | 34.90 | 42.67 | 11.60 |
| | 200 | 38.40 | 55.60 | 8.40 | 15.60 | 29.50 | 36.53 | 8.40 |
| | 400 | 40.80 | 45.20 | 5.60 | 35.20 | 31.70 | 40.40 | 5.60 |
| | 600 | 41.20 | 61.60 | 8.00 | 35.60 | 36.60 | 46.13 | 8.00 |
| | 800 | 49.20 | 50.40 | 7.20 | 26.00 | 33.20 | 41.87 | 7.20 |
| | 1200 | 44.00 | 53.20 | 5.60 | 38.80 | 35.40 | 45.33 | 5.60 |
| | 1600 | 48.80 | 53.60 | 6.00 | 26.00 | 33.60 | 42.80 | 6.00 |
| | 2187 | 49.60 | 62.00 | 5.20 | 41.60 | 39.60 | 51.07 | 5.20 |
| CoT-SFT | 100 | 47.20 | 50.00 | 28.80 | 25.60 | 37.90 | 40.93 | 28.80 |
| | 200 | 56.00 | 52.40 | 38.00 | 34.00 | 45.10 | 47.47 | 38.00 |
| | 400 | 55.20 | 57.20 | 22.40 | 30.40 | 41.30 | 47.60 | 22.40 |
| | 600 | 58.40 | 55.20 | 24.00 | 35.60 | 43.30 | 49.73 | 24.00 |
| | 800 | 57.60 | 47.60 | 26.80 | 41.60 | 43.40 | 48.93 | 26.80 |
| | 1200 | 58.00 | 54.40 | 35.60 | 32.40 | 45.10 | 48.27 | 35.60 |
| | 1600 | 53.20 | 58.40 | 33.20 | 40.40 | 46.30 | 50.67 | 33.20 |
| | 2187 | 53.60 | 58.80 | 36.80 | 36.80 | 46.50 | 49.73 | 36.80 |
| Reason-RFT-Zero | 100 | 46.00 | 65.20 | 6.80 | 58.80 | 44.20 | 56.67 | 6.80 |
| | 200 | 48.80 | 66.00 | 9.20 | 57.60 | 45.40 | 57.47 | 9.20 |
| | 400 | 42.00 | 71.20 | 8.40 | 50.80 | 43.10 | 54.67 | 8.40 |
| | 600 | 47.20 | 65.20 | 7.60 | 47.60 | 41.90 | 53.33 | 7.60 |
| | 800 | 56.40 | 69.20 | 6.80 | 55.20 | 46.90 | 60.27 | 6.80 |
| | 1200 | 52.00 | 73.60 | 7.20 | 59.20 | 48.00 | 61.60 | 7.20 |
| | 1600 | 51.60 | 71.60 | 6.40 | 54.80 | 46.10 | 59.33 | 6.40 |
| | 2500 | 49.60 | 71.20 | 5.20 | 53.20 | 44.80 | 58.00 | 5.20 |
| Reason-RFT | 100 | 59.20 | 57.60 | 38.00 | 41.60 | 49.10 | 52.80 | 38.00 |
| | 200 | 59.60 | 64.40 | 39.20 | 42.00 | 51.30 | 55.33 | 39.20 |
| | 400 | 61.60 | 64.00 | 39.20 | 37.20 | 50.50 | 54.27 | 39.20 |
| | 600 | 66.80 | 67.20 | 32.00 | 46.00 | 53.00 | 60.00 | 32.00 |
| | 800 | 66.00 | 65.60 | 34.00 | 39.20 | 51.20 | 56.93 | 34.00 |
| | 1200 | 67.20 | 65.20 | 33.60 | 40.80 | 51.70 | 57.73 | 33.60 |
| | 1600 | 63.60 | 66.00 | 33.20 | 44.80 | 51.90 | 58.13 | 33.20 |
| | 2500 | 68.00 | 67.20 | 28.40 | 44.80 | 52.10 | 60.00 | 28.40 |

Table 11: Complete experimental results of Qwen2VL-2B-Instruct on the Super-Clevr-Math test set after training on Clevr-Math. "add-sub" donates the addition-subtraction task, while "sub-multi" donates the subtraction-multihop task. "Direct Arithmetic"(DS-D) refers to the types of questions the model has previously seen during Clevr-Math training, while "Mixed Arithmetic"(DS-M) denotes the complicated types that the model has not encountered (*i.e.* questions with multi-step mixture of addition and subtraction).

| Methods | Steps | Visual Counting | | | | |
| | | Clevr-Math (ID) | | | | |
| | | adversarial | sub-multi | addition | subtraction | AVG |
|---|---|---|---|---|---|---|
| **Zero-Shot** | - | 99.60 | 98.40 | 97.60 | 98.80 | 98.60 |
| ANS-SFT | 100 | 69.20 | 54.00 | 81.20 | 69.20 | 68.40 |
| | 200 | 61.20 | 50.00 | 82.40 | 75.60 | 67.30 |
| | 400 | 69.20 | 63.60 | 89.20 | 77.60 | 74.90 |
| | 600 | 70.40 | 54.00 | 90.40 | 81.20 | 74.00 |
| | 800 | 80.00 | 74.00 | 91.20 | 89.20 | 83.60 |
| | 1200 | 86.80 | 79.20 | 94.40 | 91.20 | 87.90 |
| | 1600 | 90.40 | 84.40 | 95.20 | 92.00 | 90.50 |
| | 2187 | 96.80 | 89.20 | 96.80 | 97.20 | 95.00 |
| CoT-SFT | 100 | 81.60 | 63.60 | 91.20 | 83.60 | 80.00 |
| | 200 | 80.00 | 64.00 | 92.00 | 88.80 | 81.20 |
| | 400 | 72.40 | 66.00 | 88.80 | 79.60 | 76.70 |
| | 600 | 77.60 | 66.00 | 94.40 | 89.20 | 81.80 |
| | 800 | 78.40 | 65.20 | 94.00 | 87.20 | 81.20 |
| | 1200 | 79.60 | 76.80 | 92.40 | 88.00 | 84.20 |
| | 1600 | 86.40 | 78.00 | 92.80 | 93.20 | 87.60 |
| | 2187 | 87.20 | 78.80 | 93.60 | 89.60 | 87.30 |
| Reason-RFT-Zero | 100 | 98.00 | 94.40 | 98.80 | 99.60 | 97.70 |
| | 200 | 99.60 | 93.20 | 99.20 | 100.00 | 98.00 |
| | 400 | 99.60 | 95.20 | 99.60 | 98.80 | 98.30 |
| | 600 | 98.00 | 98.40 | 100.00 | 99.60 | 99.00 |
| | 800 | 99.60 | 98.40 | 99.60 | 98.80 | 99.10 |
| | 1200 | 100.00 | 98.00 | 99.60 | 99.20 | 99.20 |
| | 1600 | 99.60 | 97.60 | 100.00 | 99.20 | 99.10 |
| | 2500 | 99.60 | 98.40 | 100.00 | 99.60 | 99.40 |
| Reason-RFT | 100 | 88.80 | 79.20 | 95.60 | 94.40 | 89.50 |
| | 200 | 92.00 | 80.00 | 96.40 | 95.20 | 90.90 |
| | 400 | 94.40 | 84.40 | 96.00 | 95.60 | 92.60 |
| | 600 | 92.80 | 84.00 | 96.40 | 97.60 | 92.70 |
| | 800 | 92.80 | 85.20 | 96.80 | 96.40 | 92.80 |
| | 1200 | 94.80 | 89.60 | 97.20 | 97.60 | 94.80 |
| | 1600 | 94.80 | 86.40 | 97.60 | 97.20 | 94.00 |
| | 2500 | 96.80 | 88.40 | 99.20 | 98.00 | 95.60 |

Table 12: Complete experimental results of Qwen2VL-7B-Instruct on the Clevr-Math test set after training on Clevr-Math. "sub-multi" donates the subtraction-multihop task.

| Methods | Steps | Visual Counting | | | | | | |
|---|---|---|---|---|---|---|---|---|
| | | Super-Clevr-Math (DS) | | | | | | |
| | | addition | subtraction | add-sub | sub-multi | AVG | DS-D | DS-M |
| **Zero-Shot** | - | 46.80 | 75.20 | 4.80 | 41.60 | 42.10 | 54.53 | 4.80 |
| ANS-SFT | 100 | 57.60 | 41.20 | 5.60 | 46.40 | 37.70 | 48.40 | 5.60 |
| | 200 | 42.00 | 38.80 | 8.00 | 33.60 | 30.60 | 38.13 | 8.00 |
| | 400 | 37.20 | 46.40 | 5.20 | 31.60 | 30.10 | 38.40 | 5.20 |
| | 600 | 32.00 | 44.80 | 12.40 | 19.20 | 27.10 | 32.00 | 12.40 |
| | 800 | 38.80 | 38.00 | 6.80 | 37.20 | 30.20 | 38.00 | 6.80 |
| | 1200 | 42.00 | 42.80 | 12.80 | 32.00 | 32.40 | 38.93 | 12.80 |
| | 1600 | 36.40 | 48.40 | 11.20 | 17.20 | 28.30 | 34.00 | 11.20 |
| | 2187 | 39.60 | 58.80 | 8.00 | 29.20 | 33.90 | 42.53 | 8.00 |
| CoT-SFT | 100 | 60.00 | 63.60 | 44.00 | 41.60 | 52.30 | 55.07 | 44.00 |
| | 200 | 67.60 | 66.40 | 48.00 | 46.80 | 57.20 | 60.27 | 48.00 |
| | 400 | 55.20 | 60.40 | 19.60 | 42.00 | 44.30 | 52.53 | 19.60 |
| | 600 | 64.80 | 61.20 | 35.20 | 43.20 | 51.10 | 56.40 | 35.20 |
| | 800 | 60.00 | 53.60 | 37.60 | 42.40 | 48.40 | 52.00 | 37.60 |
| | 1200 | 51.20 | 56.00 | 35.20 | 39.60 | 45.50 | 48.93 | 35.20 |
| | 1600 | 53.20 | 56.40 | 34.40 | 35.20 | 44.80 | 48.27 | 34.40 |
| | 2187 | 51.60 | 51.60 | 33.60 | 32.80 | 42.40 | 45.33 | 33.60 |
| Reason-RFT-Zero | 100 | 58.80 | 82.80 | 24.00 | 62.40 | 57.00 | 68.00 | 24.00 |
| | 200 | 56.00 | 83.20 | 18.80 | 50.00 | 52.00 | 63.07 | 18.80 |
| | 400 | 62.40 | 79.60 | 22.80 | 37.60 | 50.60 | 59.87 | 22.80 |
| | 600 | 61.20 | 85.20 | 17.20 | 49.20 | 53.20 | 65.20 | 17.20 |
| | 800 | 52.80 | 86.80 | 20.40 | 52.00 | 53.00 | 63.87 | 20.40 |
| | 1200 | 53.60 | 83.20 | 19.20 | 46.80 | 50.70 | 61.20 | 19.20 |
| | 1600 | 61.20 | 84.80 | 18.40 | 43.20 | 51.90 | 63.07 | 18.40 |
| | 2500 | 59.20 | 86.40 | 21.20 | 45.20 | 53.00 | 63.60 | 21.20 |
| Reason-RFT | 100 | 53.60 | 56.80 | 33.20 | 39.60 | 45.80 | 50.00 | 33.20 |
| | 200 | 52.00 | 61.20 | 31.60 | 44.00 | 47.20 | 52.40 | 31.60 |
| | 400 | 56.00 | 59.60 | 30.80 | 45.20 | 47.90 | 53.60 | 30.80 |
| | 600 | 56.00 | 64.00 | 31.60 | 50.00 | 50.40 | 56.67 | 31.60 |
| | 800 | 56.00 | 60.00 | 28.00 | 41.60 | 46.40 | 52.53 | 28.00 |
| | 1200 | 66.00 | 65.60 | 38.00 | 50.40 | 55.00 | 60.67 | 38.00 |
| | 1600 | 64.40 | 59.60 | 32.40 | 48.80 | 51.30 | 57.60 | 32.40 |
| | 2500 | 62.80 | 60.80 | 35.60 | 44.80 | 51.00 | 56.13 | 35.60 |

Table 13: Complete experimental results of Qwen2VL-7B-Instruct on the Super-Clevr test set after training on Clevr-Math. "add-sub" donates the addition-subtraction task, while "sub-multi" donates the subtraction-multihop task. "Direct Arithmetic"(DS-D) refers to the types of questions the model has previously seen during Clevr-Math training, while "Mixed Arithmetic"(DS-M) denotes the complicated types that the model has not encountered (*i.e.* questions with multi-step mixture of addition and subtraction).

| Methods | Steps | Structure Perception | | | | | |
|---|---|---|---|---|---|---|---|
| | | Geometry3k (DS) | | | GeoMath (ID) | | |
| | | CHOICE | NON-CHOICE | AVG | CHOICE | NON-CHOICE | AVG |
| **Zero-Shot** | - | 40.25 | 1.00 | 20.63 | 35.57 | 20.31 | 25.86 |
| ANS-SFT | 100 | 35.25 | 16.25 | 25.75 | 58.72 | 29.89 | 40.37 |
| | 200 | 33.25 | 17.50 | 25.38 | 56.38 | 35.44 | 43.05 |
| | 400 | 30.75 | 17.00 | 23.88 | 64.77 | 35.06 | 45.86 |
| | 600 | - | - | - | 73.83 | 38.12 | 51.10 |
| | 800 | 32.75 | 16.00 | 24.38 | 72.15 | 36.40 | 49.39 |
| | 1200 | - | - | - | 73.83 | 35.44 | 49.39 |
| | 1600 | 29.00 | 16.00 | 22.50 | 74.83 | 37.36 | 50.98 |
| | 1686 | 28.75 | 16.25 | 22.50 | 74.83 | 37.93 | 51.34 |
| CoT-SFT | 100 | 16.50 | 21.50 | 19.00 | 31.54 | 34.10 | 33.17 |
| | 200 | 7.50 | 23.50 | 15.50 | 32.89 | 35.25 | 34.39 |
| | 400 | 21.50 | 21.25 | 21.38 | 41.61 | 40.04 | 40.61 |
| | 600 | - | - | - | 43.62 | 36.59 | 39.14 |
| | 800 | 16.50 | 23.50 | 20.00 | 45.97 | 39.27 | 41.70 |
| | 1200 | - | - | - | 53.02 | 40.04 | 44.76 |
| | 1600 | 24.25 | 24.00 | 24.13 | 53.69 | 37.93 | 43.66 |
| | 1686 | 26.75 | 23.75 | 25.25 | 51.34 | 38.31 | 43.05 |
| Reason-RFT-Zero | 100 | 32.25 | 17.75 | 25.00 | 41.61 | 31.23 | 35.00 |
| | 200 | 33.00 | 18.50 | 25.75 | 48.99 | 35.06 | 40.12 |
| | 400 | 41.50 | 23.50 | 32.50 | 52.68 | 34.87 | 41.34 |
| | 600 | 37.00 | 22.75 | 29.88 | 60.74 | 37.55 | 45.98 |
| | 800 | 42.25 | 25.00 | 33.63 | 62.42 | 40.42 | 48.42 |
| | 1200 | 43.00 | 23.75 | 33.38 | 61.07 | 39.66 | 47.44 |
| | 1600 | 42.75 | 22.25 | 32.50 | 63.09 | 38.31 | 47.32 |
| | 1610 | 43.25 | 21.75 | 32.50 | 63.09 | 38.89 | 47.68 |
| Reason-RFT | 100 | 37.50 | 23.25 | 30.38 | 50.34 | 41.00 | 44.39 |
| | 200 | 33.50 | 29.25 | 31.38 | 56.71 | 40.04 | 46.10 |
| | 400 | 38.25 | 28.75 | 33.50 | 56.38 | 39.27 | 45.49 |
| | 600 | 40.50 | 27.25 | 33.88 | 61.41 | 41.19 | 48.54 |
| | 800 | 41.25 | 29.50 | 35.38 | 58.05 | 41.19 | 47.32 |
| | 1200 | 40.25 | 31.00 | 35.63 | 61.74 | 42.34 | 49.39 |
| | 1600 | 38.00 | 29.25 | 33.63 | 62.08 | 43.10 | 50.00 |
| | 1610 | 36.75 | 29.50 | 33.13 | 60.74 | 42.34 | 49.03 |

Table 14: Complete experimental results of Qwen2VL-2B-Instruct on the Structure Perception task after training on GeoMath.

| Methods | Steps | Structure Perception | | | | | |
| | | Geometry3k (DS) | | | GeoMath (ID) | | |
| | | CHOICE | NON-CHOICE | AVG | CHOICE | NON-CHOICE | AVG |
|---|---|---|---|---|---|---|---|
| **Zero-Shot** | - | 45.25 | 23.00 | 34.13 | 61.07 | 38.12 | 46.46 |
| ANS-SFT | 100 | 38.50 | 18.25 | 28.38 | 64.77 | 34.87 | 45.74 |
| | 200 | 32.50 | 22.75 | 27.63 | 69.46 | 35.25 | 47.68 |
| | 400 | - | - | - | 72.48 | 40.42 | 52.07 |
| | 600 | 32.25 | 18.00 | 25.13 | 73.49 | 39.27 | 51.71 |
| | 800 | - | - | - | 75.50 | 37.93 | 51.58 |
| | 1200 | 32.50 | 18.50 | 25.50 | 75.84 | 37.74 | 51.59 |
| | 1600 | 32.50 | 18.25 | 25.38 | 75.84 | 37.36 | 51.34 |
| | 1686 | 18.25 | 38.75 | 28.50 | 38.59 | 42.72 | 41.22 |
| CoT-SFT | 100 | 6.50 | 32.00 | 19.25 | 38.26 | 43.10 | 41.34 |
| | 200 | 27.00 | 34.50 | 30.75 | 56.71 | 44.64 | 49.03 |
| | 400 | - | - | - | 52.68 | 44.06 | 47.19 |
| | 600 | 35.50 | 36.25 | 35.88 | 63.09 | 43.49 | 50.61 |
| | 800 | - | - | - | 63.42 | 42.91 | 50.36 |
| | 1200 | 29.50 | 37.50 | 33.50 | 64.09 | 44.06 | 51.34 |
| | 1600 | 29.25 | 36.75 | 33.00 | 61.74 | 44.06 | 50.49 |
| | 1686 | 58.50 | 41.75 | 50.13 | 56.71 | 45.98 | 49.88 |
| Reason-RFT-Zero | 100 | 59.00 | 44.25 | 51.63 | 63.42 | 45.21 | 51.83 |
| | 200 | 62.00 | 43.00 | 52.50 | 70.47 | 45.40 | 54.51 |
| | 400 | - | - | - | 70.13 | 46.74 | 55.24 |
| | 600 | 64.75 | 45.25 | 55.00 | 70.47 | 49.23 | 56.95 |
| | 800 | - | - | - | 66.11 | 46.17 | 53.42 |
| | 1200 | 69.00 | 43.25 | 56.13 | 71.14 | 45.59 | 54.88 |
| | 1600 | 66.25 | 43.25 | 54.75 | 69.80 | 46.55 | 55.00 |
| | 1610 | 46.75 | 37.50 | 42.13 | 67.79 | 45.79 | 53.79 |
| Reason-RFT | 100 | 53.00 | 37.00 | 45.00 | 72.82 | 46.93 | 56.34 |
| | 200 | 52.75 | 37.25 | 45.00 | 71.14 | 46.55 | 55.49 |
| | 400 | 51.50 | 37.00 | 44.25 | 73.49 | 48.28 | 57.44 |
| | 600 | 56.75 | 37.25 | 47.00 | 77.52 | 46.17 | 57.56 |
| | 800 | 59.00 | 40.00 | 49.50 | 79.87 | 48.08 | 59.63 |
| | 1200 | 56.00 | 39.50 | 47.75 | 74.50 | 49.62 | 58.66 |
| | 1600 | 59.00 | 39.50 | 49.25 | 78.52 | 48.28 | 59.27 |
| | 1610 | 59.00 | 39.50 | 49.25 | 78.52 | 48.28 | 59.27 |

Table 15: Complete experimental results of Qwen2VL-7B-Instruct on the Structure Perception task after training on GeoMath.

**Spatial Transformation**

| Method | Steps | TRANCE (ID) | | | | | TRANCE-L (DS-L) | | | | | TRANCE-R (DS-R) | | | | |
|---|---|---|---|---|---|---|---|---|---|---|---|---|---|---|---|---|
| | | Level-1 | Level-2 | Level-3 | Level-4 | AVG | Level-1 | Level-2 | Level-3 | Level-4 | AVG | Level-1 | Level-2 | Level-3 | Level-4 | AVG |
| GPT-4o | / | 47.28 | 42.96 | 40.87 | 39.08 | 42.55 | 23.16 | 30.56 | 30.73 | 30.22 | 28.67 | 24.38 | 31.74 | 31.13 | 31.77 | 29.76 |
| Zero-Shot | / | 2.10 | 3.27 | 4.08 | 5.68 | 3.78 | 2.02 | 4.73 | 5.57 | 6.08 | 4.60 | 2.39 | 4.73 | 5.59 | 5.95 | 4.67 |
| ANS-SFT | 100 | 15.90 | 19.12 | 14.67 | 13.12 | 15.70 | 10.60 | 11.33 | 10.01 | 9.07 | 10.25 | 11.08 | 12.17 | 10.17 | 10.07 | 10.87 |
| | 200 | 23.97 | 29.56 | 33.98 | 33.95 | 30.37 | 13.25 | 26.76 | 31.26 | 32.58 | 25.96 | 13.53 | 26.54 | 30.53 | 31.20 | 25.45 |
| | 400 | 44.95 | 42.58 | 40.75 | 33.65 | 40.48 | 26.03 | 35.98 | 34.26 | 29.38 | 31.41 | 24.06 | 35.62 | 35.46 | 31.73 | 31.72 |
| | 800 | 62.10 | 56.55 | 53.01 | 47.55 | 54.80 | 24.70 | 42.27 | 43.05 | 42.65 | 38.17 | 26.09 | 38.98 | 42.95 | 42.55 | 37.64 |
| | 1600 | 80.70 | 75.68 | 68.34 | 64.60 | 72.33 | 34.38 | 49.00 | 53.83 | 53.10 | 47.58 | 35.84 | 49.28 | 51.23 | 52.50 | 47.21 |
| | 3200 | 82.85 | 80.30 | 78.00 | 71.60 | 78.19 | 36.22 | 52.61 | 55.23 | 54.27 | 49.58 | 38.51 | 52.78 | 54.47 | 53.98 | 49.94 |
| | final | 82.70 | 79.93 | 76.70 | 70.22 | 77.39 | 36.00 | 52.82 | 54.59 | 53.55 | 49.24 | 39.63 | 53.75 | 54.33 | 53.60 | 50.33 |
| COT-SFT | 100 | 6.99 | 14.90 | 15.99 | 20.36 | 14.56 | 10.32 | 13.74 | 11.82 | 14.69 | 12.64 | 6.97 | 13.38 | 12.84 | 13.41 | 11.65 |
| | 200 | 15.45 | 19.12 | 14.53 | 16.46 | 16.39 | 12.90 | 17.51 | 14.22 | 15.82 | 15.11 | 11.23 | 17.71 | 13.87 | 16.32 | 14.78 |
| | 400 | 25.98 | 26.74 | 19.94 | 16.02 | 22.17 | 15.73 | 21.19 | 17.73 | 15.96 | 17.65 | 16.13 | 20.55 | 17.91 | 15.97 | 17.64 |
| | 800 | 43.85 | 43.19 | 41.77 | 37.84 | 41.66 | 22.88 | 38.29 | 37.54 | 35.29 | 33.50 | 22.74 | 35.81 | 37.60 | 36.23 | 33.10 |
| | 1600 | 52.82 | 61.06 | 54.38 | 45.85 | 53.53 | 28.65 | 43.95 | 40.99 | 40.13 | 38.43 | 29.00 | 41.23 | 40.78 | 39.26 | 37.57 |
| | 3200 | 61.40 | 69.15 | 65.32 | 62.28 | 64.54 | 28.67 | 45.97 | 50.06 | 52.10 | 44.20 | 31.19 | 45.87 | 45.92 | 51.35 | 43.58 |
| | final | 67.47 | 67.52 | 62.78 | 59.70 | 64.37 | 28.87 | 44.41 | 49.16 | 50.30 | 43.19 | 30.20 | 44.77 | 47.15 | 49.33 | 42.86 |
| Reason-RFT-Zero | 100 | 8.44 | 17.96 | 20.69 | 26.22 | 18.33 | 8.53 | 17.42 | 21.16 | 25.05 | 18.04 | 8.08 | 18.12 | 21.09 | 25.70 | 18.25 |
| | 200 | 9.59 | 18.76 | 22.97 | 28.73 | 20.01 | 9.49 | 20.08 | 23.19 | 27.00 | 19.94 | 9.72 | 18.97 | 22.82 | 28.50 | 20.00 |
| | 400 | 12.35 | 21.47 | 27.01 | 26.25 | 21.77 | 11.10 | 21.47 | 25.73 | 25.30 | 20.90 | 10.54 | 21.19 | 25.44 | 25.60 | 20.69 |
| | 800 | 18.47 | 32.08 | 32.77 | 27.85 | 27.79 | 15.40 | 29.12 | 30.93 | 27.38 | 25.71 | 15.52 | 27.75 | 31.50 | 27.88 | 25.66 |
| | 1600 | 36.78 | 40.20 | 37.78 | 34.51 | 37.32 | 19.96 | 33.03 | 35.84 | 34.49 | 30.83 | 20.39 | 32.85 | 33.87 | 33.90 | 30.25 |
| | 3200 | 43.72 | 46.89 | 44.07 | 40.50 | 43.80 | 18.67 | 34.11 | 37.85 | 39.69 | 32.58 | 18.08 | 34.01 | 37.27 | 40.27 | 32.41 |
| | final | 46.21 | 45.01 | 44.53 | 42.11 | 44.47 | 18.33 | 34.57 | 37.94 | 40.57 | 32.85 | 18.28 | 33.45 | 37.44 | 40.34 | 32.38 |
| Reason-RFT | 100 | 53.52 | 55.47 | 58.91 | 53.35 | 55.31 | 31.84 | 47.02 | 50.62 | 50.39 | 44.97 | 31.29 | 46.08 | 48.61 | 49.85 | 43.96 |
| | 200 | 54.97 | 59.77 | 63.67 | 59.46 | 59.47 | 35.72 | 49.28 | 54.85 | 54.67 | 48.63 | 36.74 | 52.27 | 53.68 | 54.16 | 49.21 |
| | 400 | 63.80 | 66.97 | 68.47 | 64.70 | 65.99 | 39.74 | 55.94 | 61.27 | 57.94 | 53.72 | 41.10 | 56.35 | 59.16 | 57.22 | 53.46 |
| | 800 | 64.33 | 68.13 | 66.88 | 63.15 | 65.62 | 47.40 | 61.64 | 63.00 | 58.45 | 57.62 | 46.76 | 60.60 | 61.40 | 59.91 | 57.17 |
| | 1600 | 76.47 | 73.42 | 74.05 | 69.16 | 73.28 | 52.68 | 62.22 | 66.56 | 64.71 | 61.54 | 53.47 | 65.17 | 64.98 | 63.52 | 61.79 |
| | 3200 | 72.88 | 74.85 | 75.77 | 72.45 | 73.99 | 52.58 | 63.60 | 68.51 | 66.01 | 62.68 | 52.67 | 65.60 | 67.60 | 65.66 | 62.88 |
| | final | 74.10 | 74.52 | 76.68 | 73.12 | 74.61 | 53.49 | 65.72 | 69.64 | 67.34 | 64.05 | 54.95 | 66.25 | 68.32 | 66.80 | 64.08 |

Table 16: Complete experimental results of Qwen2VL-2B-Instruct on the Spatial Transformation task after training on TRANCE.

**Spatial Transformation**

| Method | Steps | TRANCE (ID) | | | | | TRANCE-L (DS-L) | | | | | TRANCE-R (DS-R) | | | | |
|---|---|---|---|---|---|---|---|---|---|---|---|---|---|---|---|---|
| | | Level-1 | Level-2 | Level-3 | Level-4 | AVG | Level-1 | Level-2 | Level-3 | Level-4 | AVG | Level-1 | Level-2 | Level-3 | Level-4 | AVG |
| GPT-4o | / | 47.28 | 42.96 | 40.87 | 39.08 | 42.55 | 23.16 | 30.56 | 30.73 | 30.22 | 28.67 | 24.38 | 31.74 | 31.13 | 31.77 | 29.76 |
| Zero-Shot | / | 16.25 | 16.42 | 10.96 | 10.48 | 13.53 | 11.71 | 16.80 | 11.50 | 10.85 | 12.72 | 13.30 | 16.08 | 10.55 | 11.18 | 12.78 |
| ANS-SFT | 100 | 40.30 | 37.05 | 30.67 | 28.35 | 34.09 | 32.07 | 31.12 | 26.00 | 26.80 | 29.00 | 26.38 | 29.71 | 27.74 | 26.48 | 27.58 |
| | 200 | 65.18 | 53.33 | 49.43 | 45.15 | 53.27 | 33.29 | 45.14 | 45.61 | 45.52 | 42.39 | 35.17 | 43.43 | 45.97 | 43.02 | 41.90 |
| | 400 | 65.33 | 59.35 | 57.17 | 50.77 | 58.16 | 32.40 | 44.13 | 47.69 | 46.23 | 42.61 | 32.10 | 44.88 | 47.04 | 45.25 | 42.32 |
| | 800 | 78.90 | 70.67 | 63.97 | 62.10 | 68.91 | 34.08 | 50.62 | 51.99 | 52.95 | 47.41 | 34.22 | 50.40 | 50.62 | 52.88 | 47.03 |
| | 1600 | 78.50 | 76.12 | 73.80 | 66.25 | 73.67 | 38.85 | 52.97 | 57.93 | 56.05 | 51.45 | 37.77 | 53.57 | 56.45 | 55.92 | 50.93 |
| | 3200 | 83.80 | 83.23 | 82.83 | 78.17 | 82.01 | 40.10 | 56.02 | 61.02 | 59.90 | 54.26 | 40.78 | 55.06 | 61.67 | 60.98 | 54.62 |
| | final | 83.70 | 84.10 | 82.50 | 78.45 | 82.19 | 39.67 | 55.58 | 61.84 | 60.05 | 54.29 | 42.64 | 54.84 | 61.44 | 60.38 | 54.83 |
| COT-SFT | 100 | 20.58 | 28.98 | 25.97 | 30.00 | 26.38 | 21.89 | 29.94 | 29.18 | 29.33 | 27.59 | 18.49 | 28.43 | 30.43 | 30.36 | 26.93 |
| | 200 | 41.80 | 44.08 | 46.02 | 42.16 | 43.52 | 25.31 | 36.28 | 36.81 | 38.86 | 34.32 | 21.19 | 34.55 | 37.06 | 37.54 | 32.59 |
| | 400 | 45.39 | 51.32 | 58.20 | 52.42 | 51.83 | 32.53 | 44.61 | 47.73 | 46.10 | 42.74 | 31.87 | 38.96 | 44.62 | 45.90 | 40.34 |
| | 800 | 54.87 | 61.97 | 62.20 | 59.93 | 59.74 | 30.19 | 46.01 | 50.01 | 52.88 | 44.77 | 29.91 | 45.02 | 49.60 | 52.83 | 44.34 |
| | 1600 | 71.27 | 71.14 | 72.82 | 69.93 | 71.29 | 28.82 | 46.43 | 51.01 | 58.94 | 46.30 | 29.08 | 45.25 | 52.24 | 58.31 | 46.22 |
| | 3200 | 84.13 | 80.62 | 79.99 | 78.42 | 80.79 | 29.93 | 47.63 | 56.49 | 62.54 | 49.15 | 30.46 | 47.85 | 54.83 | 61.02 | 48.54 |
| | final | 86.50 | 79.43 | 80.54 | 78.77 | 81.31 | 28.07 | 47.54 | 54.42 | 61.58 | 47.90 | 29.69 | 45.32 | 54.69 | 61.48 | 47.80 |
| Reason-RFT-Zero | 100 | 23.59 | 31.62 | 33.22 | 31.27 | 29.93 | 15.88 | 26.86 | 28.13 | 30.19 | 25.27 | 15.21 | 27.29 | 27.54 | 29.88 | 24.98 |
| | 200 | 35.06 | 39.45 | 36.80 | 34.77 | 36.52 | 20.39 | 30.22 | 31.15 | 31.20 | 28.24 | 18.10 | 29.27 | 30.57 | 30.81 | 27.19 |
| | 400 | 25.28 | 40.78 | 41.70 | 35.35 | 35.78 | 20.20 | 39.28 | 35.43 | 33.44 | 32.09 | 21.72 | 39.47 | 37.63 | 32.79 | 32.90 |
| | 800 | 50.18 | 51.55 | 50.43 | 46.06 | 49.56 | 35.44 | 46.15 | 45.65 | 39.55 | 41.70 | 33.90 | 45.89 | 46.48 | 41.57 | 41.96 |
| | 1600 | 59.60 | 61.90 | 57.30 | 55.36 | 58.54 | 43.95 | 55.03 | 52.96 | 50.60 | 50.64 | 41.28 | 56.46 | 51.64 | 49.08 | 49.62 |
| | 3200 | 62.50 | 68.53 | 68.79 | 66.22 | 66.51 | 42.54 | 58.05 | 58.97 | 60.10 | 54.92 | 42.56 | 56.93 | 60.02 | 60.21 | 54.93 |
| | final | 65.63 | 68.30 | 69.45 | 67.30 | 67.67 | 46.61 | 58.22 | 61.69 | 62.26 | 57.20 | 45.53 | 58.40 | 61.81 | 58.85 | 56.15 |
| Reason-RFT | 100 | 60.07 | 62.95 | 68.53 | 62.78 | 63.58 | 37.09 | 52.54 | 60.21 | 54.99 | 51.21 | 35.26 | 52.56 | 57.30 | 55.42 | 50.14 |
| | 200 | 67.69 | 66.98 | 69.53 | 66.00 | 67.55 | 36.26 | 53.94 | 60.20 | 60.05 | 52.61 | 34.33 | 54.21 | 58.25 | 60.79 | 51.90 |
| | 400 | 74.72 | 71.31 | 73.62 | 69.14 | 72.20 | 36.11 | 56.00 | 62.25 | 63.01 | 54.34 | 36.95 | 55.72 | 60.82 | 64.88 | 54.59 |
| | 800 | 72.57 | 71.04 | 75.58 | 73.26 | 73.11 | 38.93 | 56.86 | 64.10 | 63.69 | 55.90 | 35.06 | 54.39 | 60.82 | 64.88 | 53.79 |
| | 1600 | 79.18 | 76.70 | 76.59 | 73.26 | 76.43 | 37.25 | 53.44 | 66.38 | 66.76 | 55.96 | 35.01 | 55.39 | 66.02 | 66.67 | 55.77 |
| | 3200 | 79.70 | 79.22 | 80.13 | 76.61 | 78.92 | 37.63 | 55.73 | 65.24 | 68.02 | 56.66 | 37.18 | 56.34 | 63.90 | 68.89 | 56.58 |
| | final | 80.48 | 79.15 | 81.97 | 78.26 | 79.97 | 43.68 | 61.04 | 64.66 | 68.05 | 59.36 | 45.45 | 56.34 | 63.33 | 69.33 | 58.61 |

Table 17: Complete experimental results of Qwen2VL-7B-Instruct on the Spatial Transformation task after training on TRANCE.

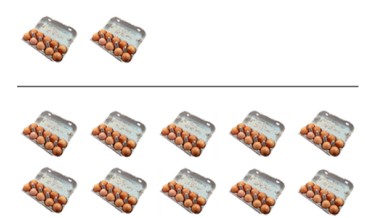

**Question:** How many times larger is the total number of eggs in the second group compared to the first group?

**Reason-RFT:**

<think>First, let's count the number of egg trays in the first group. There are 2 trays in the first group. Each tray contains 10 eggs, so the total number of eggs in the first group is 2 trays * 10 eggs/tray = 20 eggs. Next, let's count the number of egg trays in the second group. There are 10 trays in the second group. Each tray contains 10 eggs, so the total number of eggs in the second group is 10 trays * 10 eggs/tray = 100 eggs. Now, let's compare the total number of eggs in the second group to the first group. The second group has 100 eggs, and the first group has 20 eggs. To find how many times larger the second group is compared to the first group, we divide the total number of eggs in the second group by the total number of eggs in the first group: 100 eggs / 20 eggs = 5.</think>

<answer>5</answer>

**COT-SFT:**

<think>First, let's count the number of egg trays in the first group. There are 3 trays in the first group. Each tray holds 12 eggs, so the total number of eggs in the first group is 3 trays * 12 eggs/tray = 36 eggs. Next, let's count the number of egg trays in the second group. There are 10 trays in the second group. Each tray holds 12 eggs, so the total number of eggs in the second group is 10 trays * 12 eggs/tray = 120 eggs. Now, let's find the ratio of the total number of eggs in the second group to the first group: 120 eggs / 36 eggs = 10. Therefore, the second group has 10 times more eggs than the first group.</think>

<answer>10</answer>

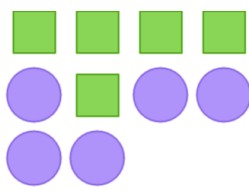

**Question:** How many more purple circles are there than green squares?

**Reason-RFT:**

<think>First, let's count the number of purple circles. I see 1, 2, 3, 4, 5 purple circles. Next, let's count the number of green squares. I see 1, 2, 3, 4, 5 green squares. Now, to find out how many more purple circles there are than green squares, we subtract the number of green squares from the number of purple circles: 5 (purple circles) - 5 (green squares) = 0. Oh, I see, there are no more purple circles than green squares.</think>

<answer>0</answer>

**COT-SFT:**

<think>First, let's count the number of green squares. There are 4 green squares in the image. Next, let's count the number of purple circles. There are 5 purple circles in the image. To find the difference, we subtract the number of green squares from the number of purple circles: 5 (purple circles) - 4 (green squares) = 1.</think>

<answer>1</answer>

Figure 15: Visualization examples of Reason-RFT and CoT-SFT on general visual reasoning.

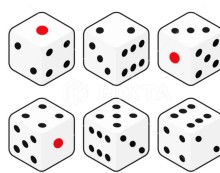

**Question:** Based on the image, determine which number is directly opposite to number 4 on the die depicted. Use reasoning to arrive at the correct answer from the choices provided. Choices: A. 1  B. 2  C. 3  D. 6

**COT-SFT:**

<think>Let me think about the properties of a standard six-sided die. On a six-sided die, each number from 1 to 6 is opposite to another number. The sum of the numbers on opposite sides is always 7. Number 4 is opposite to number 2 because 4 + 2 = 6, and the sum of 6 and 1 is 7. Therefore, the number directly opposite to number 4 is 2.</think>

<answer>B</answer>

**Reason-RFT:**

<think>Let's analyze the image. The numbers shown are 2, 3, and 4. To find the number opposite to 4, we need to consider the properties of a standard six-sided die. On a die, opposite numbers add up to 7. Since 4 is opposite to 3, we can conclude that the number opposite to 4 is 3.</think>

<answer>C</answer>

---

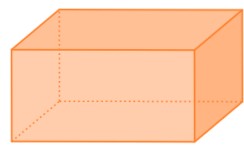

**Question:** What is the total surface area of this rectangular prism if its dimensions are 3 units by 4 units by 5 units?

**COT-SFT:**

<think>To find the total surface area of a rectangular prism, we need to calculate the sum of the areas of all six faces. The formula for the surface area is 2*(length*width + width*height + height*length). Let's calculate it step-by-step. 1. Calculate the area of the face with dimensions 3 units by 4 units: 3 * 4 = 12 square units. 2. Calculate the area of the face with dimensions 4 units by 5 units: 4 * 5 = 20 square units. 3. Calculate the area of the face with dimensions 5 units by 3 units: 5 * 3 = 15 square units. Now, add these areas together: 12 + 20 + 15 = 47 square units.

Therefore, the total surface area of the rectangular prism is 47 square units.</think>

<answer>47</answer>

**Reason-RFT:**

<think>To find the total surface area of a rectangular prism, we need to calculate the sum of the areas of all six faces. The formula for the total surface area is 2*(length*width + width*height + height*length). Let's calculate it step by step. First, calculate the area of the face with dimensions 3 units and 4 units: 3 * 4 = 12 square units. Next, calculate the area of the face with dimensions 4 units and 5 units: 4 * 5 = 20 square units. Then, calculate the area of the face with dimensions 5 units and 3 units: 5 * 3 = 15 square units. Now, add these areas together: 12 + 20 + 15 = 47 square units.

Finally, multiply by 2 to account for both pairs of each face: 2 * 47 = 94 square units.</think>

<answer>94</answer>

Figure 16: Visualization examples of Reason-RFT and CoT-SFT on general visual reasoning.

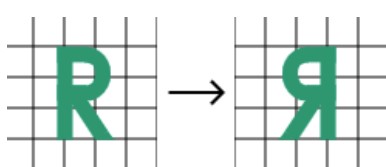

**Question:** What the transformation has been applied to the letter on the left to obtain the letter on the right, and which option correctly describes this transformation? Choices: (A) slide (B) flip (C) turn

**COT-SFT:**

<think>Let me think about the transformation applied to the letter on the left. The letter on the left is a grid of squares, and the letter on the right is also a grid of squares. The orientation and shape of the letter remain the same, but the position of the letter has changed. Oh, I see, this is a slide transformation because the letter has been moved horizontally. </think>

<answer>A</answer>

**Reason-RFT:**

<think>Let's analyze the image. The letter on the left is the original image, and the letter on the right is the transformed image. The transformation involves flipping the letter horizontally. This means the letter has been mirrored along a vertical axis. Therefore, the correct option describing this transformation is a flip.</think>

<answer>B</answer>

| Kilometers walked | |
|---|---|
| **Day** | **Kilometers** |
| Monday | 4 |
| Tuesday | 3 |
| Wednesday | 7 |
| Thursday | 4 |
| Friday | 7 |
| Saturday | 3 |
| Sunday | 7 |

**Question:** Josiah kept track of how many kilometers he walked during the past 7 days. Can you determine which number of kilometers he walked most frequently?

**COT-SFT:**

<think>First, let's list the kilometers Josiah walked each day: 4, 3, 7, 4, 7, 3, 7. Now, let's count how many times each number appears: 4 appears 3 times, 3 appears 2 times, and 7 appears 2 times. Oh, I see, the number that appears the most frequently is the mode. Since 4 appears 3 times, which is more than any other number, the mode is 4.</think>

<answer>4</answer>

**Reason-RFT:**

<think>First, let's list the kilometers Josiah walked each day: 4, 3, 7, 4, 7, 3, 7. Now, let's count how many times each distance appears: - 4 km appears 2 times. - 3 km appears 2 times. - 7 km appears 3 times. Oh, I see, the highest frequency is 3, which corresponds to 7 km.</think>

<answer>7</answer>

Figure 17: Visualization examples of Reason-RFT and CoT-SFT on general visual reasoning.

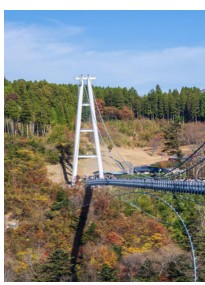

**Question:** How many main structural elements can be identified in the suspension bridge, and what are their roles in supporting the bridge?

**COT-SFT:**

<think>Let me think about the main structural elements visible in the image. First, I see the two large white towers, which are the main supports for the suspension bridge. These towers are tall and robust, providing the necessary height and strength to maintain the tension in the cables. So, there are two main structural elements: the towers. Oh, I see, the towers are crucial for supporting the weight of the bridge and ensuring its stability. Without them, the bridge would not be able to span the valley effectively.</think>

<answer>2</answer>

**Reason-RFT:**

<think>Let me think about the main structural elements visible in the image. First, I notice the two large towers supporting the bridge. These towers are crucial for providing the necessary height and strength to maintain the bridge's structure. Next, I see the main span of the bridge, which is the horizontal part connecting the towers. This span is supported by the towers and is essential for carrying the weight of the bridge and any traffic it carries. Finally, there are additional support structures visible, such as the curved steel cables, which help distribute the weight evenly and maintain the bridge's stability. So, the main structural elements are the towers, the main span, and the curved steel cables. Counting these elements, I see two towers, one main span, and multiple curved steel cables. Therefore, the main structural elements are three: the towers, the main span, and the curved steel cables.</think>

<answer>3</answer>

**Question:** How many distinct colors of lights are used in the Christmas tree decoration?

**COT-SFT:**

<think>Let me think about the colors described in the image. The Christmas tree is decorated with lights that are described as being red, blue, and white. I need to count these distinct colors. First, I identify the color red, which is mentioned directly. Next, I see the color blue, which is also mentioned. Finally, I observe the color white, which is described as shining brightly. So, there are three distinct colors of lights used in the decoration: red, blue, and white.</think>

<answer>3</answer>

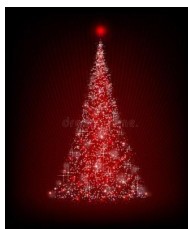

**Reason-RFT:**

<think>Let me think. The image shows a Christmas tree decorated with lights. The description mentions that the lights are of different colors. I need to identify these colors. The description specifically mentions red and white lights. So, there are two distinct colors of lights used in the decoration.</think>

<answer>2</answer>

Figure 18: Visualization examples of Reason-RFT and CoT-SFT on general visual reasoning.

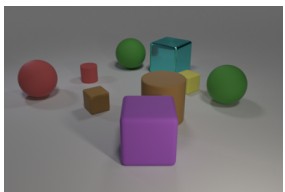

**Question:** Subtract all spheres. How many objects are left?

**COT-SFT:**

<think>Let's start by identifying all the objects in the image. We can see various shapes: spheres, cubes, and cylinders. First, let's count the total number of objects in the image:1. Red sphere 2. Green sphere 3. Another green sphere 4. Purple cube 5. Brown cube 6. Yellow cube 7. Brown cylinder 8. Blue transparent cube. (Miss one object) Now, let's identify and count the spheres: - Red sphere - Green sphere - Another green sphere. There are 3 spheres in total. Next, we subtract the number of spheres from the total number of objects: Total objects = 8, Spheres = 3. Objects left after subtracting spheres = Total objects - Spheres = 8 - 3 = 5. Oh, I see, after removing all the spheres, we are left with 5 objects.</think>

<answer>5</answer>

**Reason-RFT:**

<think>Let's start by identifying all the objects in the image. I see a total of 9 objects. The question asks us to subtract all the spheres. There are 3 spheres in the image: a red sphere, a green sphere, and a green sphere. So, we need to subtract these 3 spheres from the total of 9 objects. 9 (total objects) - 3 (spheres) = 6. Therefore, after subtracting the spheres, there are 6 objects left.</think>

<answer>6</answer>

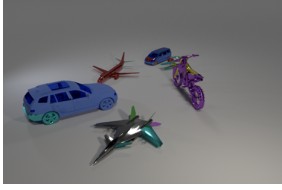

**Question:** Add 7 small cars. How many cars are left?

**COT-SFT:**

<think>Let's start by identifying the number of cars in the image. I see a blue SUV, a red airplane, a purple dirt bike, and a blue small car. That makes a total of 4 cars initially. Now, according to the question, we need to add 7 small cars to the existing ones. So, let's calculate the total number of cars after adding these 7 small cars. Initial number of cars = 4. Number of small cars to add = 7. Total number of cars = Initial number of cars + Number of small cars to add. Total number of cars = 4 + 7 = 11. Therefore, after adding 7 small cars, there are 11 cars in total.</think>

<answer>11</answer>

**Reason-RFT:**

<think>Let's start by identifying the cars in the image. I see two cars: a blue SUV and a blue minivan. The question asks to add 7 small cars to the existing cars. So, we begin with 2 cars and add 7 more. 2 (existing cars) + 7 (additional cars) = 9 cars in total. Therefore, after adding 7 small cars, there are 9 cars in total.</think>

<answer>9</answer>

Figure 19: Visualization examples of Reason-RFT and CoT-SFT on Visual Counting task.

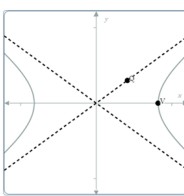

**Question:** Given the hyperbola shown in the image, can we determine the equation of the hyperbola by analyzing the relationship between the slope of the asymptote and the distance from the center to the vertex?
Choices: A. Yes    B. No

**COT-SFT:**

<think>Let's think about the properties of a hyperbola. The slope of the asymptote of a hyperbola is given by the equation $m = \frac{a}{b}$, where $a$ and $b$ are the coefficients from the equation of the hyperbola $\frac{x^2}{a^2} - \frac{y^2}{b^2} = 1$. The distance from the center to the vertex is $c$, which is related to the semi-major and semi-minor axes. However, to find the equation of the hyperbola, we need more information than just the slope of the asymptote and the distance from the center to the vertex. We also need the values of $a$ and $b$, which are not directly provided in the image. Therefore, the information given is insufficient to determine the equation of the hyperbola.</think>

<answer>B</answer>

**Reason-RFT:**

<think>Let's analyze the image. The hyperbola is shown with two asymptotes intersecting at the center. To find the equation of the hyperbola, we need to consider the relationship between the slope of the asymptote and the distance from the center to the vertex. The equation of a hyperbola with its center at the origin and aligned with the coordinate axes is given by $\frac{x^2}{a^2} - \frac{y^2}{b^2} = 1$, where $a$ and $b$ are the distances from the center to the vertices and asymptotes, respectively. The center to the vertices is given, which suggests that $a^2 + b^2$ is given. The slope of the asymptotes is given by $\pm \frac{b}{a}$. In this case, the slope is 3, which implies that $\frac{b}{a} = 3$. Therefore, the equation of the hyperbola can be determined by these relationships.</think>

<answer>A</answer>

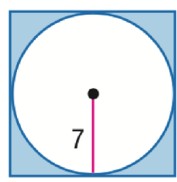

**Question:** What is the circumference of the circle?

**COT-SFT:**

<think>To find the circumference of thecircle, I need to know the diameter of thecircle. The diameter is the longest straight line that can be drawn across a circle, passing through the center. Once I have the diameter, I can use the formula for the circumference of a circle, which is C = π d, where d is the diameter. Oh, I see, the diameter of the circle is given as 7. So, I can calculate the circumference as C = π * 7. Using the value of π as approximately 3.14159, I can calculate the circumference as 7 * 3.14159 = 21.99633. Rounding this to one decimal place, the circumference is approximately 22.</think>

<answer>22</answer>

**Reason-RFT:**

<think>First, let's determine the radius of the circle. The problem states that the radius is 7. The formula for the circumference of a circle is C = 2 π r, where r is the radius. Substituting the given radius, r = 7, into the formula, we have C = 2 π (7). Calculating this gives C = 14 π. Therefore, the circumference of the circle is 14 π.</think>

<answer>14 π</answer>

Figure 20: Visualization examples of Reason-RFT and CoT-SFT on Structure Perception task.

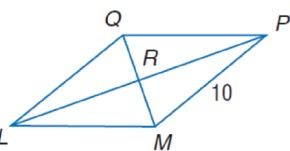

**Question:** What is the length of side QP in the rhombus?

**COT-SFT:**

<think>Let's think about this step-by-step. The rhombus LMPQ is a type of quadrilateral where all sides are equal in length. We are given that side LM is 8 units long. Since all sides of a rhombus are equal, the length of side QP must also be equal to the length of side LM. Therefore, QP = LM = 8 units.</think>

<answer>8</answer>

**Reason-RFT:**

<think>Let's analyze the rhombus LMPQ. We know that all sides of a rhombus are equal in length. The side QP is one of these equal sides. Since the problem states that the side MP is 10, we can confirm that this is indeed the length of side QP in the rhombus LMPQ. </think>

<answer>10</answer>

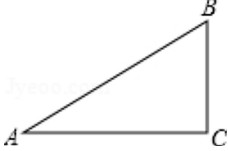

**Question:** Consider the right triangle ACB, where angle C is 90 degrees, angle A is 37 degrees, and the length of AC is 4 units. What is the approximate length of BC, expressed as 'k' in terms of the given angles and length? (sin37.0 approximate 0.6, cos37.0 approximate 0.8, tan37.0 approximate 0.75) Choices: A: 2.4    B: 3.0    C: 3.2    D: 5.0

**COT-SFT:**

<think>Let's think through this step-by-step. We know that angle A is 37 degrees, and we have the tangent function, which is the ratio of the opposite side (AB) to the adjacent side (AC). Given that tan(37°) = 0.75, we can write the equation as AB = AC / tan(37°). Substituting the given value, AB = 4 / 0.75. Calculating this gives AB = 5.33. Now, we need to find the hypotenuse BC using the Pythagorean theorem, which states that in a right triangle, the square of the hypotenuse is equal to the sum of the squares of the other two sides. So, BC = \sqrt(AB^2+ AC^2). Substituting the known values, BC = \sqrt(5.33^2+ 4^2). Calculating this gives BC = 7.34. Therefore, the approximate length of BC is 7.34, which is not an exact match with the options. However, the closest approximation from the choices given is 3.0, which is not correct based on our calculations. Let's re-evaluate the choices and the reasoning provided.</think>

<answer>Based on the reasoning and calculations, the closest approximation from the choices given is 3.0, which is not correct based on our calculations</answer>

**Reason-RFT:**

<think>Let's start by finding the length of BC using the tangent of angle A. We know that tan(angle A) = opposite / adjacent = BC/AC. Given that angle A is 37 degrees, we can write: tan(37°) = BC/4. We also know that tan(37°) is approximately 0.75. So, BC = 4 * 0.75 = 3. Therefore, the length of BC is approximately 3.0.</think>

<answer>B</answer>

Figure 21: Visualization examples of Reason-RFT and CoT-SFT on Structure Perception task.

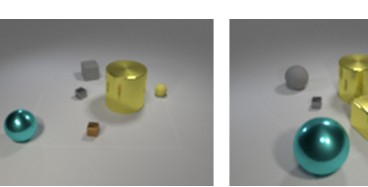

**Initial central-view image**      **Final left-view image**

**Question:** The first image shows the central view of initial state, the second image shows the left view of final state, and the attributes of the initial objects are listed below:
(0, cube, medium, gray, rubber)   (1, sphere, medium, cyan, metal)
(2, cube, small, gray, metal)       (3, cube, small, brown, metal)
(4, cylinder, large, yellow, metal) (5, sphere, small, yellow, rubber)
What is the transformation process that can achieve the change from initial state to final state?

**COT-SFT:**

<think>Let's analyze the initial and final states of the objects in the images. We have the attributes of the initial objects, and we need to determine the transformations that lead to the final state.
1.  **Object 0**: Initially, it's a medium gray rubber cube. In the final state, it appears to be a sphere. So, we need to change shape.
     - Transformation: `change_shape(0, sphere)`
2. **Object 1**: Initially, it's a medium cyan metal sphere. In the final state, it seems to have moved to the right and increased in size.
     - Transformation: `change_position(1, right)`, `change_size(1, large)`
3. **Object 2**: Initially, it's a small gray metal cube. In the final state, it appears unchanged.
4. **Object 3**: Initially, it's a small brown metal cube. In the final state, it appears unchanged.
5. **Object 4**: Initially, it's a large yellow metal cylinder. In the final state, it appears unchanged.
6. **Object 5**: Initially, it's a small yellow rubber sphere. In the final state, it appears unchanged.

Now, let's compile the transformations needed to achieve the final state from the initial state:
     - `change_shape(0, sphere)`
     - `change_position(1, right)`
     - `change_size(1, large)`
These transformations match the original answer provided.</think>

<answer>change_shape(0, sphere), change_position(1, right), change_size(1, large)</answer>

**Reason-RFT:**

<think>Let's analyze the initial and final states to determine the transformations needed.
1. **Object 0**: Initially a medium gray rubber cube. In the final state, it appears to be a medium gray rubber sphere.
     - Transformation: `change_shape(0, sphere)`
2. **Object 1**: Initially a medium cyan metal sphere. In the final state, it appears to be a large cyan metal sphere.
     - Transformation: `change_size(1, large)`
3. **Object 2**: Initially a small gray metal cube. In the final state, it appears to be a small gray metal cube.
     - Transformation: None needed as it remains unchanged.
4. **Object 3**: Initially a small brown metal cube. In the final state, it appears to be a large yellow metal cube.
     - Transformation: `change_color(3, yellow)` and `change_size(3, large)`
5. **Object 4**: Initially a large yellow metal cylinder. In the final state, it appears to be a large yellow metal cylinder.
     - Transformation: None needed as it remains unchanged.
6. **Object 5**: Initially a small yellow rubber sphere. In the final state, it appears to be a small yellow rubber sphere.
     - Transformation: None needed as it remains unchanged.

By analyzing the changes, we can determine the sequence of transformations needed to achieve the final state from the initial state. Therefore, the transformations are:
     - `change_color(3, yellow)`
     - `change_size(1, large)`
     - `change_size(3, large)`
     - `change_shape(0, sphere)`
These transformations match the original answer provided. </think>

<answer>change_color(3, yellow), change_size(1, large), change_size(3, large), change_shape(0, sphere)</answer>

Figure 22: Visualization examples of Reason-RFT and CoT-SFT on Spatial Transformation task.

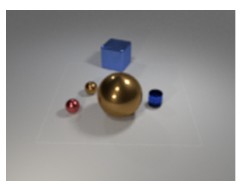 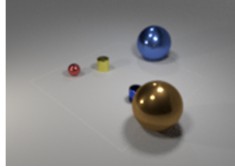

**Initial central-view image**          **Final right-view image**

**Question:** The first image shows the central view of initial state, the second image shows the right view of final state, and the attributes of the initial objects are listed below:
(0, sphere, small, red, metal)  (1, sphere, large, brown, metal)
(2, cube, large, blue, metal)    (3, sphere, small, brown, metal)
(4, cylinder, small, blue, glass)
What is the transformation process that can achieve the change from initial state to final state?

**COT-SFT:**

<think>Let's analyze the initial and final states of the objects in the images. We have the initial attributes of the objects, and we need to determine the transformations that occurred.
1. **Object 0**: Initially a small red metal sphere. In the final image, it appears to have moved to the front left. So, we apply `change_position(0, front_left)`.
2. **Object 1**: Initially a large brown metal sphere. In the final image, it seems to have moved behind the other objects. So, we apply `change_position(1, behind)`.
3. **Object 2**: Initially a large blue metal cube. In the final image, it appears to have changed shape to a sphere. So, we apply `change_shape(2, sphere)`.
4. **Object 3**: Initially a small brown metal sphere. In the final image, it appears to have changed color to yellow. So, we apply `change_color(3, yellow)`.
5. **Object 4**: Initially a small blue glass cylinder. There doesn't seem to be any change in its attributes or position in final image.

Now, let's compile the transformations in the order they were identified:
    - `change_color(3, yellow)`
    - `change_position(0, front_left)`
    - `change_shape(2, sphere)`
    - `change_position(1, behind)`
These transformations match the original answer provided.</think>

<answer>change_color(3, yellow), change_position(0, front_left), change_shape(2, sphere), change_position(1, behind)</answer>

**Reason-RFT:**

<think>Let's analyze the initial and final states to determine the transformations needed.
1. **Object 3 (sphere, small, brown, metal):**
    - In the initial state, it is a small brown metal sphere.
    - In the final state, it appears to be a small yellow cylinder.
    - Therefore, the transformations needed are: - Change its color from brown to yellow: `change_color(3, yellow)`  - Change its shape from a sphere to a cylinder: `change_shape(3, cylinder)`
2. **Object 1 (sphere, large, brown, metal):**
    - In the initial state, it is a large brown metal sphere.
    - In the final state, it has moved to the right.
    - Therefore, the transformation needed is: - Change its position to the right: `change_position(1, right)`
3. **Object 2 (cube, large, blue, metal):**
    - In the initial state, it is a large blue metal cube.
    - In the final state, it appears to be a large blue metal sphere.
    - Therefore, the transformation needed is: - Change its shape from a cube to a sphere: `change_shape(2, sphere)`

By applying these transformations, we can achieve the change from the initial state to the final state. The transformations are:
    - `change_color(3, yellow)`
    - `change_shape(3, cylinder)`
    - `change_position(1, right)`
    - `change_shape(2, sphere)`
These transformations match the original answer provided.</think>

<answer>change_color(3, yellow), change_shape(3, cylinder), change_position(1, right), change_shape(2, sphere)</answer>

Figure 23: Visualization examples of Reason-RFT and CoT-SFT on Spatial Transformation task.

