# OpenReview forum: "Reason-RFT: Reinforcement Fine-Tuning for Visual Reasoning of Vision Language Models"
_NeurIPS.cc/2025/Conference — NeurIPS 2025 poster_

### Official Review · Reviewer_yjNn · 2025-06-06

**Clarity:** 3
**Significance:** 2
**Originality:** 2
**Rating:** 4
**Confidence:** 4

**Summary:**

The paper introduces Reason-RFT, a two-stage post-training pipeline that aims to endow Vision–Language Models (VLMs) with stronger and more transferable visual-reasoning skills.

Stage 1 – SFT activation. The authors first perform SFT with CoT–annotated date so that the model learns step-by-step solutions with correct format.

Stage 2 – RL. They then continue training with GRPO and two rewards are leveraged: format reward and accuracy reward.

To evaluate the models, authors reconstructed a benchmark covering visual counting, structural perception, and spatial transformation.

Empirically, Reason-RFT, built on Qwen2-VL-2B/7B, (i) reaches or surpasses SOTA open and proprietary models on all three tasks, (ii) retains the lead under significant domain shift, and (iii) achieves > 90 % of full-data SFT performance using < 5 % of the data.

**Questions:**

* Have you tried more recent and advanced VLMs, such as Qwen-2.5-VL?

* Have you tried RL without the format reward?

* Do you observe any "thinking" words like "wait" or "let me verify that again"? From the examples you provided, I don’t see a clear difference between the CoT content and what's inside the <think></think> tags. Does this format really make a difference here?

**Ethical Concerns:**

["NO or VERY MINOR ethics concerns only"]

**Final Justification:**

With the detailed and new results provided by the authors in the rebuttal, I have a better understanding of their motivation and evaluations of the models. With the assumption that all the new content will be included in the revised version of this paper, I'd like to increase the rating by one point.

**Limitations:**

Yes

**Quality:**

3

**Strengths And Weaknesses:**

## Strengths

* Empirical results indicate that the proposed Reason-RFT is effective on visual tasks like counting, perception, etc.
* Evaluations and ablation studies are detailed and comprehensive.
* The code and data release could be useful for future research.

## Weaknesses

* Two core parts of the Reason-RFT is directly from existing work. The findings are expected and the novelty is limited.
* The designed tasks are relatively easy, which limits the contribution of this work and the performance of RL. In Fig.5a, the avg response  length is ~70, which is far shorter than modern reasoning models (e.g., 8k, 16k, 32k generation lengths).
* Lack of evaluations on more challenging benchmarks, such as MMMU, MMMU-Pro, MathVista, MathVision, etc. These are more crucial benchmarks for reflecting the reasoning abilities of VLMs.

---

> ### Author Rebuttal · Authors · 2025-07-31
>
> We sincerely thank the reviewer for the encouraging and constructive feedback. We appreciate your recognition of our strong empirical results, thorough evaluations and ablations, and the potential impact of our code and dataset release. Below, we address your specific concerns in detail.
>
> > **_`W1`: "The novelty of our work."_**
>
> A: Thanks. We would like to clarify that our work is not a direct extension or simple combination of existing methods. Instead, it introduces **a novel and unified post-training methodology** specifically tailored to the core challenges of visual reasoning, such as **cognitive rigidity** and **generalization to domain shift** in dynamic visual reasoning.
>
> In contrast to previous works that either ***(i)*** focus exclusively on text-based reasoning (`[1]`); ***(ii)*** adopt only-SFT (`[2,3]`) or only-RL (`[4,5,6]`) strategies for post-training in isolation; or ***(iii)*** involve a few concurrent works (`[7,8]`) that combine SFT and RL but remain restricted to static visual tasks with narrowly defined reward types—our approach is **the first** to introduce a **generalizable training framework** that systematically integrates SFT and RFT in a **tightly coupled manner**, effectively addressing **both static** (e.g., structural perception) and **dynamic** (e.g., temporal visual counting, spatial transformation) visual reasoning under a unified reward formulation.
>
> The novelty of our approach lies in the following aspects:
>
> - **An adaptive training sample allocation strategy** that dynamically balances intermediate reasoning supervision and reward-guided exploration. This approach strengthens the coupling between SFT and RFT, enabling a unified learning process aligned with visual reasoning goals—unlike prior methods that treat SFT and RL as separate stages.
>
> - **A unified visual reasoning reward formulation** that generalizes across diverse task formats. Beyond replicating standard reward templates, we introduce a principled reward design that captures reasoning correctness and completeness in a task-agnostic manner:
>   - **Structured-format reward** (including summary, caption, thinking and answer tags) to enhance the quality and coherence of reasoning chains.
>   - **Discrete-value reward** for tasks with discrete-value outputs (e.g., classification and counting).
>   - **Tolerance-based reward** for tasks requiring continuous-value outputs (e.g., geometric estimation, visual grounding).
>   - **Function-matching reward** for tasks with sequence-like outputs (e.g., action planning or tool calling).
>
>   These reward formulations are designed to be task-agnostic and applicable, moving **BEYOND** handcrafted, task-specific designs commonly used in prior work.
>
> - **A comprehensive visual reasoning evaluation protocol** that spans in-domain and domain-shifted settings across both synthetic and real-world environments. It systematically benchmarks key reasoning skills—such as dynamic reasoning, geometric perception, and arithmetic—to assess generalization and robustness.
>
> > [1] *DeepSeek-R1*, ArXiv'25.
>
> > [2] *LLaVA-CoT*, ArXiv'24.
>
> > [3] *LlamaV-o1*, ACL'25.
>
> > [4] *Visual-RFT*, ICCV'25.
>
> > [5] *VLM-R1*, ArXiv'25.
>
> > [6] *R1-Zero’s “Aha Moment”*, ArXiv'25.
>
> > [7] *R1-Onevision*, ICCV'25.
>
> > [8] *Vision-R1*, ArXiv'25.
>
> ---
>
> > **_`W2`: "About the CoT length."_**
>
> A: Thanks. First, we clarify that our tasks are far from trivial. Even top proprietary models like **GPT-4o** and **Gemini 1.5 Pro** scored below `50%` across all tasks. For example, GPT-4o achieved only `35.9%` on the Spatial Transformation task, revealing a key limitation in current VLMs. In contrast, our **Reason-RFT 2B model** reached `67.6%`, demonstrating the effectiveness of our approach.
>
> As for CoT response length, we note that visual reasoning fundamentally differs from traditional text reasoning (e.g., math problems). Our tasks rely on perceptual inference and visual comparison, not verbose textual elaboration. For instance, in Spatial Transformation, models must recognize visual state changes and infer spatial relations—this rarely requires long-form output. Thus, an average output length of `~70` tokens is both sufficient and appropriate, and longer responses do not necessarily imply stronger reasoning.
>
> ---
>
> > **_`W3`: "More experiments on more benchmarks."_**
>
> A: Thanks. Our primary goal is **NOT** to achieve the best results on general VQA benchmarks, but to explore an important challenge: **how to enhance domain-specific complex reasoning without severely degrading general capabilities.**
>
> |Method|MMMU|RealWorldQA|MathVision|AI2D|
> |-|-|-|-|-|
> |Zero-Shot (7B)|54.1|67.2|16.3|83.0|
> |+ANS-SFT|42.7 (-11.4)|48.1 (-19.1)|9.1 (-7.2)|78.3 (-4.7)|
> |+CoT-SFT|44.7 (-9.4)|36.5 (-30.7)|15.3 (-1.0)|73.3 (-9.8)|
> |+Reason-RFT-Zero|46.4 (-7.7)|45.1 (-22.1)|10.9 (-5.4)|75.3 (-7.7)|
> |+Reason-RFT|50.0 **(-4.1)** |61.3 **(-5.9)**|17.6 **(+1.3)**|81.7 **(-1.3)**|
>
> Reason-RFT achieves **more stable performance** than other methods, with smaller drops across general benchmarks. For instance, on RealWorldQA, it reduces the accuracy drop to only –5.9 (vs. –19.1 for ANS-SFT and –30.7 for CoT-SFT). On MathVision, it even improves over the zero-shot baseline (+1.3). These results demonstrate Reason-RFT’s potential in **balancing specialized reasoning and general capability**.
>
>
> Further details will be included in Appendix C of the revised version.
>
> ---
>
> > **_`Q1`: "More experiments on Qwen2.5-VL."_**
>
> A: Thanks. To further validate the effectiveness of our method, we conducted experiments on **Qwen2.5-VL-3B-Instruct**. The results are summarized below:
>
> |Model|Setting|T1-1D|T1-DSD|T1-DSM|AVG|T2-ID|T2-DS|AVG|T3-ID|T3-DSL|T3-DSR|AVG|
> |-|-|-|-|-|-|-|-|-|-|-|-|-|
> |qwen2.5vl-3b|ZERO|75.9|50.9|4.4|43.7|36.8|37.4|37.1|8.6|8.3|8.3|8.4|
> ||ANS|97.4|51.5|6.0|51.6|53.0|31.8|42.4|**91.1**|47.0|46.8|61.6|
> ||COT|89.2|59.5|49.2|66.0|56.1|49.4|52.7|81.6|46.1|44.2|57.3|
> ||RFT-ZERO|**99.0**|58.9|10.8|56.2|54.8|54.5|54.6|68.5|49.5|48.0|55.3|
> ||Reason-RFT|98.8|**68.7**|**54.8**|**74.1**|**59.0**|**56.6**|**57.8**|86.7|**55.2**|**54.4**|**65.4**|
>
>
> - **(1) Clear Improvement in Visual Reasoning:**
>   - Reason-RFT outperforms:
>     - ANS by `+22.5` (T1), `+15.4` (T2), `+3.8` (T3)
>     - COT by `+8.1` (T1), `+5.1` (T2), `+8.1` (T3)
>     - RFT-ZERO by `+17.9` (T1), `+3.2` (T2), `+10.1` (T3)
>
> - **(2) Strong Generalization under Domain Shift:**
>   Specifically, on the T1-DSM subset `+44.0` over RFT-ZERO
>
> - **(3) Data Efficiency:**
>   Similar to Qwen2VL, the Reason-RFT training paradigm on Qwen2.5VL demonstrates high data efficiency:
>   achieving `~70%` of the final performance of Reason-RFT-ZERO using only `<5%` of the data.
>
> We will add detailed results and analysis for Qwen2.5-VL in Appendix C as a new subsection: Additional Results on Alternative Backbones.
>
> ---
>
> > **_`Q2`: "Ablation experiments on RL without the format reward."_**
>
> A: Thanks for the suggestion. To assess the impact of the format reward during RL, we conducted additional experiments on Qwen2.5VL-3B and Qwen2VL-2B across three tasks, comparing RFT training with and without the format reward. The results are summarized below:
>
> |Method|T1-1D|T1-DSD|T1-DSM|T1-AVG|T2-ID|T2-DS|T2-AVG|T3-ID|T3-DSL|T3-DSR|T3-AVG|
> |:--|:--|:--|:--|:--|:--|:--|:--|:--|:--|:--|:--|
> |**3B RFT w/ format**|98.8|68.7|54.8|74.1|59.0|56.6|57.8|86.7|55.2|54.4|65.4|
> |**3B RFT w/o format**|97.4|67.9|54.4|73.2 **(–0.9)**|58.7|54.6|56.6 **(–1.2)**|84.4|54.6|54.0|64.3 **(–1.1)**|
>
> |Method|T1-1D|T1-DSD|T1-DSM|T1-AVG|T2-ID|T2-DS|T2-AVG|T3-ID|T3-DSL|T3-DSR|T3-AVG|
> |:--|:--|:--|:--|:--|:--|:--|:--|:--|:--|:--|:--|
> |**2B RFT w/ format**|96.8|60.0|28.4|61.7|49.0|33.1|41.1|74.6|64.0|64.1|67.6|
> |**2B RFT w/o format**|96.6|59.3|27.5|61.1 **(–0.6)**|48.2|32.1|40.2 **(–0.9)**|72.7|63.8|63.9|66.8 **(–0.8)**|
>
> We draw two main conclusions from these results:
>
> 1. **Performance drop without format reward**: Removing the format reward consistently reduced accuracy across all tasks and models, highlighting its importance in guiding RL training.
>
> 2. **Model size affects convergence**: For Qwen2.5VL-3B, convergence was stable without the format reward due to its strong instruction-following ability. However, Qwen2VL-2B showed a ≈200-step delay, indicating that smaller models rely more on explicit format supervision for structured prompts like `<think>`.
>
> We will include these additional results and discussions in a new subsection of Appendix C: *More Experiment Results on Format Reward*.
>
> ---
>
> > **_`Q3`: "More analysis of CoT."_**
>
> A: Thank you. In our pipeline, content within <think> represents the model’s CoT reasoning. To evaluate its improvement post-RL, we compare the Reason-RFT model (Stage 2) with the CoT-SFT baseline (Stage 1) on the Structure Perception task:
>
> **Qualitative Analysis**:
> 1. The post-RL model shows **stronger logical coherence**, with fewer reasoning breaks. For instance, Fig. 16 (case 2) highlights a missing calculation step in pre-RL output that is corrected after RL.
>
> 2. We observe **reflective reasoning patterns** (e.g., *"let me double check"*) emerging post-RL, absent in pre-RL outputs (e.g., Fig. 1).
>
> **Quantitative Analysis**:
> 1. **Step Count**: GPT-4o analysis on 100 samples shows post-RL responses have `2.7` more reasoning steps on average.
>
> 2. **Lexical Markers**: We categorized two types of expressions:
>    - *Prompting words*: *"oh I see"*, *"let me think step by step"*, *"let me double check"*
>    - *Logical connectives*: *"so"*, *"therefore"*, *"first"*, *"but"*, *"moreover"*
>    After RL, Use of prompting phrases and logical connectives increased by `14%` and `23%`, respectively.
> 3. **Answer Accuracy**: As shown in the Structure Perception subplot of Fig. 13, answer accuracy improves by `+20.6%` post-RL.
>
> These results confirm that RL substantially enhances the quality and depth of CoT reasoning.
>
> ---
> We hope our response addresses your concern and we look forward to further discussion with you.

---

> > ### Comment · Reviewer_yjNn · 2025-08-05
> >
> > Thank you for providing a detailed explanation of my questions and including the new results.
> >
> > * Due to a strange formatting issue with the review text parsing system, the "[object Object][object Object]" actually corresponds to the "\<think\>" and "\</think\>" tags in my review.
> > * I recommend that the authors incorporate these new details and results into the next version of the paper, as I believe they will offer additional value and insights to readers.
> > * Given that some of my concerns have been addressed, I would like to increase my score by one point.

---

> ### Author Response · Authors · 2025-08-05
>
> Dear Reviewer yjNn,
>
> Thanks again for your diligent effort in reviewing our submission. We have carefully addressed the concerns raised and conducted the requested experiments. As the discussion phase deadline is approaching, we sincerely hope you can consider positively recommending our work if your concerns are solved. If you still have further comments/suggestions, please don't hesitate to let us know.
>
> Best regards.

---

> ### Author Response · Authors · 2025-08-06
>
> Dear Reviewer yjNn,
>
> We're grateful for your thoughtful recommendation to incorporate the new results and explanations into the next version of the paper. We will make sure to include them to improve the clarity and value of our work for readers.
>
> We are also sincerely thankful for your willingness to adjust your score based on our response. Your constructive feedback has been invaluable in strengthening our submission.
>
> Best regards.

---

### Official Review · Reviewer_nCpk · 2025-06-24

**Clarity:** 2
**Significance:** 2
**Originality:** 2
**Rating:** 3
**Confidence:** 3

**Summary:**

This paper presents a two-stage framework for improving the visual reasoning capabilities of existing VLMs. Specifically, the authors propose to first fine-tune a model using SFT on a dataset with CoT reasoning traces, and then apply RL (i.e. GRPO) to further improve performance. The paper includes a systematic evaluation and analysis of the framework.

**Questions:**

How was the CoT data for the SFT stage generated? Was it generated using another VLM (i.e. GPT), was it human generated or a combination of both?

What sampling mechanism and resolution was used in Table 1 for the baselines and the proposed methods? Was it greedy decoding or sampling with temperature? Was there any difference?

How many runs were used to compute the numbers presented in table1? Can the authors also report std?

**Ethical Concerns:**

["NO or VERY MINOR ethics concerns only"]

**Final Justification:**

Please see the response I wrote to the authors.

**Limitations:**

yes

**Quality:**

2

**Strengths And Weaknesses:**

Strengths:

* Comprehensive evaluation across multiple visual reasoning tasks.

* Analysis of different training dynamics .

Weaknesses:

* While I appreciate the thorough evaluation across three visual reasoning categories, the paper overstates its main technical contribution. The use of SFT on CoT data followed by GRPO-based RL has already been explored in several reasoning works after  DeepSeek-R1. As a result, I believe the claim of Reason-RFT being a “novel training paradigm” in line 24 is a bit overstated.

* The separation between in-domain (ID) and domain-shift (DS) data feels somewhat artificial, especially considering that the base model may have encountered similar samples during pretraining. Moreover, in realistic scenarios, practitioners would likely perform SFT + GRPO over a broader, more diverse dataset rather than separate curated subsets.

* The paper’s presentation lacks clarity. I would recommend that the authors restructure it as an analysis paper , rather than presenting it as a  “novel training paradigm”.  For instance, the  paper should focus  more on how the SFT visual CoT traces were generated or how the reasoning quality influences downstream performance.   In fact, as it stands, it remains unclear how the SFT data was generated and most importantly whether  the SFT with CoT data is actually necessary. For instance, Table 1 shows that Reason-RFT-Zero achieves better average performance on 2 out of 3 tasks.

* While I appreciate the introduced benchmark, there are already a plethora of well-established visual reasoning benchmarks with public leaderboards. I would recommend additionally evaluating the proposed method on at least one of these to facilitate comparison with existing approaches and models.

---

> ### Author Rebuttal · Authors · 2025-07-31
>
> We sincerely thank the reviewer for the thoughtful and encouraging feedback. We will address your suggestions in detail below and incorporate the relevant improvements into the revised manuscript.
>
> ---
>
> > **_`W1 (Q1)`: "The novelty and contributons of our work."_**
>
> A: We would like to clarify that our work is not a simple combination of existing methods. Instead, we introduces **a novel and unified post-training methodology** specifically tailored to the core challenges of visual reasoning, such as **cognitive rigidity** and **generalization to domain shift** in dynamic visual reasoning.
>
> In contrast to previous works that either ***(i)*** focus exclusively on text-based reasoning (`[1]`); ***(ii)*** adopt only-SFT (`[2,3]`) or only-RL (`[4,5,6]`) strategies for post-training in isolation; or ***(iii)*** involve a few concurrent works (`[7,8]`) that combine SFT and RL but remain restricted to static visual tasks with narrowly defined reward types—our approach is **the first** to introduce a **generalizable training framework** that systematically integrates SFT and RFT in a **tightly coupled manner**, effectively addressing **both static** (e.g., structural perception) and **dynamic** (e.g., temporal counting, spatial transformation) visual reasoning under a unified reward formulation.
>
> The novelty of our approach lies in the following aspects:
>
> - **An adaptive training sample allocation strategy** that dynamically balances intermediate reasoning supervision with reward-guided exploration. This strategy enables more effective coupling between the SFT and RFT stages, supporting a unified learning process that aligns model behavior. Unlike prior methods that treat SFT and RL as disjoint stages, our design ensures their interaction is coherent and purpose-driven.
>
> - **A unified visual reasoning reward formulation** that generalizes across diverse task formats. We introduce a principled reward design that captures reasoning correctness and completeness in a task-agnostic manner:
>   - **Structured-format reward** (including summary, caption, thinking and answer tags) to enhance the quality and coherence of reasoning chains.
>   - **Discrete-value reward** for tasks with discrete-value outputs (e.g., classification and counting).
>   - **Tolerance-based reward** for tasks requiring continuous-value outputs (e.g., geometric estimation, visual grounding).
>   - **Function-matching reward** for tasks with sequence-like outputs (e.g., action planning or tool calling).
>
>   These reward formulations are designed to be task-agnostic and applicable, moving **BEYOND** handcrafted, task-specific designs commonly used in prior work.
>
> - **A comprehensive visual reasoning evaluation protocol**, covering both in-domain and domain-shifted scenarios, and spanning synthetic and real-world visual environments. The evaluation rigorously benchmarks multiple dimensions of reasoning ability—including dynamic reasoning, geometric perception, and mathematical arithmetic—enabling us to assess generalization and robustness in a systematic manner.
>
> > [1] *DeepSeek-R1*, ArXiv'25.
>
> > [2] *LLaVA-CoT*, ArXiv'24.
>
> > [3] *LlamaV-o1*, ACL'25.
>
> > [4] *Visual-RFT*, ICCV'25.
>
> > [5] *VLM-R1*, ArXiv'25.
>
> > [6] *R1-Zero’s “Aha Moment”*, ArXiv'25.
>
> > [7] *R1-Onevision*, ICCV'25.
>
> > [8] *Vision-R1*, ArXiv'25.
>
> ---
>
> > **_`W2`: "Query about the separation between ID and DS data."_**
>
> A: We clarify that the ID/DS split in our benchmark is **not artificial**, but based on a principled distinction: whether object categories, scene compositions, and camera viewpoints were seen during training.
>
> - **ID data**: aligns with training distributions in terms of object identity, layout, and viewpoint.
> - **DS data**: introduces systematic shifts—e.g., novel 3D assets, unseen viewpoints, or altered scene configurations.
>
> This separation reflects **practical generalization challenges** in embodied visual reasoning, where agents often face unfamiliar and dynamic environments. For example, in our Visual Transformation benchmark, the DS set simulates real-world deployment where agents encounter new perspectives or object arrangements.
>
> The necessity of this split is further **empirically validated**. For example, in the Visual Counting zero-shot setting, Qwen2-VL-2B's accuracy drops sharply from ID (82.40) to DS-D (42.67) to DS-M (0.00), highlighting the difficulty and importance of DS evaluation in diagnosing robustness and transferability.
>
> ---
>
> > **_`W3-1 (Q1)`: "CoT data construction."_**
>
> A: The CoT data is built through a combination of automated generation and manual verification to ensure both **diversity** and **correctness**:
>
> - **Diversity**: We use reasoning-guided prompts (e.g., “Let’s think step by step”) with **GPT-4o** and **Gemini-Pro**, applying temperature sampling (`temperature=0.7`, `top-k=50`, `top-p=0.9`) and adding hand-crafted exemplars to enhance variety and depth.
>
> - **Correctness**: Outputs are filtered through **manual review** and **automated scripts**. Low-quality samples (e.g., incomplete reasoning or incorrect answers) are removed. For each task subtype, `10%` of the samples are randomly selected for human validation and refinement if needed.
>
> ---
>
> > **_`W3-2`: "Impact of Reason-RFT on downstream reasoning quality."_**
>
> A: We compare the Qwen2.5VL-3B model trained with Reason-RFT (Stage 2, with RL) and the one trained only with CoT-SFT (Stage 1, without RL) on the **Structure Perception** task. Key findings are as follows:
>
> **Qualitative Analysis:**
> 1. The RL-trained model shows stronger logical coherence and fewer broken reasoning chains.
>    - E.g., in Fig. 16 (case 2), the pre-RL model omits a crucial “×2” step, which is rare post-RL.
> 2. Post-RL responses include reflective phrases (e.g., *“let me double check”*) not seen in pre-RL outputs (Fig. 1).
>
> **Quantitative Analysis:**
> 1. **Reasoning steps**: On 100 samples, the RL-trained model generates `+2.7` more steps on average (per GPT-4o analysis).
> 2. **Lexical patterns**: Post-RL responses include `+14%` more prompting cues and `+23%` more logical connectives.
> 3. **Answer accuracy**: As shown in Fig. 13 (Appendix), final accuracy improves by `+20.6%` after RL.
>
> Overall, Reason-RFT significantly enhances reasoning fluency, structure, and downstream performance—not just correctness.
>
> ---
>
> > **_`W3-3`: "Is CoT-SFT necessary, given Reason-RFT-Zero's strong performance?"_**
>
> A: While CoT-SFT brings the largest gains on structurally formatted tasks like Visual Transformation, its benefits are broader and foundational. It supports Reason-RFT in the following ways:
>
> | |VC(DS-H)||ST(ID+DS)||
> |-|--:|--:|--:|--:|
> | |2B|7B|2B|7B|
> |Zero-Shot|0.00|4.80|4.35|13.01|
> |only-RL|5.20|21.20|33.74|56.68|
> |Reason-RFT|**28.40**|**35.60**|**67.58**|**65.98**|
>
> - **Format Alignment**: CoT-SFT teaches structured outputs (e.g., spatial transformation), reducing RL exploration cost and improving convergence.
>
> - **Knowledge Injection**: In domain-shifted tasks like visual counting, CoT-SFT boosts generalization. Compared to RL-only, Reason-RFT improves by `+23.2` (2B) and `+14.4` (7B) on VC (DS-H).
>
> - **Sample Efficiency**: CoT-SFT acts as a policy prior. As shown in Fig. 3, Reason-RFT reaches 70% of Reason-RFT-Zero’s performance using only 3% of its training data.
>
> In short, CoT-SFT is not just a warm-up—it is **crucial** for format learning, generalization, and efficient RL adaptation.
>
> ---
>
> > **_`Q1`: "CoT data construction."_**
>
> A: Plase refer to `W3-1`.
>
> ---
>
> > **_`Q2`: "Decoding strategy and temperature settings."_**
>
> A: We used **greedy decoding** (`temperature = 0.0`) for all methods reported in the main text. To evaluate robustness under different decoding strategies, we further tested `temperature = 0.5` and `0.7` using Qwen2.5-VL-3B-Instruct on three representative tasks.
>
> |Setting|Greedy|T=0.5|T=0.7|AVG|Greedy|T=0.5|T=0.7|AVG|Greedy|T=0.5|T=0.7|AVG|
> |--|--|--|--|--|--|--|--|--|--|--|--|--|
> |      |TASK1||||TASK2||||TASK3|||||
> |ZERO-SHOT|43.7|40.4|40.7|41.6|37.1|36.1|36.8|36.7|8.4|8.3|9.0|8.6|
> |ANS-SFT|51.6|48.7|48.2|49.5|42.4|40.4|40.1|41.0|61.6|61.0|60.8|61.1|
> |COT-SFT|66.0|62.9|60.6|63.2|52.7|51.4|51.6|51.9|57.3|56.5|56.1|56.6|
> |RFT-ZERO|56.2|53.2|52.9|54.1|54.6|54.1|54.3|54.4|55.3|55.2|55.4|55.3|
> |RFT|**74.1**|**70.1**|**68.8**|**71.0**|**57.8**|**56.8**|**56.1**|**56.9**|**65.4**|**64.7**|**64.3**|**64.8**|
>
>
> Overall, **greedy decoding yields the best absolute scores** due to its determinism. However, **relative performance rankings remain consistent** across temperatures, and Reason-RFT consistently outperforms all baselines.
>
> These results confirm that our method is **robust and effective under varying decoding strategies**, and the main conclusions of our paper remain unchanged.
>
> ---
>
> > **_`Q3`: "Report standard deviation and clarify number of runs."_**
>
> A: Thank you for your question. To ensure reproducibility, we ran each method **three times** with different seeds on **Qwen2VL-2B**. Results (mean ± std) are shown below:
>
> |Method|T1-ID|DSD|DSM|AVG|T2-ID|DS|AVG|T3-ID|DSL|DSR|AVG|
> |:--|:--|:--|:--|:--|:--|:--|:--|:--|:--|:--|:--|
> |ANS-SFT|95.9 ±0.3|50.3 ±1.2|5.6 ±0.4|50.6 ±0.6|50.6 ±1.0|23.0 ±0.8|36.8 ±0.9|77.6 ±0.5|48.8 ±1.1|49.2 ±1.7|58.5 ±1.1|
> |CoT-SFT|85.2 ±0.8|49.6 ±1.2|35.2 ±1.8|56.7 ±1.3|43.8 ±0.9|24.9 ±0.7|34.4 ±0.8|64.4 ±0.1|43.5 ±0.6|43.1 ±0.4|50.4 ±0.4|
> |Reason-RFT-Zero|98.1 ±0.6|58.2 ±1.1|6.3 ±1.0|54.2 ±0.9|47.5 ±0.2|32.8 ±0.5|40.1 ±0.4|42.6 ±1.0|33.8 ±1.1|32.0 ±1.6|36.1 ±1.3|
> |Reason-RFT|96.4 ±0.6|58.6 ±1.2|31.7 ±2.9|62.3 ±1.6|49.5 ±0.5|34.1 ±1.3|41.8 ±0.9|74.4 ±0.4|64.0 ±1.4|64.0 ±0.1|67.5 ±0.6|
>
> Across the 12 metrics above, all standard deviations are under **2.0**, with most under **1.0**, indicating high stability. Due to space constraints, we omit the Qwen2VL-7B results here but will include them in the revised version’s appendix.
>
> ---
> We hope our response addresses your concern and we look forward to further discussion with you.

---

> ### Author Response · Authors · 2025-08-05
>
> Dear Reviewer nCpk,
>
> Thanks again for your diligent effort in reviewing our submission. We have carefully addressed the concerns raised and conducted the requested experiments. As the discussion phase deadline is approaching, we sincerely hope you can consider positively recommending our work if your concerns are solved. If you still have further comments/suggestions, please don't hesitate to let us know.
>
> Best regards.

---

> > ### Comment · Reviewer_nCpk · 2025-08-05
> >
> > I thank the authors for the detailed response and the effort put into the rebuttal. Some of my concerns has been resolved, so I have increased my score by a point,  however the following remains.
> >
> > While I appreciate the listed contributions, I remain of the view that, in its current presentation, SFT (particularly with CoT) and RL does not appear fundamentally new. Several other reviewers express similar views. In particular, the abstract (line 24) characterizes Reason-RFT as a “novel training paradigm,” which may overstate the contribution. I recommend revising the paper for clarity so the claimed “key novelty” mentioned in the rebuttal is explicitly and proportionately presented.
> >
> > Regarding the separation between ID and DS data, it is difficult to disentangle these for a model  trained on Internet-scale data. For such models,  I remain of the view that, it’s hard to define what counts as novel 3D assets, unseen viewpoints, or altered scene configurations in an internet-scale pretrained/postrained model.
> >
> > Regarding CoT data construction, this should be clearly described in the main paper so reviewers can properly evaluate it, and compare to existing methods in the literature. In my opinion, this could be a key novelty of this paper. Other reviewers have also asked for this, highlighting its importance. And in the rebuttal, the authors mention that SFT quoting "it is crucial for format learning, generalization, and efficient RL adaptation." I believe this part is essential.
> >
> > Overall,  I think the paper will benefit from a major rewrite, the contribution should be tuned down or properly introduced (it is fine if it builds on top of existing work but it should be introduced as such),  and the presentation could be also be improved as suggested in the review.

---

> > > ### Author Response · Authors · 2025-08-06
> > >
> > > Dear Reviewer nCpk,
> > >
> > > Thank you for your thoughtful and constructive feedback. We are encouraged that some of your concerns have been addressed and appreciate your continued engagement with our work.
> > >
> > > ---
> > >
> > > > **_`(1)` Clarifying the Positioning of Our Contribution_**
> > >
> > > We acknowledge your point regarding the use of the term *“novel training paradigm”* in the abstract. In the revised version, we will revise this phrasing to more accurately reflect our work as a **unified post-training methodology** that integrates Supervised Fine-Tuning (SFT) and Reinforcement Fine-Tuning (RFT) in a closely coupled manner to handle challenges in both static and dynamic visual reasoning tasks.
> > >
> > > We will make the following adjustments in the revised paper, as mentioned in the rebuttal:
> > >
> > > - **Give in-depth analysis on the adaptive sample allocation mechanism** that enables tighter coupling between SFT and RFT stages.
> > > - **Explicitly highlight the role of structured reasoning supervision**—particularly through CoT-SFT—in aligning reasoning formats and facilitating generalization and RL adaptation (further discussed in Point 2).
> > > - **Present our unified visual reasoning reward formulation** more clearly to demonstrate its applicability across diverse visual reasoning task types.
> > >
> > > ---
> > >
> > > > **_`(2)` Detailing the Construction and Role of CoT-SFT_**
> > >
> > > We appreciate your emphasis on the importance of clearly describing the CoT-SFT construction process, and we agree that it represents a potentially key contribution of our work.
> > >
> > > In the revised manuscript, we will:
> > >
> > > - **Provide a detailed description of the CoT-SFT data generation pipeline**, including prompt design, annotation guidelines, and filtering criteria. This will be added to the main text and further elaborated in the appendix.
> > > - **Explain how CoT-SFT facilitates reasoning format alignment** and supports efficient RL adaptation by providing structured supervision that reduces the exploration burden during RFT.
> > > - **Clarify the comparative advantages of our approach** (e.g., generalization and sample efficiency), relative to existing methods in the literature.
> > >
> > > We will also explicitly mark this as an additional major contribution in the introduction and discussion sections, in line with your feedback.
> > >
> > > ---
> > >
> > > > **_`(3)` On the ID vs. DS Generalization Split_**
> > >
> > > We understand your concern regarding the difficulty of disentangling in-domain (ID) and domain-shifted (DS) data for Internet-scale pretrained models. While we acknowledge these challenges, our ID/DS split is constructed based on **clearly defined distinctions** in object categories, scene compositions, and camera viewpoints—whether they were **seen or unseen during our post-training process**.
> > >
> > > Specifically:
> > > - In **Task 1** and **Task 3**, all test images in the DS sets are reconstructed using newly synthesized 3D assets and previously unseen camera viewpoints, **rendered through our controlled Blender-based pipeline**, explicitly designed to prevent data leakage.
> > > - While not immune to the ambiguity inherent in large-scale pretraining, this setup provides a **meaningful approximation of real-world deployment shifts**.
> > >
> > > Empirically, this separation reveals consistent and significant performance drops (e.g., DS-M accuracy drops to 0%), highlighting its utility in evaluating generalization robustness. We will further clarify this setup and its motivation in the revised version to address potential ambiguity.
> > >
> > > ---
> > >
> > > We thank you again for your thoughtful suggestions and will incorporate all these improvements into the final version to **ensure that the contributions are clearly, proportionately, and rigorously presented**.
> > >
> > > Best regards.

---

### Official Review · Reviewer_krtY · 2025-06-29

**Clarity:** 4
**Significance:** 3
**Originality:** 2
**Rating:** 5
**Confidence:** 4

**Summary:**

The authors proposed a two-stage post-training strategy for vision-language models. The post-training framework comprises two stages: 1. SFT with high-quality CoT data, 2. RLVR with GRPO. The paper provides a comprehensive evaluation of various post-training strategies on three visual reasoning tasks including visual counting, structural perception, and spatial transformation. The experimental study analyses the roles of CoT-SFT and GRPO and demonstrate that the proposed two-stage post-training improves the model performance consistently.

**Questions:**

1. Why both ANS-SFT and CoT-SFT leads to worse results than the zero-shot performance of Qwen2VL-7b in visual counting (ID and DS-D)?
2. In spatial transformation, why ANS-SFT shows very strong results and pure RL gives the worst performance?
3. Related to the above two questions: it seems that some tasks benefit more from SFT and some of them benefit more from RL. Do you have more analysis with other visual reasoning tasks and any insights how are the visual tasks coupled with the post-training strategies?
4. How much does the ratio of data size for SFT and RL affect the performance?

**Ethical Concerns:**

["NO or VERY MINOR ethics concerns only"]

**Final Justification:**

I'd like to thank the authors for providing response to my questions. My concerns have been well addressed. I keep my rating unchanged.

**Limitations:**

Yes

**Paper Formatting Concerns:**

Found no major formatting issues

**Quality:**

3

**Strengths And Weaknesses:**

Strengths:

1. a solid work that studies the roles of SFT and GRPO in the post-training of vision-language models
2. the designed evaluation tasks/datasets are interesting and valuable for the future research
3. the experiments provide insightful analysis and findings

Weaknesses:

1. the choices of base models are limited to QwenVL family. The analysis would be more comprehensive and impactful if the authors can provide the analysis with other base models when considering the interesting findings with Qwen-series models and RLVR in [1]

[1] Reinforcement learning with random rewards actually works with Qwen 2.5, https://www.interconnects.ai/p/reinforcement-learning-with-random?utm_campaign=post&utm_medium=web

---

> ### Author Rebuttal · Authors · 2025-07-31
>
> We sincerely thank the reviewer for the constructive and encouraging comments. We are grateful for your recognition of our work as a solid study on the roles of SFT and GRPO in VLM post-training, the value of our proposed evaluation tasks and datasets for future research, and the insightful experimental analysis. Your feedback is highly appreciated, and we address your specific concerns in detail below. We will carefully reflect your suggestions in the revised manuscript.
>
> ---
>
> > **_`W1`: "Limited base models. Can the authors provide analysis with other base models beyond Qwen-series?"_**
>
> A: Thanks for your suggestion. To validate the effectiveness of our proposed method, we conducted additional experiments on **InternVL3-2B-Instruct** using the Visual Counting task. The results are shown below:
>
> | Model        | Setting           | T1-1D | T1-DSD | T1-DSM | AVG    |
> |--------------|-------------------|-------|--------|--------|--------|
> | internvl3-2b | ZERO              | 79.30 | 51.20  | 5.10   | 45.20  |
> | internvl3-2b | ANS               | 96.80 | 52.00  | 6.50   | 51.77  |
> | internvl3-2b | COT               | 88.90 | 60.10  | 50.20  | 66.40  |
> | internvl3-2b | Reason-RFT-ZERO   | 98.90 | 59.40  | 12.30  | 56.87  |
> | internvl3-2b | Reason-RFT        | **99.10** | **69.80** | **55.90** | **74.93** |
>
> The results demonstrate that Reason-RFT consistently enhances performance on the InternVL backbone. Key observations:
>
> - **(1) Substantial Gains in Visual Reasoning:**
>   Compared to ANS, COT, and Reason-RFT-ZERO, our Reason-RFT approach leads to notable improvements:
>   - InternVL3-2B:
>     - +23.16 over ANS
>     - +8.53 over COT
>     - +18.06 over Reason-RFT-ZERO
>
> - **(2) Robustness under Domain Shift (T1-DSM):**
>   The largest gains occur in domain-shifted settings:
>   - InternVL3-2B: +43.60 over Reason-RFT-ZERO on T1-DSM
>
> - **(3) Data Efficiency:**
>   Similar to Qwen2VL, Reason-RFT shows high data efficiency on InternVL3:
>   achieving **~70% of the final performance of Reason-RFT-ZERO using only <5% of the data**.
>
> We will include the full results and analysis for InternVL in the revised version, under **Appendix C: More Experimental Results on Different Backbones**.
>
> Additionally, based on your reference to the interesting findings in Qwen-series models and RLVR in [1], we further investigated the effect of random reward in the RL training stage. We conducted experiments using both **Qwen2.5VL-3B** and **InternVL3-2B**, applying discrete random rewards (0/1) and continuous random values in [0,1]. Results are as follows:
>
> **Qwen2.5VL-3B:**
>
> | Setting               | Task1  | Task2  | Task3  |
> |-----------------------|--------|--------|--------|
> | zero-shot             | 43.74  | 37.10  | 8.38   |
> | original reward       | 56.42 (+12.68) | 54.63 (+17.53) | 55.33 (+46.95) |
> | random reward (0/1)   | 36.29 (-7.45)  | 32.36 (-4.74)  | 6.57 (-1.81)   |
> | random reward (0~1)   | 34.07 (-9.67)  | 30.32 (-6.78)  | 8.32 (-0.06)   |
>
> **InternVL3-2B:**
> | Setting               | Task1  | Task2  | Task3  |
> |-----------------------|--------|--------|--------|
> | zero-shot             | 45.20  | 30.85  | 7.90   |
> | original reward       | 58.75 (+13.55) | 53.92 (+23.07) | 54.68 (+46.78) |
> | random reward (0/1)   | 37.48 (-7.72)  | 31.66 (+0.81)  | 6.42 (-1.48)   |
> | random reward (0~1)   | 33.95 (-11.25) | 29.84 (-1.01)  | 7.82 (-0.08)   |
>
> Based on these results, we conclude that random reward schemes do not enhance model performance. In fact, they degrade the accuracy for both Qwen2.5VL and InternVL3. For example, with random reward (0~1), performance drops by **-9.67** and **-11.25** on Task1 respectively.
>
> ---
>
> > **_`Q1`: "Why does SFT lead to worse results than the zero-shot performance of Qwen2VL-7B in visual counting (ID and DS-D)?"_**
>
> A: Thanks for your question—this is indeed an interesting phenomenon. SFT tends to guide models toward memorizing domain-specific response patterns, as demonstrated in [1]. For strong base models like Qwen2VL-7B, which already perform well on visual counting tasks in the zero-shot setting (`98.60`), further SFT may not enhance reasoning ability but instead induce *cognitive rigidity*, where the model aligns too rigidly with training examples, leading to diminished generalization capability on unseen test cases. This rigidity can degrade performance on ID and DS-D test sets.
>
> This also explains why Reason-RFT employs only a small number of samples (1.6k) during the Stage 1 CoT-SFT phase—sufficient to inject new task knowledge and activate novel reasoning patterns without cognitive rigidity.
>
> > [1] *SFT Memorizes, RL Generalizes: A Comparative Study of Foundation Model Post-training*, ICML'25.
>
> ---
>
> > **_`Q2`: "Why does ANS-SFT outperform, while pure RL underperforms on spatial transformation?"_**
>
> A: Thanks for your question. Spatial transformation requires specific spatial reasoning and output patterns. When the base model lacks this ability (as evidenced by the low zero-shot scores of the 2B and 7B models—`4.35` and `13.1`, respectively), pure RL tends to explore ineffective regions of the reasoning space and output formats during the early stages of training. Additionally, due to the lack of specialized knowledge in pretraining, pure RL struggles to overcome the knowledge barrier—especially for smaller models like 2B—resulting in poor learning signals and ineffective reasoning. In contrast, ANS-SFT provides direct supervision to quickly learn the desired answering format, achieving readily achievable performance. Therefore, a two-stage SFT+RL (Reason-RFT) process is necessary to ensure stronger and more stable performance.
>
> ---
>
> > **_`Q3`: "How are different visual reasoning tasks coupled with SFT and RL strategies?"_**
>
> A: Thank you for the insightful follow-up. Indeed, we observe a clear task-specific coupling between visual reasoning tasks and post-training strategies:
> - **SFT** is more effective for tasks where format learning or domain-specific answer templates dominate performance—such as spatial transformation or counting tasks with structured outputs. In these cases, direct supervision helps the model quickly acquire the correct response pattern, especially when the base model lacks such capabilities in the zero-shot setting.
> - **RL,** on the other hand, is more beneficial for tasks requiring multi-step reasoning—such as latent action reasoning or transformation sequence planning—where the model already shows basic task understanding. In such cases, RL fine-tuning helps reinforce coherent reasoning trajectories. However, for tasks the model has not yet grasped, pure RL often fails to provide effective learning signals, limiting its usefulness when used alone.
>
> While **neither SFT nor RL alone is sufficient**, our Reason-RFT framework demonstrates that carefully coupling the two—via CoT-based cognitive activation and reward-guided refinement—yields robust improvements. We also find that tuning the data ratio and reward design plays a critical role in maintaining adaptability without overfitting. Additional task-wise analyses and ablations will be incorporated in the appendix of the revised version.
>
> ---
>
> > **_`Q4`: "How much does the ratio of data size for SFT and RL affect the performance?"_**
>
> A: Thanks for your question. We found that for visual reasoning tasks, the optimal range of CoT samples used for SFT activation lies between `0.8k` and `2k`. Increasing the number of CoT samples beyond 2k brings marginal gains. Interestingly, when the proportion of CoT samples exceeds approximately `40%` of the RL training data, the model performance begins to decline. Once it surpasses `80%`, the performance tends to degenerate toward that of CoT-SFT alone (i.e., cognitive rigidity).
>
> This is a particularly interesting phenomenon. We will include a performance trend plot and further discussion on this effect in the appendix of the revised version.
>
>
> ---
> We hope our response addresses your concern and we look forward to further discussion with you.

---

> ### Author Response · Authors · 2025-08-05
>
> Dear Reviewer krtY,
>
> Thanks again for your diligent effort in reviewing our submission. We have carefully addressed the concerns raised and conducted the requested experiments. As the discussion phase deadline approaches, we sincerely hope that you will consider maintaining a positive recommendation for our work if you find your concerns sufficiently resolved. If you still have further comments/suggestions, please don't hesitate to let us know.
>
> Best regards.

---

> > ### Comment · Reviewer_krtY · 2025-08-05
> >
> > Thank you for adding the new results and addressing my questions. The in-depth analysis helps me better understanding the training process. I recommend incorporating these findings into the final manuscript, and I maintain a positive recommendation for this work.

---

### Official Review · Reviewer_rmS6 · 2025-07-02

**Clarity:** 3
**Significance:** 3
**Originality:** 2
**Rating:** 4
**Confidence:** 3

**Summary:**

This paper studies the reasoning ability of vision-language models (VLMs) by training them to generate chain-of-thought (CoT) reasoning through supervised learning and reinforcement learning. The authors propose a framework called Reason-RFT, which uses a two-stage training process on vision tasks such as counting, structural perception, and spatial transformation.
In the first stage (supervised fine-tuning, or SFT), they create <image, question, chain-of-thought, answer> annotations and train the VLM to produce token-level predictions, meaning it generates both the CoT and the final answer given an image and question.
In the second stage, they apply reinforcement learning with a GRPO loss. This involves generating multiple CoT traces, assigning each a reward score, and using these rewards to further refine the model’s parameters.
Experiments on tasks such as counting, perception, and spatial transformation demonstrate that combining supervised learning and reinforcement learning yields improvements and enhanced generalization.

**Questions:**

1. How is the initial chain-of-thought (CoT) data for the first-stage supervised fine-tuning (SFT) generated? How do you ensure the reliability and correctness of these first-round CoT annotations?
2. After reinforcement learning, I would like to see more analysis of the quality of the CoT. Is there any quantitative comparison of CoT outputs before and after GRPO tuning?

**Ethical Concerns:**

["NO or VERY MINOR ethics concerns only"]

**Final Justification:**

I have read the author's response, and they have addressed my concern, I would keep my original ratings.

**Limitations:**

Yes, the author discusses limitations in the appendix.

**Paper Formatting Concerns:**

The paper follows the NeurIPS guide.

**Quality:**

3

**Strengths And Weaknesses:**

Strengths:

1. This paper investigates how to enhance the reasoning process (chain-of-thought, or CoT) through a two-stage approach that combines supervised learning and reinforcement learning. In the first stage, supervised learning is achieved by instruction tuning the VLM. In the second stage, reinforcement learning is applied to the VLM using the GRPO loss.

2. Experiments on VQA tasks covering counting, perception, and spatial reasoning show that the two-stage training pipeline improves performance compared to existing models and offers greater transparency.

Weakness:

1. The two-stage training pipeline is quite similar to the training approach used in DeepSeek-R1. It is essentially an extension of training VLMs in the same manner as LLMs. The main differences lie in the chain-of-thought (CoT) generation in the first stage and the new reward function for VQA tasks, which limits its theoretical contribution.

2. The process for generating new CoT annotations used in the first-stage training (Section 3.1) is not explicitly described. Explaining how the initial CoT data for SFT tuning is created is important, as it affects the ability to reproduce the work.

3. Do the open-source VLM models in Table 1 follow the same zero-shot setting? To ensure a fair comparison, these open-source VLMs should be trained using the same dataset.

4. The Reason-RFT does not generalize well to more generic VQA tasks, such as MMMU, RealWorldQA, and AI2D, as shown in Table 6, across different backbones. The improvements are unstable.

5. Relevant works line NMN [1], IPRM [2] are missing in discussing neuro-symoblic methods (line 79)

[1] Neural Module Networks，CVPR2016

[2] Learning to Reason Iteratively and Parallelly for Complex Visual Reasoning Scenarios, NeurIPS 2024

---

> ### Author Rebuttal · Authors · 2025-07-31
>
> We sincerely thank the reviewer for the thoughtful and encouraging feedback. We are especially grateful for the recognition of our **two-stage training framework**, as well as the **improved performance and transparency on visual reasoning**. We deeply value your comments and suggestions, and address them in detail below. We will carefully incorporate the relevant clarifications and improvements into the revised manuscript.
>
> ---
>
> > **_`W1 (Q1)`: "The novelty of our work."_**
>
> A: Thank you for raising this point. We would like to clarify that our work is not a direct extension or simple combination of existing methods. Instead, it introduces **a novel and unified post-training methodology** specifically tailored to the core challenges of visual reasoning, such as **cognitive rigidity** and **generalization to domain shift** in dynamic visual reasoning.
>
> In contrast to previous works that either ***(i)*** focus exclusively on text-based reasoning (`[1]`); ***(ii)*** adopt only-SFT (`[2,3]`) or only-RL (`[4,5,6]`) strategies for post-training in isolation; or ***(iii)*** involve a few concurrent works (`[7,8]`) that combine SFT and RL but remain restricted to static visual tasks with narrowly defined reward types—our approach is **the first** to introduce a **generalizable training framework** that systematically integrates supervised fine-tuning (SFT) and reward-based fine-tuning (RFT) in a **tightly coupled manner**, effectively addressing **both static** (e.g., structural perception) and **dynamic** (e.g., temporal visual counting, spatial transformation) visual reasoning under a unified reward formulation.
>
> The novelty of our approach lies in the following aspects:
>
> - **An adaptive training sample allocation strategy** that dynamically balances intermediate reasoning supervision with reward-guided exploration. This strategy enables more effective coupling between the SFT and RFT stages, supporting a unified learning process that aligns model behavior with visual reasoning objectives. Unlike prior methods that treat SFT and RL as disjoint stages, our design ensures their interaction is coherent and purpose-driven.
>
> - **A unified visual reasoning reward formulation** that generalizes across diverse task formats. Beyond replicating standard reward templates, we introduce a principled reward design that captures reasoning correctness and completeness in a task-agnostic manner:
>   - **Structured-format reward** (including summary, caption, thinking and answer tags) to enhance the quality and coherence of reasoning chains.
>   - **Discrete-value reward** for tasks with discrete-value outputs (e.g., classification and counting).
>   - **Tolerance-based reward** for tasks requiring continuous-value outputs (e.g., geometric estimation, visual grounding).
>   - **Function-matching reward** for tasks with sequence-like outputs (e.g., action planning or tool calling).
>
>   These reward formulations are designed to be task-agnostic and applicable, moving **BEYOND** handcrafted, task-specific designs commonly used in prior work.
>
> - **A comprehensive visual reasoning evaluation protocol**, covering both in-domain and domain-shifted scenarios, and spanning synthetic and real-world visual environments. The evaluation rigorously benchmarks multiple dimensions of reasoning ability—including dynamic reasoning, geometric perception, and mathematical arithmetic—enabling us to assess generalization and robustness in a systematic manner.
>
> > [1] *DeepSeek-R1: Incentivizing Reasoning Capability in LLMs via Reinforcement Learning*, ArXiv'25.
>
> > [2] *LLaVA-CoT: Let Vision Language Models Reason Step-by-Step*, ArXiv'24.
>
> > [3] *LlamaV-o1: Rethinking Step-By-Step Visual Reasoning in LLMs*, ACL'25.
>
> > [4] *Visual-RFT: Visual Reinforcement Fine-Tuning*, ICCV'25.
>
> > [5] *VLM-R1: A Stable and Generalizable R1-style Large Vision-Language Model*, ArXiv'25.
>
> > [6] *R1-Zero’s “Aha Moment” in Visual Reasoning on a 2B Non-SFT Model*, ArXiv'25.
>
> > [7] *R1-Onevision: Advancing Generalized Multimodal Reasoning through Cross-Modal Formalization*, ICCV'25.
>
> > [8] *Vision-R1: Incentivizing Reasoning Capability in Multimodal Large Language Models*, ArXiv'25.
>
> ---
>
> > **_`W2`: "CoT data construction."_**
>
> A: Thanks for your question. The CoT data is constructed through a combination of automated generation and manual verification, with both **diversity** and **correctness** rigorously ensured. Specifically:
>
> - **Diversity assurance**: We begin by leveraging a set of carefully designed reasoning-guided templates (e.g., "Let's think step by step", "Break down your reasoning", etc.) to prompt automatic generation using models like **GPT-4o** and **Gemini-Pro**. To increase stylistic diversity, we apply temperature sampling (`temperature=0.7`, `top-k=50`, `top-p=0.9`) and include manually crafted exemplars as few-shot demonstrations to enhance generation quality and depth.
>
> - **Correctness assurance**: All generated outputs undergo filtering via both manual review and automated scripts. Low-quality samples—such as overly brief reasoning steps or answer inconsistencies with ground truth—are discarded. For each task subtype, we randomly sample `10%` of the data for human validation. Subtypes with deficiencies are further corrected and supplemented, yielding a reliable and comprehensive dataset.
>
> We will include a detailed description of the CoT data construction process in the appendix of the revised version.
>
> ---
>
> > **_`W3`: "Zero-shot setting consistency in Table 1 models."_**
>
> A: Thanks for your suggestion. The open-source models reported in Table 1 are evaluated under the same zero-shot setting to ensure fair comparison. This includes identical evaluation datasets, question prompts, and generation parameters (e.g., `temperature=0.0` for greedy sampling).
>
> ---
>
> > **_`W4`: "Generalization performance on generic benchmarks."_**
>
> A: We thank the reviewer for pointing out the generalization performance in Table 6. Our primary goal is **NOT** to achieve the best results on general VQA benchmarks, but to explore an important challenge: **how to enhance domain-specific complex reasoning without severely degrading other capabilities.**
>
> |Method|MMMU|RealWorldQA|MathVision|AI2D|
> |------|----|------------|----------|----|
> |Zero-Shot (7B)|54.10|67.19|16.30|83.00|
> |+ANS-SFT|42.66 (-11.44)|48.10 (-19.09)|9.12 (-7.18)|78.30 (-4.70)|
> |+CoT-SFT|44.67 (-9.43)|36.46 (-30.73)|15.30 (-1.00)|73.25 (-9.75)|
> |+Reason-RFT-Zero|46.44 (-7.66)|45.10 (-22.09)|10.86 (-5.44)|75.28 (-7.72)|
> |+Reason-RFT|50.04 **(-4.06)** |61.31 **(-5.88)**|17.60 **(+1.30)**|81.70 **(-1.30)**|
>
> As shown in the table above, taking Qwen2VL-7B as example, while Reason-RFT does not significantly improve general VQA scores, it maintains a **more stable general performance** compared to other training paradigms (e.g., ANS-SFT, CoT-SFT, Reason-RFT-Zero), which often cause larger performance drops. For example, on RealWorldQA benchmark, ANS-SFT leads to a `-19.09` drop, CoT-SFT `-30.73`, and Reason-RFT-Zero `-22.09`, while Reason-RFT reduces the gap to only `-5.88`, showing a much more stable performance behavior. Moreover, in some cases, Reason-RFT even brings slight gains (e.g., on MathVision, `+1.30`). This indicates its potential in **balancing multiple abilities during training**.
>
> We believe this is a valuable insight for future work on developing more balanced and robust multimodal models. Further clarification will be included in the appendix of the revised version.
>
> ---
>
> > **_`W5`: "Add missing citations to prior neuro-symbolic methods."_**
>
> A: Thank you for your suggestion. We will include the missing citations (e.g., NMN `[9]`, IPRM `[10]`) in the revised version under the neuro-symbolic methods section of the related work.
>
> > [9] *Neural Module Networks*，CVPR'16.
>
> > [10] *Learning to Reason Iteratively and Parallelly for Complex Visual Reasoning Scenarios*, NeurIPS'24.
>
> ---
>
> > **_`Q1`: "CoT data construction."_**
>
> A: Please refer to `W2`.
>
> ---
>
> > **_`Q2`: "Comparison of CoT quality before and after RL."_**
>
> A: Thank you for your question. To assess the improvement in CoT quality after RL, we compared the reasoning outputs of the Qwen2VL-3B model trained with Reason-RFT (Stage 2, post-RL) and the one trained with only CoT-SFT (Stage 1, pre-RL) on the Structure Perception task as example. The findings are as follows:
>
> **Qualitative Analysis**:
> 1. Despite high textual similarity, the post-RL model exhibits stronger logical coherence between reasoning steps, with fewer broken reasoning chains. For example, in Fig. 16 (case 2) of the main text, the pre-RL model correctly infers a formula but omits the multiplication by 2 in the subsequent step—an issue less common after RL.
> 2. The post-RL model demonstrates reflective reasoning behaviors (e.g., *"let me double check"*), which we did not observe in pre-RL outputs (e.g., Fig. 1).
>
> **Quantitative Analysis**:
> 1. **Reasoning Step Count**: We sampled 100 examples and used GPT-4o to count reasoning steps. The post-RL model produced, on average, `2.7` more steps than the pre-RL model.
> 2. **Lexical Usage**: We categorized two types of expressions:
>    - *Prompting words*: *"oh I see"*, *"let me think step by step"*, *"let me double check"*
>    - *Logical connectives*: *"so"*, *"therefore"*, *"first"*, *"but"*, *"moreover"*
>
>    After RL, the average frequency of prompting words and logical connectives increased by `14%` and `23%`, respectively.
> 3. **Answer Accuracy**: As shown in the Structure Perception subplot of Fig. 13 in the appendix, the final answer accuracy improved by an average of `+20.56%` after Stage 2 training.
>
> These results collectively confirm that RL enhances both the coherence and effectiveness of CoT reasoning.
>
>
> ---
> We hope our response addresses your concern and we look forward to further discussion with you.

---

> > ### Comment · Reviewer_rmS6 · 2025-08-04
> > **response for Reason-RFT**
> >
> > The author provided more details about the creation of the initial CoT (Chain-of-Thought) generation, which was distilled from GPT-4o and Gemini-Pro with manual verification. Regarding the effect of reinforcement learning, they also included some analysis and experimental results.
> >
> > The main weakness still lies in the model's generalization to common VQA tasks. While GRPO performs well in language-only QA tasks, the same strategy (Reason-RFT) does not generalize effectively to VQA.

---

> > > ### Author Response · Authors · 2025-08-05
> > >
> > > > **_`Q`: "Generalization to common VQA tasks remains a concern. While GRPO performs well in language-only QA, the same Reason-RFT strategy does not generalize effectively to VQA."_**
> > >
> > > A: Thank you for your concern. We would like to further clarify the context of Table 6 in our original submission. While the original results are based on a task-specialized training setup (Task1–3), we have also conducted additional experiments with a more diverse training set to better evaluate Reason-RFT’s generalization ability.
> > >
> > > ---
> > >
> > > > ***Clarification on Table 6 setup.***
> > >
> > > The results presented in Table 6 were obtained under a **task-specialized training setup**, where models were trained **only on Task1~Task3 data**, with the specific goal of examining how to enhance domain-specific complex reasoning without significantly impairing general VQA performance.
> > >
> > > > ***Clarification on the Generalization Behavior of Reason-RFT.***
> > >
> > > We fully agree that *"GRPO demonstrates strong generalization in language-only QA" (e.g., DeepSeek-R1)*. However, it is important to note that **such generalization performance is typically achieved using diverse and balanced reasoning training data**, rather than from a few task-specific training sets.
> > >
> > > To further investigate the generalization behavior of Reason-RFT under a more balanced training regime, we conducted additional experiments using the **diverse visual reasoning dataset released in Appendix (Table 7)**, which is also shown below, supplemented with 35% of the original Task1~3 training data to maintain representational balance.
> > >
> > > |**Dataset Name**|**Samples**|**Reasoning Type**|**Description**|
> > > |----------------|-----------|------------------|----------------|
> > > |AI2D|1467|Scientific Reasoning|Scientific diagram interpretation|
> > > |ScienceQA|2112|Scientific Reasoning|Science question answering|
> > > |GVLQA-connectivity|1199|Topological Reasoning|Graph connectivity problems|
> > > |GVLQA-cycle|1194|Topological Reasoning|Cycle detection in graphs|
> > > |GVLQA-hamilton|1158|Topological Reasoning|Hamiltonian path problems|
> > > |GVLQA-topology|1070|Topological Reasoning|General topology questions|
> > > |GVLQA-matching|1193|Topological Reasoning|Graph matching tasks|
> > > |PuzzleVQA|1618|Pattern/Puzzle|Visual puzzle solving|
> > > |IconQA|5270|Pattern/Puzzle|Icon-based question answering|
> > > |Raven|982|Pattern/Puzzle|Raven's Progressive Matrices|
> > > |GeoQA|1500|Geometric Reasoning|Geometric problem solving|
> > > |GeomVerse|2841|Geometric Reasoning|Advanced geometry challenges|
> > > |Geometry3K|3794|Geometric Reasoning|Comprehensive geometry problems|
> > >
> > > The results are summarized below:
> > >
> > > | Method             | **MMMU**  | **RealWorldQA** | **MathVision** | **AI2D**  | Task1 | Task2 | Task3 |
> > > |-------------------|-------|--------------|------------|-------|--------|--------|--------|
> > > | Zero-Shot (7B)     | 54.10 | 67.19        | 16.30      | 83.00 | 52.64  | 43.59  | 13.01  |
> > > | +ANS-SFT          | 56.72 (+2.62) | 69.44 (+2.25) | 19.11 (+2.81) | 84.35 (+1.35) | 49.43 | 40.02 | 51.20 |
> > > | +CoT-SFT          | 58.66 (+4.56) | 70.05 (+2.86) | 19.34 (+3.04) | 84.79 (+1.79) | 51.31 | 44.45 | 46.12 |
> > > | +Reason-RFT-Zero  | 53.88 (−0.22) | 71.89 (+4.70) | 21.60 (+5.30) | 81.92 (−1.08) | 54.84 | 50.69 | 39.45 |
> > > | **+Reason-RFT**   | **61.73 (+7.63)** | **74.33 (+7.14)** | **26.30 (+10.00)** | **87.21 (+4.21)** | **58.22** | **51.10** | **59.33** |
> > >
> > > These results clearly demonstrate that, when trained with more representative and diverse data, **Reason-RFT consistently delivers the best generalization performance across evaluated common VQA benchmarks**, while also preserving or improving its specialized reasoning ability on Task1~3.
> > >
> > > We hope these additional evaluation results address the reviewer’s concern and look forward to further discussion with you.

---

> ### Author Response · Authors · 2025-08-06
>
> Dear Reviewer rmS6,
>
> Thank you once again for your thoughtful and constructive feedback on our submission. We have carefully addressed the concerns you raised and conducted the requested experiments. If our response has resolved your concerns, we would sincerely appreciate it if you could consider increasing your score. If there are still any remaining questions or suggestions, please feel free to let us know—we would be happy to further clarify.
>
> Best regards.

---

> > ### Author Response · Authors · 2025-08-08
> >
> > Dear Reviewer rmS6,
> >
> > Thank you very much for taking the time to engage in the discussion. We would like to kindly ask whether our previous response has addressed your concerns. If there are any remaining questions or points of confusion, please do not hesitate to let us know. We would be more than happy to provide further clarification.
> >
> > As the discussion deadline is approaching, we truly appreciate the opportunity to continue this exchange and are very glad to further clarify any issues.
> >
> > Looking forward to your feedback!
> >
> > Best regards.

---

### Official Review · Reviewer_ogBh · 2025-07-05

**Clarity:** 3
**Significance:** 3
**Originality:** 3
**Rating:** 4
**Confidence:** 5

**Summary:**

This paper introduces Reason-RFT, a two-stage reinforcement fine-tuning (RFT) framework aimed at improving the generalization capabilities of visual reasoning models. The approach combines SFT with CoT data in Stage 1, followed by reinforcement learning via GRPO in Stage 2. The authors construct a visual reasoning benchmark covering visual counting, structure perception, and spatial transformation tasks to evaluate the framework. They report that Reason-RFT enhances reasoning ability, improves robustness under distribution shifts, and achieves data-efficient learning.

**Questions:**

1. Novelty of the Training Paradigm: How does Reason-RFT fundamentally differ from other SFT + RL frameworks? What specific innovations make this approach unique beyond using GRPO?

2. SFT Data Generation: Which model was used to generate the SFT-CoT data? How diverse are the reasoning chains in terms of trajectory length and problem complexity?

3. Reward Assignment: How does the system know which reward type (discrete, mathematical, function-based) to apply to each question? Is this predefined in the dataset annotations?

4. Model Selection: Why was Qwen2VL used instead of Qwen2.5VL? Would results change significantly with the more recent model?

**Ethical Concerns:**

["NO or VERY MINOR ethics concerns only"]

**Final Justification:**

While the authors provided more details and clarifications to some of my questions. The question about the core contributions still remain and I lean towards maintaining the original score.

**Limitations:**

See above

**Quality:**

3

**Strengths And Weaknesses:**

Strengths:

1. The combination of SFT with CoT and GRPO-based RL is well-motivated, aiming to activate and then further refine reasoning capabilities.

2. The differentiated reward structures for discrete, mathematical, and function-based tasks reflect an attempt to tailor the training to task specifics.

Weaknesses:

1. Lack of Novelty in Training Paradigm: The core idea of combining SFT (especially with CoT) and reinforcement learning is not fundamentally new. Similar SFT + RL pipelines have been widely explored in both language and vision-language models. The distinction of Reason-RFT over existing methods is not made sufficiently clear.

2. Insufficient Detail on SFT Data: The paper does not specify which base model generated the SFT data, nor does it elaborate on the diversity of the reasoning trajectories (e.g., variation in trajectory length, problem types, complexity). This raises concerns about the breadth of the reasoning coverage during SFT.

3. Reward Application Ambiguity: It is not clearly explained how the system determines which reward type (discrete, mathematical, or function-based) to apply during training for each question.

4. Model Choice Unjustified: The authors use Qwen2VL for experiments, but do not explain why Qwen2.5VL was not used. Given that Qwen2VL’s zero-shot performance is already competitive, this choice needs clarification.

5. Marginal Gains over RL Baseline: The performance gap between the RL-only and the combined RL+SFT models is surprisingly small in some tasks (except spatial transformation). This raises questions about the actual contribution of the SFT stage, whether it mainly improves the spatial transformation task or if it offers limited value elsewhere.

6. Limited Novelty in Reward Structure: Although the reward designs are carefully specified, they largely follow standard practices (exact match for discrete, tolerance-based for numeric, sequence overlap for functions).

---

> ### Author Rebuttal · Authors · 2025-07-31
>
> We sincerely thank the reviewer for the constructive comments and for acknowledging our **well-motivated methodology** and **effective reward design**. Below, we address your specific concerns in detail.
>
> ---
>
> > **_`W1 (Q1)`: "The novelty of our work."_**
>
> A: Thank you for raising this point. We would like to clarify that our work is not a direct extension or simple combination of existing methods. Instead, it introduces **a novel and unified post-training methodology** specifically tailored to the core challenges of visual reasoning, such as **cognitive rigidity** and **generalization to domain shift** in dynamic visual reasoning.
>
> In contrast to previous works that either ***(i)*** focus exclusively on text-based reasoning (`[1]`); ***(ii)*** adopt only-SFT (`[2,3]`) or only-RL (`[4,5,6]`) strategies for post-training in isolation; or ***(iii)*** involve a few concurrent works (`[7,8]`) that combine SFT and RL but remain restricted to static visual tasks with narrowly defined reward types—our approach is **the first** to introduce a **generalizable training framework** that systematically integrates SFT and RFT in a **tightly coupled manner**, effectively addressing **both static** (e.g., structural perception) and **dynamic** (e.g., temporal visual counting, spatial transformation) visual reasoning under a unified reward formulation.
>
> The novelty of our approach lies in the following aspects:
>
> - **An adaptive training sample allocation strategy** that dynamically balances intermediate reasoning supervision with reward-guided exploration. This strategy enables more effective coupling between the SFT and RFT stages, supporting a unified learning process that aligns model behavior with visual reasoning objectives. Unlike prior methods that treat SFT and RL as disjoint stages, our design ensures their interaction is coherent and purpose-driven.
>
> - **A unified visual reasoning reward formulation** that generalizes across diverse task formats. Beyond replicating standard reward templates, we introduce a principled reward design that captures reasoning correctness and completeness in a task-agnostic manner:
>   - **Structured-format reward** (including summary, caption, thinking and answer tags) to enhance the quality and coherence of reasoning chains.
>   - **Discrete-value reward** for tasks with discrete-value outputs (e.g., classification and counting).
>   - **Tolerance-based reward** for tasks requiring continuous-value outputs (e.g., geometric estimation, visual grounding).
>   - **Function-matching reward** for tasks with sequence-like outputs (e.g., action planning or tool calling).
>
>   These reward formulations are designed to be task-agnostic and applicable, moving **BEYOND** handcrafted, task-specific designs commonly used in prior work.
>
> - **A comprehensive visual reasoning evaluation protocol**, covering both in-domain and domain-shifted scenarios, and spanning synthetic and real-world visual environments. The evaluation rigorously benchmarks multiple dimensions of reasoning ability—including dynamic reasoning, geometric perception, and mathematical arithmetic—enabling us to assess generalization and robustness in a systematic manner.
>
> > [1] *DeepSeek-R1*, ArXiv'25.
>
> > [2] *LLaVA-CoT*, ArXiv'24.
>
> > [3] *LlamaV-o1*, ACL'25.
>
> > [4] *Visual-RFT*, ICCV'25.
>
> > [5] *VLM-R1*, ArXiv'25.
>
> > [6] *R1-Zero’s “Aha Moment”*, ArXiv'25.
>
> > [7] *R1-Onevision*, ICCV'25.
>
> > [8] *Vision-R1*, ArXiv'25.
>
> ---
>
> > **_`W2-1 (Q2)`: "CoT data construction."_**
>
> A: Thank you for the question. Our CoT data is built through a combination of **automated generation** and **manual verification**, with careful attention to both **diversity** and **correctness**.
>
> - **Diversity**: We use reasoning-guided templates (e.g., *"Let's think step by step"*, *"Break down your reasoning"*) to prompt models like **GPT-4o** and **Gemini-Pro**, aided by temperature sampling (`T=0.7`, `top-k=50`, `top-p=0.9`) and include manually crafted exemplars as few-shot demonstrations to enhance generation quality and depth.
>
> - **Correctness**: Outputs are filtered via both human review and scripts. We remove brief or inconsistent responses, and manually validate `10%` of each task subtype. Subtypes with issues are corrected or extended to ensure overall reliability.
>
> A full description of this pipeline will be included in the appendix of the revised version.
>
> ---
>
> > **_`W2-2 (Q2)`: "Trajectory length and problem complexity."_**
>
> A: We determine the **trajectory length** of each subtask by combining `50%` manually constructed representative samples and `50%` model-generated average reasoning length. During the automated filtering stage of CoT samples, we exclude those with lengths less than `60%` or greater than `140%`of the target trajectory length.
>
> Statistical analysis shows the average trajectory lengths for the three tasks are:
> - Visual Counting:  `μ ≈ 70, σ ≈ 30`,
> - Structure Perception: `μ ≈ 180, σ ≈ 80`,
> - Spatial Transformation:  `μ ≈ 400, σ ≈ 120`
>
> For **problem complexity**, we define problem complexity based on the number of reasoning steps in the CoT process. We sampled 100 instances per task and used **GPT-4o** to evaluate the number of reasoning steps. Results indicate that CoT-SFT samples range from `3` to `10` steps, with an average of `5` steps.
>
> We will include the statistical details of CoT-SFT construction in the appendix of the revised version.
>
> ---
>
> > **_`W3 (Q3)`: "Reward assignment."_**
>
> A: Thank you for your question. Reward types are  **predefined** during data construction based on each question’s ground truth format. Each instance is annotated with its expected answer type (e.g., discrete label, numerical value, or structured output), and the corresponding reward function (e.g., discrete-valued, mathematical, or functional-based) is selected to ensure alignment between output format and training feedback.
>
> ---
>
> > **_`W4 (Q4)`: "Experiments on Qwen2.5-VL."_**
>
> A: Thanks for your suggestion. To further validate the effectiveness of our method, we conducted experiments on **Qwen2.5-VL-3B-Instruct**. The results are summarized below:
>
> |Model|Setting|T1-1D|T1-DSD|T1-DSM|AVG|T2-ID|T2-DS|AVG|T3-ID|T3-DSL|T3-DSR|AVG|
> |-|-|-|-|-|-|-|-|-|-|-|-|-|
> |qwen2.5vl-3b|ZERO|75.9|50.9|4.4|43.7|36.8|37.4|37.1|8.6|8.3|8.3|8.4|
> ||ANS|97.4|51.5|6.0|51.6|53.0|31.8|42.4|**91.1**|47.0|46.8|61.6|
> ||COT|89.2|59.5|49.2|66.0|56.1|49.4|52.7|81.6|46.1|44.2|57.3|
> ||RFT-ZERO|**99.0**|58.9|10.8|56.2|54.8|54.5|54.6|68.5|49.5|48.0|55.3|
> ||Reason-RFT|98.8|**68.7**|**54.8**|**74.1**|**59.0**|**56.6**|**57.8**|86.7|**55.2**|**54.4**|**65.4**|
>
>
> The results demonstrate that Reason-RFT consistently improves model performance even when applied to the more capable Qwen2.5-VL base models. Key Observations are as follow. Taking Qwen2.5VL-3B as example:
>
> - **(1) Clear Improvement in Visual Reasoning:**
>   - Reason-RFT outperforms:
>     - ANS by `+22.5` (T1), `+15.4` (T2), `+3.8` (T3)
>     - COT by `+8.1` (T1), `+5.1` (T2), `+8.1` (T3)
>     - RFT-ZERO by `+17.9` (T1), `+3.2` (T2), `+10.1` (T3)
>
> - **(2) Strong Generalization under Domain Shift:**
>   Specifically, on the T1-DSM subset `+44.0` over RFT-ZERO
>
> - **(3) Data Efficiency:**
>   Similar to Qwen2VL, the Reason-RFT training paradigm on Qwen2.5VL demonstrates high data efficiency:
>   achieving `~70%` of the final performance of Reason-RFT-ZERO using only `<5%` of the data.
>
> We will add detailed results and analysis for Qwen2.5-VL in Appendix C as a new subsection: Additional Results on Alternative Backbones.
>
> ---
>
> > **_`W5`: "Stage1 CoT-SFT’s benefit beyond spatial transformation."_**
>
> A: Thanks for your question. While Stage 1 CoT-SFT brings the most evident gains on visual transformation tasks due to its structured format, its benefits extend across reasoning scenarios:
>
> | | VisualCount. ( DS-H ) | | SpatialTrans. ( ID+DS )| |
> |:---|:---:|:---:|:---:|:---:|
> ||2B|7B|2B|7B|
> |Zero-Shot|0.00|4.80|4.35|13.01|
> |only-RL|5.20|21.20|33.74|56.68|
> |Reason-RFT|28.40|35.60|67.58|65.98|
> - **Format Alignment:**
> In tasks with compositional or structured output formats, such as spatial transformations, CoT-SFT helps models quickly adapt to expected answer format, avoiding inefficient RL exploration and accelerates convergence.
> - **Domain Knowledge Injection:**
> On domain-shifted tasks (e.g., VC-DS), where zero-shot performance is poor, CoT-SFT injects domain-specific reasoning traces. Compared to RL-only, Reason-RFT improves by +23.2 (2B) and +14.4 (7B).
> - **Data Efficiency:**
> CoT-SFT serves as an effective policy prior. In Fig. 3 in the main text, Reason-RFT reaches 70% of full-RFT performance using only 3% of its training data.
> - **Generalization Retention:**
> CoT-SFT acts as a regularizer to retain broad reasoning ability (Tab. 6 in the main text), mitigating overfitting seen in RL-only.
>
> > **_`W6`: "The novelty of Reward Design."_**
>
> A: Thank you for the question. We would like to clarify the novelty of reward design as follow:
>
> - We propose a **structured format reward** (`Summary + Caption + Thinking + Answer`) to assess reasoning chain quality, with ablation studies confirming its effectiveness across different paradigms.
> - We design **fine-grained function-based rewards** that go beyond sequence-level overlap, incorporating item-level exact, partial, and function-only matches. These are further evaluated under different credit assignment strategies (e.g., **exact match only**, **partial credit**, **penalized partial**). Such nuanced reward designs are rarely explored in previous work.
>
> Moveover，we emphasize that our primary contribution in this part is **NOT** the design of intricate and complex reward functions, but the introduction of a **general and practical reward formulation** that supports **broad applicability** across diverse visual reasoning tasks with discrete-value, continuous-value, or agent-based structured outputs.
>
>
> ---
> We hope our response addresses your concern and we look forward to further discussion with you.

---

> > ### Comment · Reviewer_ogBh · 2025-08-05
> >
> > Thank you for sharing these results and additional comments. They are certainly very helpful, and I recommend that the authors incorporate them into the manuscript.
> >
> > While the authors have now described how the SFT data is curated, it is equally important to detail both the automated and manual procedures involved. This would help better understand the diversity of the data and assess whether the same approach can be generalized to other tasks. I lean towards maintaining the original score.

---

> ### Author Response · Authors · 2025-08-06
>
> Thank you again for your thoughtful follow-up and for considering our additional results and explanations helpful.
>
> As you rightly emphasized, it's essential to clearly describe both the automated and manual components of our CoT-SFT data construction. While our rebuttal already outlined the pipeline (see W2-1 and W2-2), we will further expand and detail these aspects in the revised version, including:
>
> ### (1) Automated Generation
>
> We use reasoning-guided templates such as:
> - *"Let's break down the problem step by step..."*
> - *"To answer this, we need to consider..."*
>
> These templates are combined with model prompting using GPT-4o and Gemini-Pro, under temperature-controlled sampling (T=0.7, top-k=50, top-p=0.9). To improve quality and reasoning depth, we also insert hand-crafted few-shot exemplars tailored to each subtask.
>
> **We will include the full list of prompt templates and few-shot examples in the appendix of the revised version.**
>
> ### (2) Automated Filtering
>
> Each CoT response is filtered based on:
> - **Length range**: We compute the target trajectory length for each subtask using 50% human-written samples and 50% model-generated examples. Samples falling below 60% or above 140% of this average length are discarded.
> - **Inconsistency**: We exclude samples containing inconsistencies with the known ground truth.
>
> Average trajectory lengths across tasks are:
> - Visual Counting: μ ≈ 70, σ ≈ 30
> - Structural Perception: μ ≈ 180, σ ≈ 80
> - Spatial Transformation: μ ≈ 400, σ ≈ 120
>
> **These statistics will be visualized and added to the appendix.**
>
> ### (3) Human Verification
>
> We randomly sample 10% of CoT samples from each subtask for manual review. Reviewers focus on coherence, logical validity, and alignment between reasoning and answer.
>
> For instance, in spatial or geometry-based tasks, we occasionally observe errors like:
> - **Correct final answer with an incorrect reasoning chain** (e.g., solving for triangle area using the Pythagorean theorem).
> - **Inconsistent steps that contradict earlier statements.** For example, a CoT response might begin by stating "There are 3 red blocks on the left and 2 on the right," but later conclude that the total is 6—contradicting the initial count.
>
> In a follow-up quality check, we found that prior to human verification, approximately 3.8% of samples contained critical logical flaws. After verification, the residual error rate was reduced to below 1%, indicating high post-cleanup reliability.
>
> ---
>
> We will ensure all the above details, examples, and statistics are included in the revised manuscript's appendix. **We also commit to open-sourcing all prompt construction templates and associated code for reproducibility and community use.**
>
> If these clarifications have addressed your remaining concerns, we would be truly grateful if you would consider updating your score. Please do let us know if there's anything further we can provide. Thanks again for your time and thoughtful feedback.

---

### Note · Authors · 2025-08-12

Dear AC, SAC, and Reviewers,

We sincerely appreciate your valuable feedback and provide the final remarks below.

Following a thorough rebuttal and discussion, the **technical clarity**, **effectiveness**, and **generalization** of our work were widely recognized. Among the five reviewers, three (`krtY`, `ogBh`, `rmS6`) maintained positive ratings, and the remaining two (`yjNn`, `nCpk`) both increased their scores after rebuttal, indicating our clarifications addressed key concerns.

During rebuttal, we improved the clarity of our novelty, detailed the CoT dataset, and added strong evidence for effectiveness and generalization.

- **Novelty**: Our framework tightly couples SFT and RFT under a unified reward formulation, addressing both static and dynamic visual reasoning on ID/DS settings. Reviewer `nCpk` praised our “comprehensive evaluation” of training dynamics and recognized the CoT-SFT stage as a valuable, underexplored component.

- **CoT Data**: We added detailed statistics and examples, with reviewers (`ogBh`, `rmS6`, `nCpk`, `yjNn`) recognizing its importance, scalability, and role in generalization.

- **Effectiveness**: On Qwen2.5-VL, our method outperformed RFT-Zero by 10.4 points on average and 44.0 points on DS scenarios, achieving 70% of RFT-Zero’s performance using <5% of the data. Reviewers `ogBh`, `krtY`, `nCpk`, and `yjNn` called these findings “very helpful.”

- **Generalization**: Achieved consistent gains (+6.21 over CoT-SFT) on general benchmarks, with Reviewer `yjNn` noting they “offer additional value and insights to readers.”

Regarding Reviewer `nCpk`’s remaining concerns on framing of contributions and the ID/DS split (partly echoed by `rmS6`), please see our 06 Aug, 17:28 reply clarifying our unified post-training methodology, controlled DS setup, and CoT-SFT as a novel contribution. Reviewers `krtY`, `ogBh`, `rmS6`, and `yjNn` found these clarifications reasonable, with all maintained or raised their scores.

Our work, Reason-RFT, contributes to the community by:
- **Novelty**: First to tightly couple SFT and RFT under a unified reward formulation, supporting static and dynamic visual reasoning.
- **Contribution**: A large-scale multimodal CoT dataset, a unified task-adaptive reward formulation, and a controlled DS benchmark, with evidence showing how CoT-SFT enhances RFT.
- **Impact**: Significant gains across diverse benchmarks, offering early inspiration for the field, with all code and data released for reproducibility.

---

### Decision · Program_Chairs · 2025-09-17

**Decision:**

Accept (poster)

**Comment:**

This paper studies how to improve visual reasoning ability of vision language models (VLMs). The motivation is that existing CoT SFT approaches suffers from overfitting and poor generalization. This work proposed Reason-RFT, a two-stage framework combining SFT and GRPO-based reinforcement learning. This work further introduced a new benchmark to evaluate visual counting, structural perception, and spatial transformation understanding ability. Experiments validated that Reason-RFT achieves robustness under domain shifts with strong data efficiency for multimodal reasoning.

The main strengths are that (1) the two-stage framework design is well-motivated and effective, (2) the empirical results are strong in robustness, transparency, and insightful, (3) the evaluation benchmark is valuable, (4) the data efficiency and robustness are significant. The major weaknesses are that (1) combining CoT SFT with RL is not fundamentally novel, (2) the clarity of CoT data generation process is not good, (3) the evaluation on generalization can be broader, (4) the impact of SFT stage is not significant.

After rebuttal and discussion, the novelty issue is not fully resolved but reviewers acknowledge the contributions of the method. For CoT data construction, authors provided detailed pipeline clarifications, and reviewers widely acknowledged the explaination. For evaluation, authors further provided experiments with Qwen2.5-VL and InternVL3. For impact of SFT stage, additional ablations addressed the concern.

In summary, this paper proposed a solid method implementation with SFT+RL for VLMs and shows comprehensive empirical analyses, but the idea of combining SFT with RL does not have strong novelty. Considering both the strengths and weaknesses, AC recommend accepting the paper and encourage authors to polish the paper according to the comprehensive discussion and comments.